# INFERENCE-TIME SCALING FOR TIME-SERIES PROCESSING

## ABSTRACT

Scaling laws have fundamentally driven AI progress, particularly in large-scale models. However, as Web-scale pretraining data for such models nears saturation, focus increasingly shifts to new paradigms like inference-time scaling. While validated across various AI domains, its application to time-series tasks remains largely unexplored. This study addresses this gap by investigating whether inference-time scaling can be successfully adapted for time-series processing. First, multiple candidate outputs for a given input are generated based on a trained model. Second, motivated by the principle that better candidates reconstruct the observed data more accurately, we compute the reconstruction error for each candidate output. Third, these errors are used to determine weights of each candidate, and the final prediction is then formed as a weighted combination of the candidates. We present specific algorithmic instantiations of this new framework for two fundamental time-series tasks, namely forecasting and missing value imputation. Furthermore, we provide a theoretical analysis for the forecasting case to support the method's validity from a Bayesian uncertainty perspective. Extensive experimental evaluation across 7 benchmark datasets for both tasks convincingly verifies the effectiveness of our methodology: Incorporation of our methodology during the inference phase led to performance improvements in all 9 recent time-series methods. Source codes have been uploaded in the supplementary files.

## 1 INTRODUCTION

Current advancements in AI models, particularly large-scale models, predominantly adhere to the scaling law, an empirical principle demonstrating scaling-law relationships between model performance and computational resources, parameter size, or pretraining data volume. However, this law faces an imminent crisis: high-quality pretraining data is projected to be exhausted in the near future, causing diminishing returns from traditional scaling methods. Consequently, more and more researchers are shifting focus toward inference-time scaling, which allocates additional computation during inference to iteratively refine outputs (Muennighoff et al., 2025). This approach has demonstrated efficacy in domains like mathematical reasoning and planning tasks. Time-series processing stands as a critical AI application domain, underpinning decision-making in healthcare, climate science, and industrial systems. Yet, despite the proliferation of specialized time-series models, inference-time scaling for time-series tasks remains severely under-explored. Current studies predominantly optimize training-phase scaling, while strategies to enhance inference-time performance, such as iterative refinement of forecasts in the inference stage for new inputs, lack systematic frameworks and empirical validation in temporal data contexts. Bridging this gap could unlock new frontiers in more effective time-series processing techniques.

Mainstream time-series models predominantly rely on classical sequential DNNs such as LSTM, TCN, and Transformer. Recent years have witnessed the emergence of time-series foundation models like Moirai (Woo et al., 2024), Moment (Goswami et al., 2024), and Time-MoE (Shi et al., 2025). With the rise of large language models (LLMs), researchers increasingly explore repurposing LLMs for temporal data processing, exemplified by Time-LLM's input reprogramming (Jin et al., 2024) and the LLM time-series forecaster in AutoTimes (Liu et al., 2024b). To our knowledge, only Liu et al. (2025b) have preliminarily explored self-consistency mechanisms for inference-time scaling in time-series tasks using LLMs as base processors, leaving applicability to mainstream time-series models

unexplored. We attempt to address this gap in scenarios using conventional sequential DNNs (while our weighting scheme is still applicable to the scaling strategy in Liu et al. (2025b)).

This study proposes a general inference-time scaling framework for time-series processing tasks by applying an ensemble method at inference time, comprises three key stages: First, we leverage a given trained model to generate multiple candidate outputs for a given input sequence. Second, each candidate output is utilized to reconstruct observable historical segments with reconstruction strategies adapted to task-specific objectives. Third, we compute reconstruction errors between generated sequences and ground-truth observations[1], converting these into normalized candidate weights. Finally, the framework outputs a refined prediction through weighted aggregation of candidates. We instantiate this inference-time scaling framework for two fundamental tasks: (1) forecasting, where the task aims to predict the time series in the next time period, and (2) missing value imputation, where the task aims to fill variable-length gaps assessed through observable points. Furthermore, we provide theoretical analysis to support our methodology.

We leveraged three model architectures (e.g., MLP, Transformer, and CNN) augmented with our framework against their vanilla counterparts. Rigorous analytical studies include sensitivity test for key hyper-parameters (e.g., dropout ratio, candidate number, and subsequence length) and computational cost analysis. Results demonstrate consistent performance gains: all involved SOTA methods are improved when using our inference-time scaling method, the maximum average MSE reductions achieved are 12.57% for forecasting and 32.67% for imputation. Generating more candidate outputs (i.e., assigning more test-time resources) helps mitigate prediction uncertainty and brings more gains.

Our contributions are summarized as follows:

- We proposed a general inference-time scaling framework for mainstream time-series models. To our knowledge, this is the first work that explores the potential of inference-time scaling for such models, which may open up more thoughts and techniques for time-series AI.
- For forecasting and imputation tasks, we develop tailored algorithms for the reconstruction of observed data, and perform a theoretical analysis for the usefulness of our proposed methodology.
- Seven time-series benchmark datasets across 3 DNN architectures are leveraged for empirical evaluation. The results demonstrates consistent gains of the proposed methodology over 9 SOTA methods, with comprehensive sensitivity analyses.

## 2 RELATED WORK

This section briefly reviews related studies in time-series processing and inference-time scaling.

### 2.1 TIME-SERIES PROCESSING

Deep learning offers powerful alternatives to traditional time-series analysis methods by automatically learning complex temporal patterns and dependencies. This subsection categorizes prominent deep learning approaches into two main streams: conventional DNNs and LLM adaptations.

Extensive research develops sequential DNNs for time series. RNNs (LSTM/GRU) capture long-term dependencies (Jia et al., 2024; Kong et al., 2025; Jia et al., 2023). Adapted CNNs extract hierarchical features via 1D convolutions (Luo & Wang, 2024; Wu et al., 2023; Wang et al., 2023a). Transformers process sequences in parallel, modeling long-range dependencies effectively. Variants like TFT handle covariates and known future inputs (Wang et al., 2024c). These models form the backbone of many SOTA solutions across finance, healthcare, and industrial monitoring.

Recent LLMs success has spurred interest in harnessing their sequence modeling capabilities for time series. This adapts pre-trained LLMs to model time-series data, facing key challenges: token-compatible encoding of continuous multivariate series (via patching, quantization, or specialized embeddings) and effective prompting design (Hu et al., 2025; Lu et al., 2025; Demirel & Holz, 2025). Techniques like modality adaptation (fine-tuning LLMs on time-series data) and prompt

---

[1]This study posits that superior candidates demonstrate stronger consistency with historical observations, underpinning our weighting scheme and inspired by LLM self-consistency mechanisms.

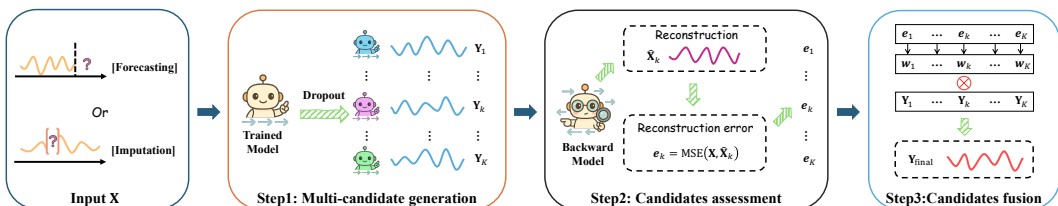

Figure 1: The architecture of inference-time scaling for time-series processing.

engineering are explored (Liu et al., 2025a). Models such as GPT4TS (Zhou et al., 2023) and AutoTimes using GPT2 (Radford et al., 2019) or LLaMA (Touvron et al., 2023) demonstrate that LLMs can achieve performance comparable to or surpassing conventional models by leveraging knowledge and reasoning from large-scale pretraining.

## 2.2 INFERENCE-TIME SCALING

Inference-time scaling adapts model behavior or confidence without weight modification, primarily for probabilistic forecasting calibration. It addresses DNNs' overconfidence and poor calibration through techniques like Temperature Scaling, where a scalar temperature $T$ divides the logits and is optimized on a validation set to adjust the output distribution and improve calibration metrics (Dabah & Tirer, 2025; Huang et al., 2025). Beyond calibration, it involves test-time adaptation (TTA) strategies that apply parameter-free or lightweight transformations to outputs or intermediate features (Lim et al., 2023; Shin & Kim, 2024), such as scaling attention weights, adjusting variance for uncertainty quantification, or normalizing features with streaming statistics (Wu et al., 2025; Chen et al., 2024; Zheng & Sun, 2024). These methods leverage inference-time information (e.g., input instances, batch statistics, or data drift) to enhance performance, reliability, and interpretability (Li & Rodríguez, 2025; Grover & Etemad, 2025; Fan et al., 2023).

Recent studies have also explored self-consistency (SC) and self-reflection (SR) as inference-time strategies for improving reasoning quality and correctness in LLMs. SC enhances robustness by sampling multiple reasoning paths (e.g., different chain-of-thought approaches) and aggregating the final answer via majority voting or weighted likelihood (Wang et al., 2023b; Jiang et al., 2025). SR introduces iterative self-evaluation steps, where the model critiques and revises its own outputs using verification prompts or feedback signals, effectively reducing logical errors, hallucinations, and factual inconsistencies (Shinn et al., 2023; Madaan et al., 2023; Zhang et al., 2024). These methods are appealing for their low computational overhead compared to full fine-tuning, making them practical for resource-constrained deployment and essential for building models that remain dependable and adaptive in real-world scenarios.

## 3 METHODOLOGY

### 3.1 THE MAIN FRAMEWORK

Assuming that we have a trained (forward) model $f_{\text{forward}}$ ready for a time-series inference task. Given historical observations $\mathbf{X} = \{x_1, \ldots, x_L\} \in \mathbb{R}^{L \times C}$ with $L$ time steps and $C$ variates, the task is to utilize $f_{\text{forward}}$ and $\mathbf{X}$ for prediction. Our proposed framework is illustrated in Fig. 1, which consists of three main steps:

- **Multi-candidate generation**: Although numerous methods exist for generating stochastic outputs, this study introduces stochasticity during inference via Monte Carlo Dropout (MC Dropout) (Gal & Ghahramani, 2016). By enabling dropout during inference and performing multiple forward passes, each with independent random neuron selections, we obtain a set of $K$ diverse candidate outputs $\bar{\mathbf{Y}} = \{\mathbf{Y}_1, \cdots, \mathbf{Y}_k, \cdots, \mathbf{Y}_K\}$, where each $\mathbf{Y}_k$ is generated by a dropout $f_{\text{forward}}$ with the input $\mathbf{X}$.
- **Candidates assessment**: This step evaluates the quality of the $K$ candidate outputs by employing reconstruction error as the evaluation metric in the present study. Specifically, each candidate $\mathbf{Y}_k$ is leveraged to reconstruct the historical observations (i.e., $\mathbf{X}$ or its part) and a reconstruction error can thus be calculated using the mean squared error (MSE). In this study, a backward time-series model $f_{\text{backward}}$ is trained as a supervised predictor to reconstruct historical observations of the input sequence along the temporal dimension, which will be detailed in the experiments.

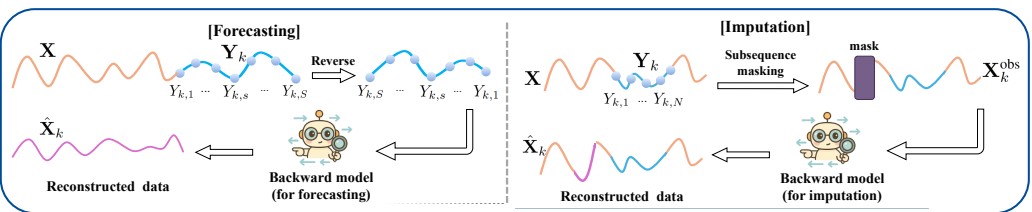

Figure 2: Reconstruction processes for two tasks.

- **Candidates fusion**: Let $e_k$ be the reconstruction error for $\mathbf{Y}_k$. Let $\mu_e$ and $\sigma_e$ be the mean and standard deviation for the errors. Based on the assumption that superior candidates demonstrate stronger consistency with historical data, we first normalize the errors using $z$-score normalization $\tilde{e}_k = (e_k - \mu_e)/\sigma_e$, and then the weight for $\mathbf{Y}_k$ is defined as $w_k = \frac{\exp(-\tilde{e}_k)}{\sum_{j=1}^{K} \exp(-\tilde{e}_j)}$. Finally, the inference result is computed as:

$$\mathbf{Y}_{\text{final}} = \sum_{k=1}^{K} w_k \cdot \mathbf{Y}_k = \sum_{k=1}^{K} \frac{\exp(-e_k/\sigma_e)}{\sum_{j=1}^{K} \exp(-e_j/\sigma_e)} \cdot \mathbf{Y}_k, \tag{1}$$

  where $\mu_e$ is finally removed according to mathematical simplification. Our theoretical analysis will show that such weight definition and ensemble-fusion schemes can be well explained with Bayesian uncertainty theory.

As mentioned in Step 2, which employs a reconstruction process for candidates assessment, this process varies across time-series tasks. The following subsections take two representative tasks, namely forecasting and imputation as examples to describe the concrete reconstruction details.

## 3.2 RECONSTRUCTION FOR TIME-SERIES FORECASTING

Given $\mathbf{X}$, the forecasting task is to predict the next $S$ time steps $\mathbf{Y} = \{X_{L+1}, \ldots, X_{L+S}\} \in \mathbb{R}^{S \times C}$. As illustrated in the forecasting part of Fig. 2, the reconstruction process leverages predicted data (e.g, $\mathbf{Y}_k$) to perform reverse-temporal prediction of past data (e.g., $\mathbf{X}$), for which we employ a backward model. This model is elaborated in the experimental section and further detailed in Appendix E.

**Implementation details.** This section describes the reconstruction process for the autoregressive framework. Our method also supports non-autoregressive frameworks, where the forward model predicts all of the prediction horizon at once and the backward model reconstructs over the full forecast horizon. Further details are provided in Appendix A.1. In autoregressive frameworks, the prediction horizon $S$ is divided into $M = \lceil S/F \rceil$ segments, where $F$ is the fixed forecast window size, specifying the number of time steps predicted at once. The model forecasts these segments sequentially. For the $m$th segment, Step 1 uses $f_{\text{forward}}$ to generate $K$ candidates $\mathbf{Y}_k^{(m)}, k = \{1, \cdots, K\}$ for the next $F$ points. Each candidate $\mathbf{Y}_k^{(m)}$ is reversed along the temporal dimension:

$$\mathbf{Y}_k^{(m),\text{rev}} = \text{Reverse}(\mathbf{Y}_k^{(m)}) = \text{Reverse}[f_{\text{forward}}(\mathbf{X}^{(m)})], \tag{2}$$

where $\mathbf{Y}_k^{(m),\text{rev}} \in \mathbb{R}^{F \times C}$ and then $\hat{\mathbf{X}}_k^{(m)}$ is reconstructed by the backward model as follows:

$$\hat{\mathbf{X}}_k^{(m)} = \text{Reverse}[f_{\text{backward}}(\mathbf{Y}_k^{(m),\text{rev}})], \tag{3}$$

where $\hat{\mathbf{X}}_k^{(m)} \in \mathbb{R}^{F \times C}$. The reconstruction error between $\hat{\mathbf{X}}_k^{(m)}$ and its corresponding part in $\mathbf{X}$ is calculated as $e_k^m$. The inference-time scaling (ITS) method for forecasting is shown in Algorithm 1.

## 3.3 RECONSTRUCTION FOR IMPUTATION

In the imputation scenario, when considering a single[2] missing subsequence of length $N$, the sequence $\mathbf{X}$ can be expressed as $\{x_1, \ldots, x_{l_1}, x_{l_1+N+1}, \ldots, x_L\}$. As illustrated in the imputation part of Fig. 2, a random subsequence of the past $\{x_1, \ldots, x_{l_1}\}$ or future $\{x_{l_1+N+1}, \ldots, x_L\}$ observable data is

---

[2]Missingness typically occurs in contiguous subsequences rather than isolated points in practice (Liu et al., 2024c). While multiple subsequences may be missing, we illustrate using a single-subsequence case.

---

**Algorithm 1** Inference-time scaling (ITS) for forecasting

---

**Input**: $\mathbf{X} \in \mathbb{R}^{L \times C}$, $M$, $F$, $\mathbf{X}^{(0)} = \mathbf{X}$, $\mathbf{Y}_{\text{final}}^{(0)} = \emptyset$, $f_{\text{forward}}$, $f_{\text{backward}}$.
**Output**: Next $S$ time steps $\mathbf{Y}_{\text{final}} \in \mathbb{R}^{S \times C}$.
 1: Enable dropout for MC sampling.
 2: **for** $m = 1$ **to** $M$ **do**
 3:     Concatenate $\mathbf{X}^{(m)} = \text{Concat}\{\mathbf{X}^{(m-1)}, \mathbf{Y}_{\text{final}}^{(m-1)}\}$, taking the last $L$ time steps as the next input.
 4:     Generate $K$ candidate outputs via MC Dropout.
 5:     **for** $k = 1$ **to** $K$ **do**
 6:         Infer $\mathbf{Y}_k^{(m),\text{rev}}$ with Eq. (2).
 7:         Infer $\hat{\mathbf{X}}_k^{(m)}$ with Eq. (3).
 8:         Compute $e_k^m = \text{MSE}(\mathbf{X}_{L-F+1:L}^{(m)}, \hat{\mathbf{X}}_k^{(m)})$.
 9:     **end for**
10:     Calculate the weights and generate $\mathbf{Y}_{\text{final}}^{(m)}$ with Eq. (1).
11: **end for**
12: Concatenate all segments $\mathbf{Y}_{\text{final}}^{(1)}, \ldots, \mathbf{Y}_{\text{final}}^{(M)}$.
13: Truncate $\mathbf{Y}_{\text{final}}$ to length $S$ if $S \bmod F \neq 0$.
14: **return** $\mathbf{Y}_{\text{final}}$

---

selected for masking and the backward model is leveraged to reconstruct it based on masked $\mathbf{X}$ and one candidate $\mathbf{Y}_k$. This simple yet effective strategy is validated by our experiments. In our implementation, more than one subsequence are selected.

**Implementation details.** We first divide $\mathbf{X}$ into $P = L/N$ non-overlapping subsequences, and one of these subsequences is masked. Although multiple past or future observable subsequences could be reconstructed, we only consider a single subsequence for simplicity.

We first apply a masking function to randomly obscure one subsequence for $\mathbf{X}$. For each candidate output $\mathbf{Y}_k$ for imputation, we insert it into the masked $\mathbf{X}$ to form a filled sequence, resulting in:

$$\mathbf{X}_k^{\text{obs}} = \text{Insert}[\text{Mask}(\mathbf{X}, q), \mathbf{Y}_k], \tag{4}$$

where $q$ denotes the masking rate. Finally, $\mathbf{X}_k^{\text{obs}}$ is reconstructed using $f_{\text{backward}}$ within the inference-time scaling:

$$\hat{\mathbf{X}}_k = f_{\text{backward}}(\mathbf{X}_k^{\text{obs}}). \tag{5}$$

The MSE between the masked subsequence and its corresponding part in $\mathbf{X}$ is calculated. While $f_{\text{forward}}$ can also be used for reconstruction, our experiments show that the trained backward model yields substantially better performance. The overall inference-time scaling method (ITS) for imputation is shown in Appendix A.2.

## 3.4 THEORETICAL ANALYSIS

This subsection shows that the proposed inference-scaling strategy can be explained via Bayesian uncertainty theory which is effective for epistemic uncertainty modeling[3]. Gal & Ghahramani (2016) proposed a Bayesian approximation method based on MC Dropout. Given a trained model $\boldsymbol{W}_0$, they sampled models with the manner:

$$\boldsymbol{W}_k = \text{Dropout}(\boldsymbol{W}_0, \mathbf{z}_k), \quad \mathbf{z}_k \sim \text{Bernoulli}(p_d), k = 1, \cdots, K, \tag{6}$$

where $\mathbf{z}_k$ represents dropout variables sampled from the Bernoulli distribution parameterized by $p_d$ in the $k$th run. Eq. (6) utilizes $p(\mathbf{z})$ to approximate $p(\boldsymbol{W}|D)$, i.e., $p(\boldsymbol{W}|D) \sim p(\mathbf{z})$ where $D$ is the training corpus, and $p(\boldsymbol{W}|D)$ is distribution of the model $\boldsymbol{W}$. The inference for a sample $\boldsymbol{x}$ becomes:

$$p(y|\boldsymbol{x}, D) \approx \sum_{\mathbf{z}} p(y|\boldsymbol{x}, \boldsymbol{W}_0, \mathbf{z})p(\mathbf{z}) \approx \frac{1}{K} \sum_k p(y|\boldsymbol{x}, \boldsymbol{W}_k). \tag{7}$$

Even though the above approximation achieves great success in various applications, $p(\mathbf{z})$ provides only a structural approximation to $p(\boldsymbol{W}|D)$, derived solely from the neural network architecture. In

---

[3]Due to lack of space, a full version with more details of this subsection is presented in Appendix B.

practice, additional factors, such as model performance, can contribute to a more refined approximation. Motivated by this rationale, we define the following approximate formulation:

$$p(\boldsymbol{W}|D) \sim p(\mathbf{z})f[E(\boldsymbol{W})], \tag{8}$$

where $E(\boldsymbol{W})$ denotes model error (e.g., classification error). $f[E(\boldsymbol{W})]$ can be defined as:

$$f[E(\boldsymbol{W})] \sim \begin{cases} \frac{1}{\sigma}\exp\left[-\frac{E(\boldsymbol{W})-E^*}{\sigma}\right], & E(\boldsymbol{W}) \leq E^* \\ 0, & \text{Otherwise} \end{cases}, \tag{9}$$

where $E^*$ is the lowest model error trained on $D$. This definition implies that models exhibiting performance closer to $E^*$ attain higher posterior probability mass. Accordingly, Eq. (7) becomes:

$$\begin{aligned} p(y|\boldsymbol{x}, D) &\approx \frac{1}{\sum_j f[E(\boldsymbol{W}_j)]} \sum_k p(y|\boldsymbol{x}, \boldsymbol{W}_k)f[E(\boldsymbol{W}_k)] \\ &= \sum_k \frac{f[E(\boldsymbol{W}_k)]}{\sum_j f[E(\boldsymbol{W}_j)]} p(y|\boldsymbol{x}, \boldsymbol{W}_k) = \sum_k w_k p(y|\boldsymbol{x}, \boldsymbol{W}_k) \end{aligned}, \tag{10}$$

where $E^*$ is mathematically removed. There are three typical cases for Eq. (10):

- If $\sigma \to +\infty$, then $f[E(\boldsymbol{W}_k)]$ is a constant. Thus, $w_k \equiv \frac{1}{K}$ and Eq. (10) is reduced to Eq. (7).
- If $\sigma \to 0$, then only the best model achieving the minimum error among the $K$ models has a weight of 1 and the weights of the rest models are 0 (i.e., Winner takes all). Let $\hat{\boldsymbol{W}}^*$ be the best model among the sampled ones. Then, $p(y|\boldsymbol{x}, D) = p(y|\boldsymbol{x}, \hat{\boldsymbol{W}}^*)$.
- Otherwise, with mathematical simplification, Eq. (10) becomes:

$$p(y|\boldsymbol{x}, D) = \sum_k \frac{\exp\left[-E(\boldsymbol{W}_k)/\sigma\right]}{\sum_j \exp\left(-E(\boldsymbol{W}_j)/\sigma\right]} p(y|\boldsymbol{x}, \boldsymbol{W}_k). \tag{11}$$

The first case corresponds to the most common manner, namely, Majority Voting (or Averaging) when confronted with multiple outputs, which is also employed in Liu et al. (2025b). The second and the third cases require the assessment of model performance, which is still infeasible for real applications. We can denote the reconstruction error for the current input sample as an assessment of the involved model, namely, $E(\boldsymbol{W}_k) \sim e_k$. Eq. (10) then becomes:

$$p(y|\boldsymbol{x}, D) = \sum_k \frac{\exp(-e_k/\sigma)}{\sum_j \exp(-e_j/\sigma)} p(y|\boldsymbol{x}, \boldsymbol{W}_k), \tag{12}$$

which equals to Eq. (1). That is to say, the weighted scheme proposed earlier essentially serves as a ensemble method to mitigate model uncertainty and thus outputs more robust results.

## 4 EXPERIMENTS

We extensively evaluate the proposed inference-time scaling (ITS) method on benchmark datasets and SOTA methods, covering both forecasting and imputation tasks.

**Datasets.** Seven benchmark datasets are used, namely, ETT (including 4 corpora: ETTh1, ETTh2, ETTm1, ETTm2) (Zhou et al., 2021), Electricity (Li et al., 2019), Weather (Zhou et al., 2021), and Exchange (Wu et al., 2021). We adopt the standard data splitting strategy from Zhou et al. (2021), dividing each dataset into train, validation, and test. The ETT datasets are chronologically split with a 3:1:1 ratio, whereas the other datasets with a 7:1:2 ratio.

**Methods.** Nine SOTA methods spanning various classical sequential DNN architectures are involved, including iTransformer (Liu et al., 2024a), TimeMixer (Wang et al., 2024a), TimeXer (Wang et al., 2024c), PatchTST (Nie et al., 2023), Crossformer (Zhang & Yan, 2023), TimesNet (Wu et al., 2023), MICN (Wang et al., 2023a), Autoformer (Wu et al., 2021), and Timer (Liu et al., 2024c). It should be noted that our methodology is still applicable for LLMs-based architectures. Appendix F.4 presents that our method achieves consistent performance improvements on these architectures.

Table 1: Average MSE results for forecasting with $S \in \{96, 192, 336, 720\}$ and $L = 96$. Vanilla denotes the original methods, and **+ITS** denotes that our ITS method is used in the inference stage, with full results provided in Appendix D.1.

| Method | iTransformer | | TimeMixer | | TimeXer | | PatchTST | | Crossformer | | TimesNet | | MICN | | Autoformer | | Timer | |
|---|---|---|---|---|---|---|---|---|---|---|---|---|---|---|---|---|---|---|---|
| Dataset | Vanilla | +ITS | Vanilla | +ITS | Vanilla | +ITS | Vanilla | +ITS | Vanilla | +ITS | Vanilla | +ITS | Vanilla | +ITS | Vanilla | +ITS | Vanilla | +ITS |
| ETTh1 | 0.469 | **0.464** | 0.459 | **0.454** | 0.457 | **0.450** | 0.464 | **0.462** | 0.644 | **0.622** | 0.464 | **0.460** | **0.432** | 0.433 | 0.504 | **0.503** | 0.408 | **0.390** |
| ETTh2 | 0.400 | **0.398** | 0.388 | **0.387** | 0.373 | **0.369** | 0.382 | 0.382 | 0.970 | **0.963** | 0.408 | **0.405** | 0.415 | **0.413** | 0.429 | **0.423** | 0.378 | **0.364** |
| ETTm1 | 0.421 | **0.418** | 0.386 | **0.374** | 0.384 | **0.373** | 0.393 | **0.381** | 0.454 | **0.448** | 0.408 | **0.394** | 0.388 | **0.383** | 0.537 | **0.506** | 0.388 | **0.351** |
| ETTm2 | 0.292 | **0.286** | 0.278 | **0.272** | 0.277 | **0.274** | 0.284 | **0.272** | 0.707 | **0.690** | 0.317 | **0.314** | 0.287 | **0.285** | 0.311 | **0.309** | 0.290 | **0.270** |
| Exchange | 0.366 | **0.359** | 0.443 | **0.422** | 0.368 | 0.362 | 0.377 | **0.371** | 0.934 | **0.923** | 0.417 | **0.414** | 0.309 | **0.307** | 0.532 | **0.491** | 0.373 | **0.362** |
| Weather | 0.274 | **0.271** | 0.245 | **0.240** | 0.241 | 0.242 | 0.258 | **0.252** | 0.250 | **0.249** | 0.277 | **0.276** | 0.258 | **0.252** | 0.358 | **0.313** | 0.240 | **0.229** |
| Electricity | 0.180 | **0.171** | 0.184 | **0.182** | 0.173 | **0.164** | 0.215 | **0.211** | 0.514 | **0.511** | **0.204** | 0.205 | 0.185 | **0.175** | 0.262 | **0.257** | 0.167 | **0.164** |

**Evaluation protocol.** The proposed ITS method is primarily designed to enhance the inference-time performance of existing methods. Consequently, our evaluation protocol does not involve a direct comparison between ITS and existing SOTA methods. Instead, it focuses on comparing the same existing method during inference without ITS (denoted as vanilla) with its performance when augmented with ITS (denoted as '+ITS'). A lower Mean Squared Error (MSE) or Mean Absolute Error (MAE) for the '+ITS' configuration compared to the vanilla configuration in the test data demonstrates the effectiveness of our ITS method. Due to page limitation, the comparison on MAE is shown in the Appendix D.1.

## 4.1 FORECASTING

**Setup.** All involved methods are trained in PyTorch using their original code and hyperparameters, and Timer fine-tuned on the released checkpoint. Experiments are conducted on a single NVIDIA A100 80GB GPU. In the vanilla setting, we use a look-back window of $L = 96$ (with Timer using $L = 672$) and prediction length $S \in \{96, 192, 336, 720\}$. For ITS, we retain these settings while the number of candidates $K$ is set to 64 and keeping the dropout ratio $p_d$ consistent with each model's training setup. The backward model's look-back and prediction windows are both 96, corresponding to the segmentation and reconstruction.

The backward model is constructed as follows: First, the 12G-UTSD[4] (Liu et al., 2024c) dataset is used, and all sequences are reversed. A base backward model is then pretrained using the Timer architecture (Liu et al., 2024c) on this reversed dataset. When applying the model to a specific dataset, it is fine-tuned using reversed sequences derived from a portion of that dataset's training sequences. Once fine-tuned, this backward model can be employed for reconstruction. Further implementation details, along with its comparison with the forward model, are provided in the Appendix E.1.

**Results.** Table 1 and Fig. 3(a) present the average MSE for both the vanilla methods and the methods with ITS, evaluated across various architectures and prediction lengths. ITS consistently reduces forecasting errors at inference time, effectively mitigating the uncertainty inherent in time-series models. On datasets with fewer variables and simpler dynamics, including the four ETT and Exchange datasets, ITS almost always surpasses the vanilla methods, achieving average MSE reductions of 4.74% for TimeMixer on Exchange and 9.54% for Timer on ETTm1. On more complex datasets with larger variable counts, namely Electricity and Weather, ITS also demonstrates strong competi-

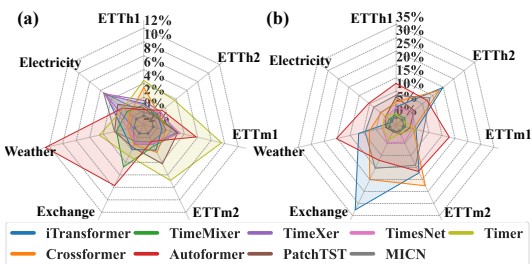

Figure 3: Average MSE reductions ratio ($\frac{\text{MSE}_{\text{vanilla}} - \text{MSE}_{\text{ITS}}}{\text{MSE}_{\text{vanilla}}} \times 100\%$) results achieved by our proposed inference-time scaling method (ITS) on 9 classical time-series models. The left shows forecasting results, and the right shows imputation result.

tiveness, with MSE reductions of 12.57% for Autoformer and 4.58% for Timer on Weather. These results indicate that ITS is particularly effective in forecasting scenarios with high uncertainty.

In the Theoretical Analysis Section 3.4, we note that as the parameter $\sigma$ approaches either 0 or positive infinity, the fusion strategy converges to two distinct common schemes: 1) Majority Voting (Majority),

---

[4]12G-UTSD is tailored for pretraining time-series foundation models and is not used as an evaluation dataset in previous studies.

Table 2: Average MSE results for imputation by randomly masking $\{12.5\%, 25\%, 37.5\%, 50\%\}$ of subsequences within time series of length $L = 96$. Full results are provided in Appendix D.1.

| Method | iTransformer | | TimeMixer | | PatchTST | | Crossformer | | TimesNet | | MICN | | Autoformer | | Timer | |
|---|---|---|---|---|---|---|---|---|---|---|---|---|---|---|---|---|
| **Dataset** | Vanilla | +ITS | Vanilla | +ITS | Vanilla | +ITS | Vanilla | +ITS | Vanilla | +ITS | Vanilla | +ITS | Vanilla | +ITS | Vanilla | +ITS |
| ETTh1 | 0.320 | **0.307** | 0.308 | **0.304** | 0.384 | **0.354** | 0.349 | **0.328** | 0.291 | **0.280** | 0.367 | **0.336** | 0.489 | **0.426** | 0.330 | **0.325** |
| ETTh2 | 0.313 | **0.251** | 0.135 | **0.133** | 0.150 | **0.141** | 0.361 | **0.296** | 0.125 | **0.118** | 0.471 | **0.407** | 1.484 | **1.304** | 0.154 | 0.154 |
| ETTm1 | 0.240 | **0.228** | 0.215 | **0.213** | 0.232 | **0.226** | 0.176 | **0.169** | 0.241 | **0.232** | 0.191 | **0.176** | 0.472 | **0.388** | 0.459 | **0.434** |
| ETTm2 | 0.145 | **0.120** | 0.078 | 0.078 | 0.081 | **0.080** | 0.137 | **0.106** | 0.068 | **0.065** | 0.241 | **0.207** | 1.093 | **0.911** | 0.126 | **0.125** |
| Exchange | 0.101 | **0.068** | 0.013 | 0.013 | 0.014 | 0.014 | 0.564 | **0.451** | 0.021 | **0.020** | 0.176 | **0.149** | 1.598 | **1.406** | 0.029 | **0.028** |
| Weather | 0.111 | **0.098** | 0.083 | **0.082** | 0.078 | **0.077** | 0.079 | **0.073** | 0.075 | **0.073** | 0.087 | **0.079** | 0.370 | **0.295** | 0.119 | **0.116** |
| Electricity | **0.111** | 0.112 | 0.138 | **0.137** | 0.172 | **0.168** | 0.198 | **0.189** | 0.141 | **0.138** | 0.200 | **0.184** | 0.375 | **0.336** | 0.174 | **0.169** |

which assigns equal weights to all candidates and simply averages their outputs, 2) Winner-Take-All (WTA), which selects only the candidate with the lowest reconstruction error. Therefore, within the ITS framework, we compare three weighting schemes: Majority, WTA, and our proposed reconstruction-based strategy (i.e., $\sigma = \sigma_e$). The results are shown in the left of Fig. 4 under different prediction lengths on the ETTm1 dataset when Crossformer is used. Our adopted scheme achieves the best performance. It can be observed that Majority Voting (also adopted by Liu et al. (2025b)) demonstrates the poorest performance. While the WTA strategy can enhance performance in certain cases, its effectiveness significantly deteriorates when both sequence length and mask ratio are substantial. We also discuss a weighting scheme based on the candidates' own uncertainty, and complete results and discussions are provided in Appendix F.1.

## 4.2 IMPUTATION

**Setup.** All involved methods are trained using their code and default hyperparameters, and Timer fine-tuned on the released checkpoint. For tasks without prior implementations, results are reported using the benchmark framework provided by the Time-Series Library (Wang et al., 2024b). TimeXer is excluded due to the lack of implementation for this task in the library, ensuring fairness. In the vanilla setting, we set the subsequence length to $N = 12$ (with Timer using $N = 24$) and the imputation length to $L = 96$, with random mask ratios of $\{12.5\%, 25\%, 37.5\%, 50\%\}$. In the ITS setting, we keep these configurations and set the candidate outputs $K = 64$, while maintaining the dropout ratios $p_d$ used during training. The mask rate for reconstruction is fixed at $q = 0.3$, and the backward model is set to imputation length $L = 96$ and subsequence length $N = 12$.

Similarly, we employ the backward model architecture originally developed for the forecasting task and adopt an analogous fine-tuning strategy. Although the use of a backward model is not theoretically mandatory in this context, our empirical evidence demonstrates that utilizing the backward model yields superior performance compared to employing the forward model. Specific details regarding this comparison are provided in the Appendix E.2.

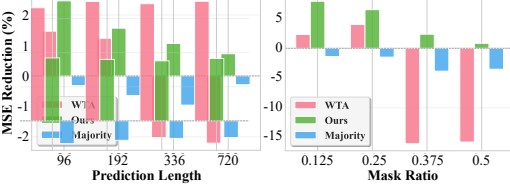

Figure 4: MSE reductions of Crossformer achieved by three schemes on the ETTm1 dataset with $L = 96$. The left shows forecasting results, and the right shows imputation results.

Figure 5: MSE reductions of TimesNet achieved by different $K$ values on the ETTm1 dataset with $L = 96$. The left shows forecasting results, and the right shows imputation results.

**Results.** Table 2 and Fig. 3(b) report the average MSE results of the vanilla methods and their counterparts with the proposed ITS for imputation. ITS consistently outperforms the vanilla methods across all datasets and models. On datasets with fewer variables and simpler temporal dependencies, such as the four ETT datasets and Exchange, ITS delivers particularly strong improvements. For example, on ETTh2, iTransformer reduces the MSE by 19.81%, while Crossformer achieves an 18.01% reduction. On Exchange, the MSE of iTransformer decreases by 32.67%, while Crossformer decreases by 20.04%. Even on complex, high-dimensional datasets such as Weather and Electricity, ITS remains competitive and often achieves noticeable gains; for instance, on Weather, the MSE of

iTransformer decreases by 11.71% and MICN decreases by 9.20%. Moreover, the foundation model Timer, pretrained with large-scale data and already exhibiting strong performance, still benefits from ITS with reductions of 5.45% and 2.87% on ETTm1 and Electricity, respectively. These results highlight the robustness of ITS and confirm its ability to enhance imputation quality across all methods. To further demonstrate the effectiveness of ITS across subsequence lengths, we investigate the impact of $N \in \{3, 6, 24\}$ in Appendix D.2. The analysis shows that 93.37%, 93.88%, and 83.69% of the test cases achieve MSE reductions under these settings, indicating that ITS remains robust across varying missing lengths.

Likewise, we evaluated the performance of three weighting strategies for imputation across varying mask ratios as shown in the right part of Fig. 4. It can be observed that our proposed scheme consistently outperforms others, while the Majority strategy performs the poorest.

## 4.3 DISCUSSION

**Sensitivity analysis for the number of candidates.** We examine the effect of the number of candidates on the ITS by testing different values of $K \in \{8, 16, 32, 64, 128\}$, as shown in Fig. 5. Increasing $K$ consistently reduces MSE, indicating that using more candidate outputs helps mitigate the model's prediction uncertainty, although the improvement tends to saturate when $K$ becomes excessively large. Details are provided in Appendix F.2.

Table 3: Average MSE results for LLMs-based models on ETT datasets with $S \in \{96, 192, 336, 720\}$ and $L = 672$. Full results are provided in Appendix F.4.

| Method | ETTh1 | ETTh2 | ETTm1 | ETTm2 |
|---|---|---|---|---|
| **AutoTimes** (Vanilla) | 0.393 | 0.364 | 0.352 | 0.268 |
| **AutoTimes** (+ITS) | **0.385** | **0.358** | **0.349** | **0.264** |
| **GPT4TS** (Vanilla) | 0.430 | 0.366 | 0.353 | 0.267 |
| **GPT4TS** (+ITS) | **0.426** | **0.357** | **0.342** | **0.259** |

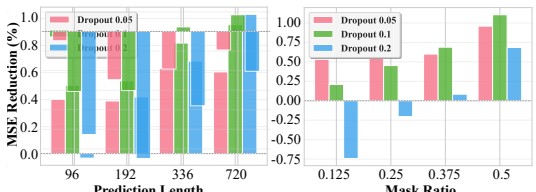

Figure 6: MSE reductions of TimeMixer under different $p_d$ values on the Electricity dataset with $L = 96$. The left shows forecasting results, and the right shows imputation results.

**Sensitivity analysis for dropout ratio $p_d$.** We further analyze the effect of the dropout ratio $p_d$ on ITS. As shown in Fig. 6, experiments with varying dropout ratios reveal that a moderate value, which matches the optimal setting used during pretraining, achieves the best balance between predictive accuracy and sufficient diversity among candidate outputs. Details are provided in Appendix F.3.

**Application to LLMs-based model.** We show that ITS further enhances the strong capabilities of LLMs-based time-series processing. As reported in Table 3, ITS consistently reduces the MSE of GPT4TS (Zhou et al., 2023) and AutoTimes (Liu et al., 2024b) on ETT datasets with $K = 32$ candidates. These results indicate that ITS is compatible with LLMs-based models and can effectively improve their inference performance. Additional experimental configurations and complete results are provided in Appendix F.4.

**Analysis for the computational cost.** We evaluate the computational cost of the proposed ITS on the ETTh2 dataset with $K = 64$ candidate outputs. For fairness, we adopt the batch sizes specified by the original authors for each method. As shown in Fig. 7, iTransformer and TimeMixer introduce minimal overhead, whereas more complex architectures (e.g., MICN, TimesNet, Autoformer) introduce higher overhead. Overall, despite ITS involves extra operations, such as generating candidate outputs, performing reconstruction, and applying a weighting scheme, the computational overhead remains moderate. Over 85% of the total inference time is primarily consumed by MC Dropout sampling.

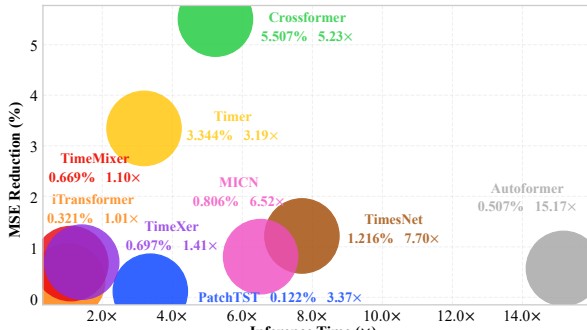

Figure 7: Computational cost and MSE reduction of ITS over 9 models for forecasting with input-96-predict-96 on ETTh2.

Improving MC Dropout has the potential to substantially decrease its execution time. Details for both tasks is provided in Appendix F.5.

## 5 CONCLUSIONS

This study constructs an inference-time scaling framework to enhance inference performance for two fundamental time-series tasks: forecasting and imputation. First, generates multiple candidates from a trained forward model via MC Dropout. Next, reconstruct each candidate output with a backward model and calculating reconstruction errors. Finally, converting these errors into weights to aggregate the candidate outputs for the final result. Our approach achieves improved accuracy across diverse datasets and architectures, demonstrating strong generalizability and effectiveness. Future work will explore more efficient candidates generating strategies and more sophisticated weighting schemes.

## ETHICS STATEMENT

Our work only focuses on the scientific problem, so there is no potential ethical risk.

## REPRODUCIBILITY STATEMENT

We are committed to ensuring the reproducibility of our research. To this end, we provide comprehensive details of our experimental setup, baseline implementations, and evaluation procedures in Section 4 and a detailed appendix (Appendices C, E, etc.). The appendix includes dataset descriptions, metric calculation and experiment configuration. Furthermore, all code, scripts, and trained model checkpoints required to reproduce the results presented in this paper will be made available in an Anonymized repository (https://anonymous.4open.science/r/Inference-time-Scaling-22C8) and in the Supplementary Material.

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

## USE OF LARGE LANGUAGE MODELS

In accordance with ICLR policy, we disclose that LLMs were used only for grammar and style polishing. All ideas and analyses are by the authors, who take full responsibility for the content.

## A    RECONSTRUCTION PROCESS FOR INFERENCE-TIME SCALING

### A.1    FORECASTING

This section supplements the inference-time scaling for non-autoregressive models. Given historical observations $\mathbf{X} = \{x_1, \ldots, x_l, \ldots, x_L\} \in \mathbb{R}^{L \times C}$ and candidate outputs $\mathbf{Y}_k$ generated by $f_{\text{forward}}$, we employ $f_{\text{backward}}$ to reconstruct reverse-temporal prediction of past data (e.g., $\mathbf{X}$).

**Implementation details.** In non-autoregressive models, $f_{\text{forward}}$ directly predicts all $S$ time steps in a single pass, rather than iteratively as in autoregressive models. Let $F$ denote a fixed window length for applying inference-time scaling. We divide each candidate output $\mathbf{Y}_k$ into $M = \lceil S/F \rceil$ segments. Within the $m$th segment, $\mathbf{Y}_k^{(m)}$ is reversed and reconstructed by $f_{\text{backward}}$:

$$\mathbf{Y}_k^{(m),\text{rev}} = \text{Reverse}(\mathbf{Y}_k^{(m)}), \ \hat{\mathbf{X}}_k^{(m),\text{rev}} = f_{\text{backward}}(\mathbf{Y}_k^{(m),\text{rev}}), \ \hat{\mathbf{X}}_k^{(m)} = \text{Reverse}(\hat{\mathbf{X}}_k^{(m),\text{rev}}). \quad (13)$$

When handling the final segment, an adjustment is required if the final segment is shorter than $F$ (i.e., $R = S \bmod F$). Unlike autoregressive models, non-autoregressive models cannot predict beyond their trained horizon and then trim the output. To construct a valid segment of length $F$, we append the last $(F - R)$ time steps from the $(M-1)$th segment $\mathbf{Y}_k^{(M-1)}$ to the $R$ time steps of the final segment $\mathbf{Y}_k^{(M)}$:

$$\mathbf{Y}_k^{(M)} = \text{Concat}(\mathbf{Y}_{k,R+1:F}^{(M-1)}, \ \mathbf{Y}_{k,1:R}^{(M)}), \quad (14)$$

The extended segment is then reconstructed following Eq. (13).

### A.2    RECONSTRUCTION FOR IMPUTATION

Due to space constraints, we provide the algorithm for inference-time scaling for imputation in this section, as shown in Algorithm 2.

---

**Algorithm 2** Inference-time scaling (ITS) for imputation

---

**Input**: $\mathbf{X} \in \mathbb{R}^{L \times C}$, $N$, $q$, $\mathbf{Y}_{\text{final}}^{(0)} = \emptyset$, $f_{\text{forward}}$, $f_{\text{backward}}$.
**Output**: Reconstructed sequence $\mathbf{Y}_{\text{final}} \in \mathbb{R}^{L \times C}$.
1:  Enable dropout for MC sampling.
2:  Divide $\mathbf{X}$ into $P = L/N$ non-overlapping subsequences, and some of them are randomly masked.
3:  Generate $K$ candidate outputs via MC Dropout.
4:  $\mathbf{X}_{\text{mask}} = \text{Mask}(\mathbf{X}, q)$                          ▷ Mask random subsequences from past or future observable data.
5:  **for** $k = 1$ **to** $K$ **do**
6:     $\mathbf{Y}_k = f_{\text{forward}}(\mathbf{X})$
7:     $\mathbf{X}_k^{\text{obs}} = \text{Insert}(\mathbf{X}_{\text{mask}}, \mathbf{Y}_k)$.
8:     $\hat{\mathbf{X}}_k = f_{\text{backward}}(\mathbf{X}_k^{\text{obs}})$
9:     Compute $e_k = \text{MSE}(\mathbf{X}, \hat{\mathbf{X}}_k)$ on the masked subsequences.
10: **end for**
11: Compute weights and generate $\mathbf{Y}_{\text{final}} = \sum_{k=1}^{K} w_k \cdot \mathbf{Y}_k$.
12: **return** $\mathbf{Y}_{\text{final}}$

---

### A.3    COMPARISON WITH INFERENCE-FOCUSED APPROACHES

We conceptually compare our proposed ITS with several representative inference-focused approaches:

- **Test-time adaptation (TTA)** performs dynamic optimization at inference time, where model parameters are slightly updated to gradually align the model with the test-time distribution. For example, Woo et al. (2023) mitigate long-horizon distribution shift by updating time-index representations online, while Le Naour et al. (2024) optimize latent codes in an implicit neural representation framework to better capture local dynamics during test time.

- **Test-time training (TTT)** similarly introduces self-supervised auxiliary objectives during inference and jointly optimizes the model without requiring future labels, enabling better adaptation to the current input. For instance, Christou et al. (2024) employ test-time training modules to adjust weights during inference, thereby improving adaptivity and dependency modeling.

- **In-context learning (ICL)** leverages a model's pretrained contextual learning capabilities by concatenating additional examples, instructions, or task descriptions to the input, enabling rapid adaptation without any gradient updates. A representative example is Hoo et al. (2025), who reformulate time-series forecasting as a tabular regression task and use TabPFN's attention mechanisms to condition on historical observations as context for future prediction.

- **Inference-time scaling (ITS)** differs from the above paradigms in that it neither updates model parameters at test time nor relies on constructing additional contextual sequences. With the model kept fully frozen, ITS allocates extra compute at inference time to generate a diverse set of candidate predictions and then evaluates their temporal consistency with historical observations to reweight these candidates. In this way, it provides a plug-and-play mechanism that can systematically improve the performance of a wide range of time-series models.

## B  THEORETICAL ANALYSIS

This subsection presents a complete derivation showing that the complete proposed inference-scaling strategy can be explained via Bayesian deep learning theory (BDL) which is effective for epistemic uncertainty modeling. In BDL, the inference for a sample $\boldsymbol{x}$ is defined as:

$$p(y|\boldsymbol{x}, D) = \int_{\boldsymbol{W}} p(y|\boldsymbol{x}, \boldsymbol{W})p(\boldsymbol{W}|D)d\boldsymbol{W}, \tag{15}$$

where $D$ is the training corpus, and $p(\boldsymbol{W}|D)$ is distribution of the model $\boldsymbol{W}$. This inference can eliminate the epistemic uncertainty and yield more robust prediction. In practice, the integral in Eq. (15) is approximated as follows. First, $K$ models are sampled from the posterior distribution $p(\boldsymbol{W}|D)$, and the output is computed for each. The average of these outputs is then taken as the final result, namely:

$$p(y|\boldsymbol{x}, D) \approx \frac{1}{K} \sum_k p(y|\boldsymbol{x}, \boldsymbol{W}_k). \tag{16}$$

However, exact inference of the posterior distribution $p(\boldsymbol{W}|D)$ is generally intractable. To address this issue, Gal and Ghahramani Gal & Ghahramani (2016) proposed a Bayesian approximation method based on MC Dropout. Given a trained model $\boldsymbol{W}_0$, they sampled models with the following manner:

$$\boldsymbol{W}_k = \text{Dropout}(\boldsymbol{W}_0, \mathbf{z}_k), \quad \mathbf{z}_k \sim \text{Bernoulli}(p_d), k = 1, \cdots, K, \tag{17}$$

where $\mathbf{z}_k$ represents dropout variables sampled from the Bernoulli distribution parameterized by $p_d$. Eq. (17) actually utilizes $p(\mathbf{z})$ to approximate $p(\boldsymbol{W}|D)$, i.e., $p(\boldsymbol{W}|D) \sim p(\mathbf{z})$. Eq. (15) then becomes:

$$p(y|\boldsymbol{x}, D) \approx \sum_{\mathbf{z}} p(y|\boldsymbol{x}, \boldsymbol{W}_0, \mathbf{z})p(\mathbf{z}). \tag{18}$$

Even though the above approximation achieves great success in various applications, $p(\mathbf{z})$ provides only a structural approximation to $p(\boldsymbol{W}|D)$, derived solely from the neural network architecture. In practice, additional factors, such as model performance, can contribute to a more refined approximation. Motivated by this rationale, we define the following approximate formulation:

$$p(\boldsymbol{W}|D) \sim p(\mathbf{z})f[E(\boldsymbol{W})], \tag{19}$$

where $E(\boldsymbol{W})$ denotes model error (e.g., classification error). For example, $f[E(\boldsymbol{W})]$ can be defined as:

$$f[E(\boldsymbol{W})] \sim \begin{cases} \frac{1}{\sigma} \exp\left[-\frac{E(\boldsymbol{W})-E^*}{\sigma}\right], & E(\boldsymbol{W}) \leq E^* \\ 0, & otherwise \end{cases}, \tag{20}$$

where $E^*$ is the lowest model error trained on $D$. This definition implies that models exhibiting performance closer to $E^*$ attain higher posterior probability mass. This formulation is theoretically

grounded in the observation that models sufficiently trained on $D$ exhibit bounded performance deviation from the achievable optimum. Accordingly, Eq. (18) becomes:

$$p(y|\boldsymbol{x}, D) \approx \sum_{\mathbf{z}} p(y|\boldsymbol{x}, \boldsymbol{W}_0, \mathbf{z})p(\mathbf{z})p(E(\boldsymbol{W}_z)), \tag{21}$$

where $\boldsymbol{W}_z$ is the sampled model with the dropout variables $\mathbf{z}$. Accordingly, Eq. (16) becomes:

$$
\begin{aligned}
p(y|\boldsymbol{x}, D) &\approx \frac{1}{\sum_j f[E(\boldsymbol{W}_j)]} \sum_k p(y|\boldsymbol{x}, \boldsymbol{W}_k)f[E(\boldsymbol{W}_k)] \\
&= \sum_k \frac{f[E(\boldsymbol{W}_k)]}{\sum_j f[E(\boldsymbol{W}_j)]}p(y|\boldsymbol{x}, \boldsymbol{W}_k) = \sum_k w_k p(y|\boldsymbol{x}, \boldsymbol{W}_k)
\end{aligned}, \tag{22}
$$

where $E^*$ is mathematically removed. There are three typical cases for Eq. (22):

- If $\sigma \to +\infty$, then $f[E(\boldsymbol{W}_k)]$ is a constant. Thus, $w_k \equiv \frac{1}{K}$ and Eq. (22) is reduced to Eq. (16).
- If $\sigma \to 0$, then only the best model among the $K$ models has a weight of 1 and the weights of the rest models are 0 (i.e., Winner takes all). From Eq. (22), the weight $w_k$ can be written as:

$$w_k = \frac{\exp\left[-E(\boldsymbol{W}_k)/\sigma\right]}{\sum_j \exp\left[-E(\boldsymbol{W}_j)/\sigma\right]} = \frac{1}{1 + \sum_{j \neq k} \exp\left[(E(\boldsymbol{W}_k) - E(\boldsymbol{W}_j))/\sigma\right]}. \tag{23}$$

Let $k^* = \arg\min_k E(\boldsymbol{W}_k)$, if $k = k^*$, then:

$$\exp\left[(E(\boldsymbol{W}_k) - E(\boldsymbol{W}_j))/\sigma\right] \to 0, \tag{24}$$

leading to $w_k \to 1$. And if $k \neq k^*$, $\exists\, j = k^*$ for which:

$$\exp\left[(E(\boldsymbol{W}_k) - E(\boldsymbol{W}_j))/\sigma\right] \to \infty, \tag{25}$$

leading to $w_k \to 0$. Therefore, when $\hat{\boldsymbol{W}}^*$ is the best among the sampled models, it follows that $p(y|\boldsymbol{x}, D) = p(y|\boldsymbol{x}, \hat{\boldsymbol{W}}^*)$.

- Otherwise, with mathematical simplification, Eq. (22) becomes:

$$p(y|\boldsymbol{x}, D) = \sum_k \frac{\exp\left[-E(\boldsymbol{W}_k)/\sigma\right]}{\sum_j \exp\left(-E[\boldsymbol{W}_j)/\sigma\right)}p(y|\boldsymbol{x}, \boldsymbol{W}_k). \tag{26}$$

The first case corresponds to the most common manner, namely, Majority Voting (or Averaging) when confronting with multiple outputs, which is also employed in Liu et al. (2025b). The second and the third cases require the assessment of model performance, which is still infeasible for real applications. We can denote the reconstruction error for the current input sample as an assessment of the involved model, namely, $E(\boldsymbol{W}_k) \sim e_k$. Eq. (22) then becomes:

$$p(y|\boldsymbol{x}, D) = \sum_k \frac{\exp(-e_k/\sigma)}{\sum_j \exp(-e_j/\sigma)}p(y|\boldsymbol{x}, \boldsymbol{W}_k), \tag{27}$$

which equals to Eq. (1) in the main submission of this study. That is to say, the weighted scheme proposed earlier essentially serves as a ensemble method to mitigate model uncertainty and thus outputs more robust results.

## C  IMPLEMENTATION DETAILS

### C.1  DATASETS DETAILS

We conduct experiments on seven real-world datasets to evaluate the performance of the proposed framework, including: (1) ETT (Zhou et al., 2021): The Electricity Transformer Temperature dataset records 7 variables from July 2016 to July 2018. It consists of four subsets: ETTh1 and ETTh2, which are recorded hourly, and ETTm1 and ETTm2, which are recorded every 15 minutes. (2) Exchange (Wu et al., 2021): This dataset contains daily exchange rates of 8 major countries over the period from 1990 to 2016. (3) Weather (Zhou et al., 2021): This dataset includes 21 meteorological variables collected every 10 minutes in 2020 from the Weather Station of the Max Planck Biogeochemistry Institute. (4) Electricity (Li et al., 2019): This dataset contains hourly electricity consumption data from 321 clients. Detailed descriptions of these datasets are provided in Table 4.

Table 4: Detailed dataset descriptions. Dim denotes the number of variables in each dataset. Dataset Size denotes the total number of time points in the (Train, Validation, Test) splits, respectively. Series Length denotes the number of future time points to be predicted, with four prediction settings included in each dataset. Frequency denotes the sampling interval between time points.

| Dataset | Dim | Series Length | Dataset Size | Frequency | Information |
|---|---|---|---|---|---|
| ETTh1 | 7 | {96, 192, 336, 720} | (8545, 2881, 2881) | Hourly | Electricity |
| ETTh2 | 7 | {96, 192, 336, 720} | (8545, 2881, 2881) | Hourly | Electricity |
| ETTm1 | 7 | {96, 192, 336, 720} | (34465, 11521, 11521) | 15min | Electricity |
| ETTm2 | 7 | {96, 192, 336, 720} | (34465, 11521, 11521) | 15min | Electricity |
| Exchange | 8 | {96, 192, 336, 720} | (5120, 665, 1422) | Daily | Exchange rate |
| Weather | 21 | {96, 192, 336, 720} | (36792, 5271, 10540) | 10min | Weather |
| Electricity | 321 | {96, 192, 336, 720} | (18317, 2633, 5261) | Hourly | Electricity |

## C.2 METRIC DETAILS

All experiments are conducted using a fixed random seed to ensure reproducibility. We adopt the mean square error (MSE) and mean absolute error (MAE) for both forecasting and imputation tasks, while MSE reduction is used to quantify the decrease in MSE achieved by our framework. Specifically, the metrics are computed as follows:

$$\text{MSE} = \frac{1}{S} \sum_{i=1}^{S} \left( \mathbf{Y}_i - \hat{\mathbf{Y}}_i \right)^2, \qquad \text{MAE} = \frac{1}{S} \sum_{i=1}^{S} \left| \mathbf{Y}_i - \hat{\mathbf{Y}}_i \right|. \tag{28}$$

$$\text{MSE Reduction} = \frac{\text{MSE}_{\text{vanilla}} - \text{MSE}_{\text{ITS}}}{\text{MSE}_{\text{vanilla}}} \times 100\%, \tag{29}$$

where $\mathbf{Y}, \hat{\mathbf{Y}} \in \mathbb{R}^{S \times C}$ are the ground-truth and predicted results of the future, with $S$ time points and $C$ variables.

# D FULL RESULTS

## D.1 FULL RESULTS OF FORECASTING AND IMPUTATION

The full results are provided in this section due to space limitations in the main text. We conducted a comprehensive evaluation of the proposed ITS on challenging forecasting and imputation tasks across a wide range of methods. Table 5 reports detailed MSE results for all prediction lengths on seven datasets for forecasting tasks, while Table 6 provides the corresponding MAE results. Table 7 presents detailed MSE results for all mask rates on seven datasets for imputation tasks, with comprehensive MAE results shown in Table 8. The proposed ITS consistently reduces MSE across diverse datasets and model architectures, demonstrating its superior effectiveness.

Table 5: Full MSE results for forecasting with prediction lengths $S \in \{96, 192, 336, 720\}$ are reported on seven datasets, using $K = 64$ and a fixed lookback length of $L = 96$ (Timer uses $L = 672$). Vanilla denotes the original methods, while **+ITS** represents our proposed framework.

| Method | | iTransformer | | TimeMixer | | TimeXer | | PatchTST | | Crossformer | | TimesNet | | MICN | | Autoformer | | Timer | |
|---|---|---|---|---|---|---|---|---|---|---|---|---|---|---|---|---|---|---|---|
| Dataset | | Vanilla | +ITS | Vanilla | +ITS | Vanilla | +ITS | Vanilla | +ITS | Vanilla | +ITS | Vanilla | +ITS | Vanilla | +ITS | Vanilla | +ITS | Vanilla | +ITS |
| ETTh1 | 96 | 0.411 | **0.407** | **0.373** | 0.374 | **0.390** | 0.393 | **0.408** | 0.409 | 0.395 | **0.391** | 0.395 | **0.394** | **0.382** | 0.386 | 0.445 | **0.452** | 0.370 | **0.355** |
| | 192 | 0.462 | **0.459** | 0.446 | **0.441** | **0.435** | 0.436 | 0.460 | **0.459** | 0.628 | **0.614** | 0.449 | **0.446** | 0.437 | **0.436** | 0.536 | **0.524** | 0.406 | **0.388** |
| | 336 | 0.492 | **0.487** | 0.516 | **0.508** | 0.486 | **0.473** | 0.494 | **0.492** | 0.665 | **0.657** | 0.499 | **0.495** | **0.445** | 0.448 | 0.519 | **0.508** | 0.422 | **0.402** |
| | 720 | 0.512 | **0.503** | 0.501 | **0.493** | 0.518 | **0.497** | 0.495 | **0.489** | 0.887 | **0.825** | 0.512 | **0.503** | 0.464 | **0.462** | 0.517 | **0.526** | 0.435 | **0.414** |
| ETTh2 | 96 | 0.311 | **0.310** | 0.299 | **0.297** | 0.287 | **0.285** | 0.296 | 0.296 | 0.690 | **0.652** | 0.329 | **0.325** | 0.372 | **0.369** | 0.349 | **0.347** | 0.299 | **0.289** |
| | 192 | 0.402 | **0.401** | 0.381 | **0.380** | 0.365 | **0.363** | 0.388 | **0.387** | 0.852 | 0.857 | 0.395 | **0.392** | 0.400 | **0.399** | 0.426 | **0.421** | 0.370 | **0.358** |
| | 336 | 0.441 | **0.440** | 0.433 | 0.433 | 0.415 | **0.410** | 0.419 | **0.418** | 1.070 | 1.060 | 0.450 | **0.447** | 0.427 | **0.426** | 0.452 | **0.447** | 0.393 | **0.378** |
| | 720 | 0.444 | **0.443** | 0.438 | **0.436** | 0.424 | **0.418** | 0.425 | 0.425 | 1.268 | 1.284 | 0.459 | **0.455** | 0.461 | **0.458** | 0.488 | **0.477** | 0.450 | **0.430** |
| ETTm1 | 96 | 0.353 | **0.349** | 0.320 | **0.309** | 0.318 | **0.311** | 0.341 | **0.329** | 0.360 | **0.352** | 0.348 | **0.333** | 0.314 | **0.310** | 0.557 | **0.508** | 0.309 | **0.283** |
| | 192 | 0.397 | **0.395** | 0.372 | **0.362** | 0.364 | **0.352** | 0.373 | **0.362** | 0.433 | **0.426** | 0.391 | **0.378** | 0.368 | **0.363** | 0.507 | **0.487** | 0.363 | **0.330** |
| | 336 | 0.435 | **0.433** | 0.395 | **0.381** | 0.398 | **0.385** | 0.398 | **0.387** | 0.459 | **0.454** | 0.414 | **0.399** | 0.400 | **0.396** | 0.570 | **0.531** | 0.403 | **0.365** |
| | 720 | 0.498 | **0.496** | 0.458 | **0.443** | 0.457 | **0.445** | 0.458 | **0.446** | 0.563 | **0.559** | 0.480 | **0.466** | 0.471 | **0.463** | 0.513 | **0.499** | 0.478 | **0.427** |
| ETTm2 | 96 | 0.186 | **0.182** | 0.176 | **0.172** | **0.170** | 0.171 | 0.177 | **0.166** | 0.268 | **0.253** | 0.186 | **0.185** | 0.187 | **0.184** | 0.215 | **0.213** | 0.185 | **0.175** |
| | 192 | 0.253 | **0.249** | 0.237 | **0.225** | 0.239 | **0.238** | 0.243 | **0.233** | 0.412 | **0.371** | 0.301 | **0.297** | 0.257 | **0.256** | 0.273 | **0.272** | 0.257 | **0.239** |
| | 336 | 0.316 | **0.310** | 0.298 | **0.293** | 0.300 | **0.291** | 0.305 | **0.293** | 0.578 | 0.581 | 0.317 | **0.315** | 0.312 | **0.310** | 0.332 | **0.328** | 0.315 | **0.292** |
| | 720 | 0.412 | **0.404** | 0.399 | **0.396** | 0.400 | **0.385** | 0.410 | **0.395** | 1.570 | **1.556** | 0.463 | **0.459** | 0.392 | 0.392 | 0.422 | 0.422 | 0.403 | **0.375** |
| Exchange | 96 | 0.087 | **0.081** | 0.084 | 0.084 | **0.086** | 0.088 | **0.089** | 0.090 | 0.378 | **0.366** | 0.109 | **0.107** | 0.085 | **0.081** | 0.147 | **0.142** | 0.095 | **0.085** |
| | 192 | 0.180 | **0.172** | 0.203 | **0.201** | **0.187** | 0.184 | 0.196 | **0.193** | 0.609 | **0.596** | 0.212 | **0.205** | 0.163 | **0.161** | 0.372 | **0.245** | 0.197 | **0.188** |
| | 336 | 0.335 | **0.325** | 0.384 | **0.367** | **0.335** | 0.326 | 0.342 | **0.338** | 1.081 | **1.051** | 0.382 | **0.378** | 0.302 | **0.300** | 0.453 | 0.456 | 0.360 | **0.341** |
| | 720 | 0.862 | **0.859** | 1.102 | **1.034** | **0.862** | 0.849 | 0.882 | **0.863** | 1.669 | 1.679 | 0.968 | **0.965** | 0.686 | **0.684** | 1.154 | **1.122** | 0.841 | **0.835** |
| Weather | 96 | 0.196 | **0.195** | 0.164 | **0.160** | **0.158** | 0.160 | 0.177 | 0.177 | 0.149 | **0.147** | 0.173 | **0.172** | 0.176 | **0.172** | 0.332 | **0.217** | 0.155 | **0.151** |
| | 192 | 0.240 | **0.238** | 0.209 | **0.204** | **0.204** | 0.205 | 0.224 | **0.222** | 0.208 | **0.206** | 0.275 | **0.273** | 0.240 | **0.232** | 0.319 | **0.288** | 0.204 | **0.196** |
| | 336 | 0.294 | **0.289** | 0.263 | **0.258** | 0.262 | **0.261** | 0.278 | **0.268** | 0.277 | 0.277 | 0.280 | 0.280 | 0.290 | **0.282** | 0.355 | **0.345** | 0.262 | **0.248** |
| | 720 | 0.367 | **0.363** | 0.345 | **0.339** | 0.341 | 0.341 | 0.352 | **0.342** | 0.367 | **0.366** | 0.381 | **0.380** | 0.325 | **0.320** | 0.427 | **0.403** | 0.340 | **0.319** |
| Electricity | 96 | 0.150 | **0.145** | 0.153 | **0.152** | 0.139 | **0.133** | **0.193** | 0.196 | 0.219 | **0.218** | 0.172 | **0.171** | 0.160 | **0.149** | 0.213 | **0.210** | 0.135 | **0.132** |
| | 192 | 0.167 | **0.158** | 0.167 | 0.167 | 0.157 | **0.152** | 0.198 | **0.193** | 0.237 | 0.238 | 0.194 | 0.194 | 0.179 | **0.168** | 0.316 | **0.308** | 0.155 | **0.151** |
| | 336 | 0.181 | **0.172** | 0.188 | **0.187** | 0.177 | **0.161** | 0.214 | **0.207** | 0.785 | **0.778** | 0.204 | 0.206 | 0.196 | **0.185** | 0.256 | **0.248** | 0.171 | **0.167** |
| | 720 | 0.222 | **0.210** | 0.226 | **0.224** | 0.218 | **0.211** | 0.256 | **0.249** | 0.813 | **0.810** | 0.244 | 0.249 | 0.207 | **0.196** | 0.263 | **0.261** | 0.206 | **0.205** |

Table 6: Full MAE results for forecasting with prediction lengths $S \in \{96, 192, 336, 720\}$ are reported on seven datasets, using $K = 64$ and a fixed lookback length of $L = 96$ (Timer uses $L = 672$). Vanilla denotes the original methods, while **+ITS** represents our proposed framework.

| Method | | iTransformer | | TimeMixer | | TimeXer | | PatchTST | | Crossformer | | TimesNet | | MICN | | Autoformer | | Timer | |
|---|---|---|---|---|---|---|---|---|---|---|---|---|---|---|---|---|---|---|---|
| Dataset | | Vanilla | +ITS | Vanilla | +ITS | Vanilla | +ITS | Vanilla | +ITS | Vanilla | +ITS | Vanilla | +ITS | Vanilla | +ITS | Vanilla | +ITS | Vanilla | +ITS |
| ETTh1 | 96 | 0.421 | **0.417** | 0.398 | 0.398 | 0.404 | **0.403** | 0.415 | **0.412** | 0.422 | **0.417** | 0.416 | **0.415** | 0.416 | **0.415** | **0.439** | 0.448 | 0.396 | **0.390** |
| | 192 | 0.450 | **0.447** | 0.431 | **0.430** | 0.435 | **0.433** | 0.445 | **0.443** | 0.538 | **0.530** | 0.449 | **0.447** | 0.459 | **0.457** | 0.499 | **0.492** | 0.420 | **0.413** |
| | 336 | 0.461 | **0.459** | 0.473 | **0.471** | 0.455 | **0.453** | 0.462 | **0.460** | 0.567 | **0.563** | 0.472 | **0.470** | 0.455 | 0.457 | 0.501 | **0.494** | 0.431 | **0.423** |
| | 720 | 0.495 | **0.491** | 0.477 | **0.472** | 0.500 | **0.491** | 0.484 | **0.482** | 0.742 | **0.712** | 0.492 | **0.488** | 0.484 | 0.487 | **0.518** | 0.519 | 0.449 | **0.442** |
| ETTh2 | 96 | 0.359 | **0.358** | 0.349 | **0.348** | 0.339 | **0.337** | 0.345 | **0.344** | 0.602 | **0.574** | 0.368 | **0.365** | 0.418 | **0.417** | 0.395 | **0.390** | 0.350 | **0.345** |
| | 192 | 0.413 | **0.412** | 0.400 | **0.399** | 0.390 | **0.388** | 0.400 | **0.399** | 0.688 | 0.691 | 0.408 | **0.405** | 0.414 | **0.412** | 0.439 | **0.433** | 0.397 | **0.390** |
| | 336 | 0.446 | **0.444** | 0.442 | **0.441** | 0.426 | **0.423** | 0.430 | **0.429** | 0.750 | 0.753 | 0.450 | **0.447** | 0.444 | **0.442** | 0.464 | **0.459** | 0.421 | **0.412** |
| | 720 | 0.457 | **0.456** | 0.448 | **0.447** | 0.443 | **0.439** | 0.443 | **0.442** | 0.829 | 0.838 | 0.463 | **0.460** | 0.475 | **0.473** | 0.497 | **0.488** | 0.469 | **0.456** |
| ETTm1 | 96 | 0.380 | **0.375** | 0.358 | **0.355** | 0.357 | **0.354** | 0.374 | **0.370** | 0.400 | **0.393** | 0.379 | **0.376** | 0.363 | **0.359** | 0.496 | **0.468** | 0.350 | **0.337** |
| | 192 | 0.401 | **0.398** | 0.388 | **0.387** | 0.384 | **0.381** | 0.391 | **0.388** | 0.452 | **0.446** | 0.406 | **0.404** | 0.402 | **0.398** | 0.482 | **0.469** | 0.383 | **0.367** |
| | 336 | 0.425 | **0.423** | 0.406 | **0.404** | 0.408 | **0.404** | 0.408 | **0.406** | 0.455 | **0.451** | 0.423 | **0.419** | 0.426 | **0.422** | 0.512 | **0.491** | 0.411 | **0.392** |
| | 720 | 0.460 | **0.458** | **0.444** | 0.445 | 0.442 | **0.440** | 0.442 | **0.440** | 0.523 | **0.520** | 0.456 | **0.455** | 0.468 | **0.464** | 0.502 | **0.489** | 0.455 | **0.427** |
| ETTm2 | 96 | 0.272 | **0.270** | 0.258 | **0.257** | 0.256 | **0.255** | 0.261 | **0.260** | 0.353 | **0.338** | 0.263 | **0.262** | 0.284 | **0.280** | 0.298 | **0.295** | 0.260 | **0.252** |
| | 192 | 0.313 | **0.312** | 0.340 | **0.339** | 0.301 | **0.300** | 0.304 | **0.303** | 0.479 | **0.444** | 0.354 | **0.351** | 0.317 | **0.315** | 0.331 | **0.330** | 0.307 | **0.295** |
| | 336 | 0.352 | **0.351** | 0.339 | 0.339 | 0.340 | **0.337** | 0.346 | **0.344** | 0.531 | **0.530** | 0.347 | **0.346** | 0.352 | **0.349** | 0.369 | **0.365** | 0.346 | **0.332** |
| | 720 | 0.405 | **0.404** | 0.402 | **0.401** | 0.398 | **0.397** | 0.407 | **0.403** | 0.909 | **0.903** | 0.442 | **0.439** | 0.404 | **0.403** | 0.418 | 0.419 | 0.403 | **0.388** |
| Exchange | 96 | 0.207 | 0.207 | 0.202 | 0.202 | **0.205** | 0.206 | **0.207** | 0.209 | 0.450 | 0.453 | 0.239 | **0.238** | 0.209 | 0.209 | 0.275 | **0.272** | **0.218** | 0.220 |
| | 192 | 0.304 | **0.303** | 0.318 | **0.317** | 0.307 | **0.306** | 0.313 | **0.311** | 0.590 | **0.586** | 0.332 | **0.330** | 0.308 | 0.308 | 0.445 | **0.363** | **0.319** | 0.321 |
| | 336 | 0.421 | **0.419** | 0.449 | **0.438** | 0.417 | **0.411** | 0.423 | **0.420** | 0.808 | **0.804** | 0.452 | **0.450** | 0.426 | 0.426 | **0.500** | 0.503 | **0.435** | 0.438 |
| | 720 | 0.702 | 0.702 | 0.784 | **0.762** | 0.694 | **0.688** | 0.707 | **0.700** | 1.042 | 1.044 | **0.752** | 0.753 | 0.652 | 0.652 | 0.834 | **0.821** | **0.688** | 0.694 |
| Weather | 96 | 0.235 | **0.234** | 0.211 | **0.207** | **0.207** | 0.209 | 0.217 | 0.219 | 0.222 | **0.218** | 0.223 | **0.222** | 0.249 | **0.243** | 0.376 | **0.288** | 0.198 | **0.196** |
| | 192 | 0.274 | **0.273** | 0.252 | **0.243** | **0.248** | 0.249 | 0.259 | 0.260 | 0.283 | **0.279** | 0.308 | **0.306** | 0.307 | **0.299** | 0.376 | **0.348** | 0.245 | **0.240** |
| | 336 | 0.310 | **0.309** | 0.291 | 0.291 | 0.291 | 0.291 | **0.298** | 0.299 | 0.341 | 0.341 | 0.304 | 0.304 | 0.342 | **0.335** | 0.389 | **0.378** | 0.290 | **0.281** |
| | 720 | 0.356 | **0.351** | 0.345 | **0.344** | 0.343 | **0.342** | **0.346** | 0.347 | 0.411 | 0.412 | 0.373 | **0.372** | 0.363 | **0.359** | 0.433 | **0.414** | 0.346 | **0.331** |
| Electricity | 96 | 0.243 | **0.242** | 0.245 | **0.243** | 0.240 | **0.236** | 0.284 | **0.277** | 0.318 | **0.317** | 0.278 | **0.277** | 0.266 | **0.265** | 0.326 | **0.323** | 0.225 | **0.224** |
| | 192 | 0.258 | **0.257** | 0.257 | **0.256** | 0.256 | **0.253** | 0.291 | **0.285** | 0.332 | **0.331** | 0.296 | 0.297 | 0.286 | **0.285** | 0.385 | **0.381** | 0.244 | **0.243** |
| | 336 | 0.274 | **0.273** | 0.275 | **0.274** | 0.275 | **0.273** | 0.304 | 0.309 | 0.729 | **0.725** | 0.307 | 0.307 | 0.303 | **0.302** | 0.359 | **0.354** | 0.260 | 0.260 |
| | 720 | 0.308 | 0.308 | 0.315 | **0.313** | 0.309 | **0.306** | 0.337 | 0.341 | 0.741 | **0.739** | 0.336 | 0.340 | 0.313 | 0.313 | 0.366 | **0.364** | 0.291 | 0.294 |

Table 7: Full MSE results for imputation are reported on seven datasets. We randomly mask $\{12.5\%, 25\%, 37.5\%, 50\%\}$ of subsequences in time series with length $L = 96$. The results use $K = 64$ and a fixed subsequence length of $N = 12$ (Timer uses $N = 24$). Vanilla denotes the original methods, while **+ITS** represents our proposed framework.

| Method | | iTransformer | | TimeMixer | | PatchTST | | Crossformer | | TimesNet | | MICN | | Autoformer | | Timer | |
|---|---|---|---|---|---|---|---|---|---|---|---|---|---|---|---|---|---|
| Dataset | | Vanilla | +ITS | Vanilla | +ITS | Vanilla | +ITS | Vanilla | +ITS | Vanilla | +ITS | Vanilla | +ITS | Vanilla | +ITS | Vanilla | +ITS |
| ETTh1 | 12.5% | 0.262 | **0.255** | 0.274 | **0.268** | 0.320 | **0.300** | 0.300 | **0.285** | 0.232 | **0.223** | 0.275 | **0.256** | 0.398 | **0.348** | 0.314 | **0.308** |
| | 25% | 0.303 | **0.290** | 0.296 | **0.291** | 0.370 | **0.342** | 0.335 | **0.315** | 0.268 | **0.258** | 0.325 | **0.299** | 0.494 | **0.432** | 0.318 | **0.315** |
| | 37.5% | 0.337 | **0.321** | 0.316 | **0.312** | 0.401 | **0.368** | 0.362 | **0.339** | 0.304 | **0.293** | 0.397 | **0.361** | 0.498 | **0.435** | 0.339 | **0.334** |
| | 50% | 0.379 | **0.361** | 0.347 | **0.343** | 0.444 | **0.407** | 0.398 | **0.373** | 0.359 | **0.344** | 0.472 | **0.427** | 0.565 | **0.488** | 0.349 | **0.344** |
| ETTh2 | 12.5% | 0.230 | **0.185** | 0.118 | **0.115** | 0.134 | **0.124** | 0.290 | **0.237** | 0.108 | **0.100** | 0.278 | **0.251** | 1.030 | **0.887** | 0.134 | 0.134 |
| | 25% | 0.292 | **0.231** | 0.128 | **0.126** | 0.142 | **0.133** | 0.343 | **0.281** | 0.116 | **0.109** | 0.390 | **0.341** | 1.416 | **1.247** | 0.147 | 0.147 |
| | 37.5% | 0.347 | **0.276** | 0.140 | **0.138** | 0.155 | **0.145** | 0.383 | **0.315** | 0.129 | **0.123** | 0.532 | **0.457** | 1.612 | **1.411** | 0.160 | 0.160 |
| | 50% | 0.384 | **0.313** | 0.155 | **0.154** | 0.169 | **0.160** | 0.429 | **0.350** | 0.146 | **0.139** | 0.683 | **0.579** | 1.878 | **1.672** | 0.176 | 0.176 |
| ETTm1 | 12.5% | 0.159 | **0.154** | 0.142 | 0.142 | 0.154 | **0.150** | 0.146 | **0.134** | 0.143 | **0.138** | 0.116 | **0.109** | 0.339 | **0.258** | 0.373 | **0.350** |
| | 25% | 0.205 | **0.196** | 0.179 | **0.177** | 0.190 | **0.187** | 0.163 | **0.153** | 0.204 | **0.196** | 0.160 | **0.148** | 0.426 | **0.339** | 0.434 | **0.407** |
| | 37.5% | 0.267 | **0.252** | 0.233 | **0.232** | 0.253 | **0.247** | 0.179 | **0.175** | 0.271 | **0.261** | 0.211 | **0.193** | 0.514 | **0.429** | 0.480 | **0.457** |
| | 50% | 0.327 | **0.310** | 0.305 | **0.302** | 0.330 | **0.321** | 0.217 | **0.215** | 0.347 | **0.334** | 0.278 | **0.252** | 0.610 | **0.524** | 0.547 | **0.521** |
| ETTm2 | 12.5% | 0.114 | **0.094** | 0.065 | **0.064** | 0.067 | **0.066** | 0.110 | **0.082** | 0.054 | **0.051** | 0.130 | **0.113** | 0.854 | **0.651** | 0.110 | **0.109** |
| | 25% | 0.138 | **0.112** | 0.072 | **0.071** | 0.074 | **0.073** | 0.127 | **0.093** | 0.061 | **0.059** | 0.188 | **0.157** | 1.021 | **0.875** | 0.121 | **0.120** |
| | 37.5% | 0.160 | **0.131** | 0.081 | **0.080** | 0.083 | **0.082** | 0.146 | **0.109** | 0.072 | **0.068** | 0.249 | **0.204** | 1.209 | **1.018** | 0.131 | **0.130** |
| | 50% | 0.169 | **0.142** | 0.095 | 0.095 | 0.099 | **0.098** | 0.166 | **0.138** | 0.086 | **0.083** | 0.397 | **0.353** | 1.288 | **1.100** | 0.143 | **0.142** |
| Exchange | 12.5% | 0.078 | **0.050** | 0.010 | 0.010 | 0.010 | 0.010 | 0.521 | **0.410** | 0.018 | **0.016** | 0.116 | **0.100** | 1.112 | **0.965** | 0.024 | 0.024 |
| | 25% | 0.095 | **0.062** | 0.011 | 0.011 | 0.012 | 0.012 | 0.540 | **0.429** | 0.019 | **0.018** | 0.157 | **0.134** | 1.404 | **1.234** | 0.026 | 0.026 |
| | 37.5% | 0.105 | **0.071** | 0.013 | 0.013 | 0.014 | 0.014 | 0.576 | **0.462** | 0.022 | **0.020** | 0.190 | **0.158** | 1.755 | **1.559** | 0.030 | **0.029** |
| | 50% | 0.124 | **0.088** | 0.017 | 0.017 | 0.018 | 0.018 | 0.618 | **0.501** | 0.026 | **0.024** | 0.242 | **0.205** | 2.119 | **1.864** | 0.035 | **0.034** |
| Weather | 12.5% | 0.100 | **0.087** | 0.069 | **0.068** | 0.067 | **0.066** | 0.087 | **0.077** | 0.063 | **0.061** | 0.078 | **0.071** | 0.299 | **0.218** | 0.105 | **0.102** |
| | 25% | 0.117 | **0.101** | 0.079 | **0.078** | 0.072 | **0.071** | 0.072 | **0.068** | 0.069 | **0.067** | 0.081 | **0.074** | 0.346 | **0.286** | 0.119 | **0.115** |
| | 37.5% | 0.133 | **0.114** | 0.087 | **0.085** | 0.079 | **0.078** | 0.076 | **0.072** | 0.077 | **0.075** | 0.090 | **0.082** | 0.426 | **0.334** | 0.120 | **0.118** |
| | 50% | 0.093 | **0.089** | 0.097 | **0.095** | 0.093 | **0.092** | 0.081 | **0.076** | 0.089 | **0.087** | 0.098 | **0.089** | 0.410 | **0.342** | 0.133 | **0.130** |
| Electricity | 12.5% | **0.092** | 0.094 | 0.107 | **0.106** | 0.138 | 0.138 | 0.174 | **0.167** | 0.117 | **0.115** | 0.179 | **0.166** | 0.353 | **0.311** | 0.156 | **0.151** |
| | 25% | **0.103** | 0.104 | 0.123 | **0.122** | 0.155 | **0.153** | 0.188 | **0.178** | 0.129 | **0.126** | 0.195 | **0.180** | 0.372 | **0.331** | 0.168 | **0.162** |
| | 37.5% | **0.116** | 0.117 | 0.146 | **0.145** | 0.180 | **0.176** | 0.206 | **0.195** | 0.146 | **0.144** | 0.207 | **0.190** | 0.383 | **0.345** | 0.180 | **0.174** |
| | 50% | **0.132** | 0.133 | 0.177 | **0.176** | 0.213 | **0.206** | 0.226 | **0.214** | 0.171 | **0.168** | 0.218 | **0.201** | 0.393 | **0.357** | 0.193 | **0.188** |

Table 8: Full MAE results for imputation are reported on seven datasets. We randomly mask $\{12.5\%, 25\%, 37.5\%, 50\%\}$ of subsequences in time series with length $L = 96$. The results use $K = 64$ and a fixed subsequence length of $N = 12$ (Timer uses $N = 24$). Vanilla denotes the original methods, while **+ITS** represents our proposed framework.

| Method | | iTransformer | | TimeMixer | | PatchTST | | Crossformer | | TimesNet | | MICN | | Autoformer | | Timer | |
|---|---|---|---|---|---|---|---|---|---|---|---|---|---|---|---|---|---|
| Dataset | | Vanilla | +ITS | Vanilla | +ITS | Vanilla | +ITS | Vanilla | +ITS | Vanilla | +ITS | Vanilla | +ITS | Vanilla | +ITS | Vanilla | +ITS |
| ETTh1 | 12.5% | 0.347 | **0.335** | 0.339 | **0.334** | 0.367 | **0.353** | 0.373 | **0.357** | 0.322 | **0.314** | 0.384 | **0.367** | 0.461 | **0.424** | 0.363 | **0.360** |
| | 25% | 0.372 | **0.357** | 0.349 | **0.346** | 0.391 | **0.374** | 0.395 | **0.376** | 0.341 | **0.333** | 0.421 | **0.399** | 0.514 | **0.472** | 0.368 | **0.367** |
| | 37.5% | 0.392 | **0.376** | 0.361 | **0.359** | 0.406 | **0.388** | 0.413 | **0.393** | 0.360 | **0.352** | 0.472 | **0.445** | 0.529 | **0.485** | 0.379 | **0.377** |
| | 50% | 0.413 | **0.396** | 0.377 | **0.375** | 0.425 | **0.406** | 0.433 | **0.411** | 0.388 | **0.378** | 0.515 | **0.485** | 0.563 | **0.514** | 0.385 | **0.383** |
| ETTh2 | 12.5% | 0.332 | **0.294** | 0.225 | **0.222** | 0.244 | **0.234** | 0.359 | **0.319** | 0.223 | **0.215** | 0.342 | **0.317** | 0.636 | **0.565** | **0.237** | 0.238 |
| | 25% | 0.374 | **0.329** | 0.234 | **0.232** | 0.251 | **0.241** | 0.397 | **0.355** | 0.231 | **0.224** | 0.424 | **0.387** | 0.764 | **0.695** | **0.244** | 0.245 |
| | 37.5% | 0.405 | **0.357** | 0.244 | **0.242** | 0.261 | **0.251** | 0.422 | **0.378** | 0.241 | **0.235** | 0.510 | **0.462** | 0.834 | **0.762** | 0.253 | 0.253 |
| | 50% | 0.428 | **0.382** | 0.257 | **0.255** | 0.271 | **0.262** | 0.453 | **0.402** | 0.254 | **0.248** | 0.599 | **0.539** | 0.931 | **0.862** | 0.264 | 0.264 |
| ETTm1 | 12.5% | 0.262 | **0.254** | **0.237** | 0.238 | 0.245 | **0.242** | 0.256 | **0.241** | 0.237 | **0.233** | 0.223 | **0.215** | 0.396 | **0.333** | 0.378 | **0.369** |
| | 25% | 0.297 | **0.285** | 0.261 | **0.260** | 0.270 | **0.267** | 0.276 | **0.273** | 0.279 | **0.273** | 0.270 | **0.256** | 0.446 | **0.385** | 0.403 | **0.392** |
| | 37.5% | 0.338 | **0.320** | 0.293 | **0.291** | 0.306 | **0.302** | 0.296 | **0.279** | 0.317 | **0.311** | 0.315 | **0.296** | 0.496 | **0.440** | 0.423 | **0.414** |
| | 50% | 0.369 | **0.351** | 0.328 | **0.326** | 0.345 | **0.340** | 0.327 | **0.309** | 0.355 | **0.348** | 0.366 | **0.342** | 0.549 | **0.497** | 0.450 | **0.439** |
| ETTm2 | 12.5% | 0.224 | **0.200** | 0.153 | **0.153** | 0.159 | **0.157** | 0.229 | **0.192** | 0.155 | **0.149** | 0.238 | **0.215** | 0.535 | **0.431** | 0.207 | **0.206** |
| | 25% | 0.251 | **0.222** | 0.163 | **0.163** | 0.169 | **0.167** | 0.248 | **0.204** | 0.165 | **0.160** | 0.292 | **0.259** | 0.615 | **0.551** | 0.217 | 0.217 |
| | 37.5% | 0.272 | **0.242** | 0.175 | **0.174** | 0.181 | **0.179** | 0.264 | **0.221** | 0.178 | **0.173** | 0.346 | **0.303** | 0.706 | **0.620** | 0.226 | 0.226 |
| | 50% | 0.281 | **0.252** | 0.190 | **0.189** | 0.198 | **0.197** | 0.277 | **0.251** | 0.192 | **0.188** | 0.430 | **0.392** | 0.770 | **0.686** | 0.238 | 0.238 |
| Exchange | 12.5% | 0.190 | **0.147** | 0.061 | 0.061 | **0.063** | 0.065 | 0.503 | **0.441** | 0.089 | **0.084** | 0.233 | **0.207** | 0.671 | **0.602** | **0.095** | 0.096 |
| | 25% | 0.214 | **0.167** | 0.064 | **0.065** | **0.070** | 0.071 | 0.518 | **0.455** | 0.092 | **0.088** | 0.282 | **0.249** | 0.758 | **0.694** | 0.101 | 0.101 |
| | 37.5% | 0.229 | **0.181** | 0.072 | 0.072 | **0.076** | 0.077 | 0.539 | **0.475** | 0.098 | **0.094** | 0.318 | **0.278** | 0.885 | **0.816** | 0.110 | **0.109** |
| | 50% | 0.249 | **0.200** | 0.081 | 0.081 | **0.085** | 0.086 | 0.563 | **0.499** | 0.106 | **0.103** | 0.364 | **0.321** | 1.003 | **0.914** | 0.119 | **0.118** |
| Weather | 12.5% | 0.154 | **0.134** | 0.095 | **0.093** | 0.093 | **0.092** | 0.159 | **0.132** | 0.105 | **0.100** | 0.144 | **0.124** | 0.339 | **0.263** | 0.136 | **0.134** |
| | 25% | 0.187 | **0.161** | 0.110 | **0.108** | 0.098 | **0.097** | 0.131 | **0.110** | 0.112 | **0.108** | 0.151 | **0.131** | 0.376 | **0.322** | 0.154 | **0.153** |
| | 37.5% | 0.205 | **0.178** | 0.119 | **0.116** | 0.105 | **0.104** | 0.140 | **0.117** | 0.122 | **0.118** | 0.164 | **0.143** | 0.441 | **0.371** | 0.150 | **0.149** |
| | 50% | 0.136 | **0.127** | 0.128 | **0.126** | 0.124 | **0.121** | 0.147 | **0.126** | 0.133 | **0.130** | 0.172 | **0.151** | 0.430 | **0.374** | 0.163 | **0.161** |
| Electricity | 12.5% | **0.195** | 0.198 | 0.205 | **0.204** | 0.249 | **0.244** | 0.293 | **0.284** | 0.233 | **0.231** | 0.292 | **0.280** | 0.433 | **0.401** | 0.248 | **0.246** |
| | 25% | **0.206** | 0.209 | 0.220 | **0.219** | 0.261 | **0.255** | 0.304 | **0.293** | 0.247 | **0.244** | 0.304 | **0.291** | 0.443 | **0.413** | 0.256 | **0.254** |
| | 37.5% | **0.219** | 0.221 | 0.238 | **0.237** | 0.280 | **0.272** | 0.319 | **0.308** | 0.263 | **0.260** | 0.313 | **0.300** | 0.449 | **0.421** | 0.264 | **0.262** |
| | 50% | **0.234** | 0.235 | 0.261 | **0.260** | 0.303 | **0.293** | 0.335 | **0.324** | 0.285 | **0.282** | 0.321 | **0.307** | 0.454 | **0.427** | 0.272 | **0.271** |

## D.2 FULL RESULTS FOR IMPUTATION WITH VARIOUS SUBSEQUENCE LENGTHS

In this section, we conduct a comprehensive analysis of the imputation performance of ITS under varying subsequence lengths. Timer is excluded from this comparison, as it is fine-tuned from a released checkpoint, and modifying the subsequence length $N$ would conflict with its architecture. Building on the default experimental settings, we evaluate ITS with three different lengths $N \in \{3, 6, 24\}$. Table 9 presents the MSE results for $N = 3$, Table 10 reports the results for $N = 6$, and Table 11 shows the results for $N = 24$.

The experimental findings indicate that ITS achieves MSE reductions in 93.37%, 93.88%, and 83.69% of the test cases for $N = 3$, $N = 6$, and $N = 24$, respectively, compared to the vanilla methods. These results highlight that ITS consistently maintains strong imputation performance across different missing-length configurations and demonstrates adaptability to a variety of missing-data scenarios.

Table 9: Full MSE results for imputation are reported on seven datasets. We randomly mask $\{12.5\%, 25\%, 37.5\%, 50\%\}$ of subsequences in time series with length $L = 96$. The results use $K = 64$ and a fixed subsequence length of $N = 3$. Vanilla denotes the original methods, while **+ITS** represents our proposed framework.

| Method | | iTransformer | | TimeMixer | | PatchTST | | Crossformer | | TimesNet | | MICN | | Autoformer | |
|---|---|---|---|---|---|---|---|---|---|---|---|---|---|---|---|
| Dataset | | Vanilla | +ITS | Vanilla | +ITS | Vanilla | +ITS | Vanilla | +ITS | Vanilla | +ITS | Vanilla | +ITS | Vanilla | +ITS |
| ETTh1 | 12.5% | 0.201 | **0.187** | 0.167 | **0.163** | 0.236 | **0.215** | 0.338 | **0.310** | 0.193 | **0.183** | 0.132 | **0.130** | 0.130 | **0.120** |
| | 25% | 0.232 | **0.215** | 0.190 | **0.187** | 0.261 | **0.239** | 0.226 | **0.205** | 0.205 | **0.195** | 0.180 | **0.170** | 0.175 | **0.159** |
| | 37.5% | 0.271 | **0.252** | 0.219 | **0.216** | 0.297 | **0.273** | 0.250 | **0.228** | 0.223 | **0.214** | 0.241 | **0.223** | 0.240 | **0.212** |
| | 50% | 0.326 | **0.302** | 0.249 | **0.246** | 0.338 | **0.310** | 0.312 | **0.288** | 0.256 | **0.246** | 0.335 | **0.306** | 0.330 | **0.290** |
| ETTh2 | 12.5% | 0.190 | **0.150** | 0.078 | **0.077** | 0.097 | **0.089** | 0.208 | **0.170** | 0.073 | **0.069** | 0.168 | **0.157** | 0.302 | **0.226** |
| | 25% | 0.239 | **0.186** | 0.086 | **0.085** | 0.099 | **0.093** | 0.225 | **0.184** | 0.077 | **0.074** | 0.245 | **0.223** | 0.391 | **0.308** |
| | 37.5% | 0.287 | **0.223** | 0.094 | **0.093** | 0.105 | **0.100** | 0.251 | **0.207** | 0.085 | **0.082** | 0.347 | **0.309** | 0.517 | **0.418** |
| | 50% | 0.329 | **0.264** | 0.106 | **0.105** | 0.114 | **0.110** | 0.296 | **0.244** | 0.097 | **0.093** | 0.477 | **0.416** | 0.696 | **0.588** |
| ETTm1 | 12.5% | 0.090 | **0.083** | 0.062 | **0.061** | 0.069 | **0.064** | 0.081 | **0.070** | 0.054 | **0.053** | 0.052 | **0.049** | 0.092 | **0.073** |
| | 25% | 0.118 | **0.107** | 0.070 | **0.069** | 0.079 | **0.074** | 0.093 | **0.083** | 0.074 | **0.071** | 0.076 | **0.070** | 0.129 | **0.105** |
| | 37.5% | 0.151 | **0.135** | 0.078 | 0.078 | 0.091 | **0.088** | 0.107 | **0.095** | 0.094 | **0.090** | 0.106 | **0.096** | 0.175 | **0.144** |
| | 50% | 0.200 | **0.177** | 0.099 | 0.099 | 0.117 | **0.114** | 0.127 | **0.114** | 0.118 | **0.113** | 0.154 | **0.137** | 0.295 | **0.245** |
| ETTm2 | 12.5% | 0.090 | **0.071** | 0.084 | **0.083** | 0.037 | 0.037 | 0.097 | **0.068** | 0.032 | **0.031** | 0.085 | **0.076** | 0.144 | **0.098** |
| | 25% | 0.118 | **0.091** | 0.083 | **0.082** | 0.043 | **0.042** | 0.104 | **0.077** | 0.037 | **0.035** | 0.133 | **0.113** | 0.220 | **0.191** |
| | 37.5% | 0.141 | **0.107** | 0.044 | **0.043** | 0.049 | **0.048** | 0.109 | **0.083** | 0.042 | **0.040** | 0.184 | **0.151** | 0.338 | **0.311** |
| | 50% | 0.169 | **0.129** | 0.048 | **0.047** | 0.053 | **0.052** | 0.121 | **0.089** | 0.048 | **0.047** | 0.255 | **0.208** | 0.411 | **0.337** |
| Exchange | 12.5% | 0.056 | **0.036** | 0.004 | 0.004 | 0.006 | **0.005** | 0.450 | **0.355** | 0.008 | **0.007** | 0.094 | **0.087** | **0.341** | 0.349 |
| | 25% | 0.075 | **0.047** | 0.004 | 0.004 | 0.006 | 0.006 | 0.432 | **0.340** | 0.008 | **0.007** | 0.127 | **0.113** | 0.501 | **0.465** |
| | 37.5% | 0.078 | **0.045** | 0.005 | 0.005 | 0.007 | **0.006** | 0.443 | **0.351** | 0.009 | **0.008** | 0.204 | **0.181** | 0.698 | **0.636** |
| | 50% | 0.076 | **0.052** | 0.006 | 0.006 | 0.008 | **0.007** | 0.467 | **0.373** | 0.011 | **0.010** | 0.361 | **0.317** | 0.954 | **0.843** |
| Weather | 12.5% | 0.225 | **0.181** | 0.082 | **0.081** | 0.085 | **0.081** | 0.223 | **0.192** | 0.062 | **0.061** | 0.313 | **0.289** | 0.287 | **0.247** |
| | 25% | 0.237 | **0.205** | 0.085 | **0.083** | 0.086 | **0.083** | 0.204 | **0.177** | 0.068 | **0.067** | 0.267 | **0.246** | 0.354 | **0.296** |
| | 37.5% | 0.250 | **0.216** | 0.087 | **0.086** | 0.092 | **0.090** | 0.182 | **0.158** | 0.076 | **0.075** | 0.270 | **0.246** | 0.423 | **0.342** |
| | 50% | 0.271 | **0.235** | 0.103 | **0.102** | 0.102 | **0.100** | 0.167 | **0.145** | 0.088 | **0.086** | 0.288 | **0.259** | 0.473 | **0.394** |
| Electricity | 12.5% | 0.075 | 0.075 | 0.071 | **0.070** | 0.103 | **0.098** | 0.139 | **0.127** | 0.007 | **0.006** | 0.110 | **0.105** | 0.262 | **0.228** |
| | 25% | 0.086 | 0.086 | 0.082 | **0.081** | 0.122 | **0.115** | 0.156 | **0.144** | 0.008 | **0.007** | 0.131 | **0.124** | 0.301 | **0.262** |
| | 37.5% | 0.095 | 0.095 | 0.096 | **0.095** | 0.142 | **0.132** | 0.176 | **0.162** | 0.009 | **0.008** | 0.150 | **0.141** | 0.327 | **0.289** |
| | 50% | **0.108** | 0.109 | 0.117 | **0.115** | 0.168 | **0.157** | 0.199 | **0.185** | 0.011 | **0.010** | 0.168 | **0.157** | 0.374 | **0.338** |

Table 10: Full MSE results for imputation are reported on seven datasets. We randomly mask $\{12.5\%, 25\%, 37.5\%, 50\%\}$ of subsequences in time series with length $L = 96$. The results use $K = 64$ and a fixed subsequence length of $N = 6$. Vanilla denotes the original methods, while **+ITS** represents our proposed framework.

| Method | | iTransformer | | TimeMixer | | PatchTST | | Crossformer | | TimesNet | | MICN | | Autoformer | |
|---|---|---|---|---|---|---|---|---|---|---|---|---|---|---|---|
| Dataset | | Vanilla | +ITS | Vanilla | +ITS | Vanilla | +ITS | Vanilla | +ITS | Vanilla | +ITS | Vanilla | +ITS | Vanilla | +ITS |
| ETTh1 | 12.5% | 0.242 | **0.228** | 0.222 | **0.217** | 0.281 | **0.259** | 0.271 | **0.249** | 0.220 | **0.209** | 0.179 | **0.172** | 0.205 | **0.183** |
| | 25% | 0.272 | **0.256** | 0.248 | **0.243** | 0.315 | **0.290** | 0.306 | **0.281** | 0.242 | **0.231** | 0.251 | **0.234** | 0.285 | **0.251** |
| | 37.5% | 0.309 | **0.292** | 0.276 | **0.272** | 0.354 | **0.327** | 0.335 | **0.310** | 0.275 | **0.264** | 0.322 | **0.297** | 0.412 | **0.350** |
| | 50% | 0.354 | **0.333** | 0.303 | **0.299** | 0.395 | **0.365** | 0.357 | **0.331** | 0.318 | **0.305** | 0.418 | **0.381** | 0.436 | **0.383** |
| ETTh2 | 12.5% | 0.440 | **0.337** | 0.133 | **0.129** | 0.141 | **0.130** | 0.661 | **0.541** | 0.103 | **0.097** | 0.757 | **0.708** | 1.213 | **1.052** |
| | 25% | 0.447 | **0.349** | 0.138 | **0.135** | 0.148 | **0.138** | 0.623 | **0.511** | 0.112 | **0.107** | 0.891 | **0.817** | 1.516 | **1.356** |
| | 37.5% | 0.505 | **0.397** | 0.150 | **0.147** | 0.158 | **0.149** | 0.696 | **0.580** | 0.125 | **0.120** | 1.120 | **1.013** | 1.703 | **1.494** |
| | 50% | 0.607 | **0.484** | 0.164 | **0.162** | 0.172 | **0.164** | 0.766 | **0.648** | 0.142 | **0.137** | 1.286 | **1.153** | 1.924 | **1.696** |
| ETTm1 | 12.5% | 0.117 | **0.110** | 0.087 | **0.086** | 0.092 | **0.090** | 0.102 | **0.092** | 0.082 | **0.079** | 0.074 | **0.070** | 0.167 | **0.130** |
| | 25% | 0.149 | **0.138** | **0.100** | 0.100 | 0.110 | **0.109** | 0.121 | **0.110** | 0.113 | **0.109** | 0.107 | **0.098** | 0.236 | **0.190** |
| | 37.5% | 0.192 | **0.176** | 0.128 | 0.129 | 0.140 | **0.139** | 0.142 | **0.131** | 0.150 | **0.145** | 0.152 | **0.137** | 0.315 | **0.259** |
| | 50% | 0.239 | **0.220** | **0.166** | 0.168 | 0.181 | 0.181 | 0.154 | **0.149** | 0.199 | **0.192** | 0.213 | **0.189** | 0.409 | **0.345** |
| ETTm2 | 12.5% | 0.110 | **0.087** | 0.048 | **0.047** | 0.049 | **0.048** | 0.103 | **0.075** | 0.041 | **0.039** | 0.100 | **0.086** | 0.343 | **0.217** |
| | 25% | 0.146 | **0.113** | 0.051 | **0.050** | **0.053** | 0.054 | 0.110 | **0.085** | 0.046 | **0.045** | 0.155 | **0.127** | 0.476 | **0.416** |
| | 37.5% | 0.176 | **0.136** | 0.057 | 0.057 | 0.062 | **0.061** | 0.119 | **0.090** | 0.053 | **0.052** | 0.227 | **0.183** | 0.692 | **0.559** |
| | 50% | 0.209 | **0.162** | 0.067 | **0.066** | 0.070 | 0.070 | 0.139 | **0.108** | 0.063 | **0.061** | 0.300 | **0.240** | 0.919 | **0.787** |
| Exchange | 12.5% | 0.162 | **0.112** | 0.011 | 0.011 | 0.013 | 0.013 | 0.601 | **0.482** | 0.016 | **0.015** | 0.354 | **0.328** | 1.266 | **1.049** |
| | 25% | 0.164 | **0.114** | 0.012 | 0.012 | 0.015 | **0.014** | 0.615 | **0.497** | 0.018 | **0.017** | 0.456 | **0.411** | 1.539 | **1.293** |
| | 37.5% | 0.179 | **0.128** | 0.014 | 0.014 | 0.016 | **0.015** | 0.653 | **0.535** | 0.020 | **0.019** | 0.534 | **0.477** | 1.811 | **1.532** |
| | 50% | 0.195 | **0.146** | 0.017 | 0.017 | 0.020 | **0.019** | 0.698 | **0.579** | 0.024 | **0.023** | 0.641 | **0.573** | 2.091 | **1.759** |
| Weather | 12.5% | 0.153 | **0.126** | 0.069 | **0.068** | 0.077 | **0.074** | 0.141 | **0.119** | 0.062 | **0.061** | 0.149 | **0.137** | 0.277 | **0.228** |
| | 25% | 0.162 | **0.136** | 0.072 | **0.070** | 0.082 | **0.080** | 0.099 | **0.086** | 0.068 | **0.067** | 0.152 | **0.139** | 0.337 | **0.285** |
| | 37.5% | 0.182 | **0.153** | 0.090 | **0.089** | 0.087 | **0.086** | 0.099 | **0.087** | 0.077 | **0.075** | 0.138 | **0.123** | 0.410 | **0.343** |
| | 50% | 0.156 | **0.140** | 0.102 | **0.101** | 0.101 | **0.100** | 0.102 | **0.091** | 0.088 | **0.087** | 0.149 | **0.133** | 0.457 | **0.344** |
| Electricity | 12.5% | 0.128 | **0.125** | 0.122 | **0.121** | 0.178 | **0.169** | 0.217 | **0.200** | 0.120 | **0.117** | 0.215 | **0.202** | 0.335 | **0.289** |
| | 25% | 0.132 | **0.129** | 0.133 | **0.132** | 0.183 | **0.174** | 0.222 | **0.205** | 0.132 | **0.130** | 0.215 | **0.200** | 0.349 | **0.305** |
| | 37.5% | 0.143 | **0.139** | 0.158 | **0.156** | 0.208 | **0.196** | 0.236 | **0.217** | 0.151 | **0.149** | 0.221 | **0.205** | 0.373 | **0.331** |
| | 50% | 0.161 | **0.157** | 0.189 | **0.187** | 0.244 | **0.231** | 0.253 | **0.236** | 0.179 | **0.176** | 0.234 | **0.217** | 0.392 | **0.352** |

Table 11: Full MSE results for imputation are reported on seven datasets. We randomly mask $\{12.5\%, 25\%, 37.5\%, 50\%\}$ of subsequences in time series with length $L = 96$. The results use $K = 64$ and a fixed subsequence length of $N = 24$. Vanilla denotes the original methods, while **+ITS** represents our proposed framework.

| Method | | iTransformer | | TimeMixer | | PatchTST | | Crossformer | | TimesNet | | MICN | | Autoformer | |
|---|---|---|---|---|---|---|---|---|---|---|---|---|---|---|---|
| Dataset | | Vanilla | +ITS | Vanilla | +ITS | Vanilla | +ITS | Vanilla | +ITS | Vanilla | +ITS | Vanilla | +ITS | Vanilla | +ITS |
| ETTh1 | 12.5% | 0.274 | **0.270** | 0.296 | **0.292** | 0.342 | **0.325** | 0.331 | **0.313** | 0.241 | **0.234** | 0.324 | **0.295** | 0.673 | **0.571** |
| | 25% | 0.297 | **0.289** | 0.307 | **0.303** | 0.357 | **0.336** | 0.351 | **0.327** | 0.260 | **0.252** | 0.466 | **0.418** | 0.616 | **0.523** |
| | 37.5% | 0.323 | **0.313** | 0.320 | **0.316** | 0.381 | **0.356** | 0.373 | **0.346** | 0.295 | **0.285** | 0.463 | **0.417** | 0.768 | **0.687** |
| | 50% | 0.346 | **0.334** | 0.332 | **0.329** | 0.392 | **0.367** | 0.391 | **0.362** | 0.342 | **0.324** | 0.490 | **0.443** | 0.756 | **0.648** |
| ETTh2 | 12.5% | 0.267 | **0.224** | 0.150 | **0.148** | 0.171 | **0.162** | 0.360 | **0.293** | 0.125 | **0.120** | 0.424 | **0.388** | 2.152 | **1.811** |
| | 25% | 0.307 | **0.257** | 0.168 | **0.166** | 0.179 | **0.174** | 0.405 | **0.329** | 0.143 | **0.137** | 0.523 | **0.472** | 2.364 | **2.030** |
| | 37.5% | 0.327 | **0.275** | 0.184 | **0.182** | 0.196 | **0.190** | 0.439 | **0.357** | 0.159 | **0.154** | 0.599 | **0.535** | 2.463 | **2.148** |
| | 50% | 0.332 | **0.287** | 0.201 | **0.200** | 0.215 | **0.208** | 0.482 | **0.391** | 0.182 | **0.177** | 0.647 | **0.572** | 2.572 | **2.279** |
| ETTm1 | 12.5% | **0.262** | 0.275 | 0.272 | **0.271** | **0.296** | 0.364 | **0.189** | 0.256 | 0.257 | **0.255** | 0.208 | **0.202** | 0.604 | **0.519** |
| | 25% | **0.316** | 0.324 | 0.331 | 0.331 | **0.345** | 0.395 | **0.210** | 0.259 | 0.354 | **0.348** | 0.269 | **0.258** | 0.666 | **0.525** |
| | 37.5% | **0.368** | 0.375 | **0.392** | 0.393 | **0.393** | 0.436 | **0.236** | 0.277 | 0.438 | **0.430** | 0.328 | **0.310** | 0.730 | **0.567** |
| | 50% | **0.423** | 0.431 | 0.475 | **0.463** | **0.471** | 0.510 | **0.284** | 0.322 | 0.513 | **0.504** | 0.382 | **0.361** | 0.841 | **0.653** |
| ETTm2 | 12.5% | 0.124 | **0.113** | 0.099 | **0.098** | **0.098** | 0.103 | 0.148 | **0.115** | 0.080 | **0.079** | 0.209 | **0.186** | 1.886 | **1.300** |
| | 25% | 0.135 | **0.123** | 0.110 | **0.109** | **0.111** | 0.114 | 0.163 | **0.131** | 0.090 | **0.089** | 0.252 | **0.219** | 1.815 | **1.500** |
| | 37.5% | 0.147 | **0.135** | 0.121 | **0.120** | **0.124** | 0.126 | 0.191 | **0.149** | 0.103 | **0.102** | 0.298 | **0.259** | 1.868 | **1.604** |
| | 50% | 0.164 | **0.152** | 0.135 | **0.133** | **0.139** | 0.140 | 0.213 | **0.172** | 0.119 | **0.117** | 0.385 | **0.345** | 2.064 | **1.770** |
| Exchange | 12.5% | 0.095 | **0.069** | 0.019 | 0.019 | 0.022 | **0.021** | 0.567 | **0.449** | 0.029 | **0.026** | 0.147 | **0.133** | 2.175 | **1.748** |
| | 25% | 0.088 | **0.062** | 0.023 | **0.022** | 0.023 | 0.023 | 0.580 | **0.460** | 0.033 | **0.030** | 0.153 | **0.134** | 2.374 | **1.846** |
| | 37.5% | 0.075 | **0.054** | 0.026 | **0.025** | 0.028 | **0.027** | 0.600 | **0.478** | 0.035 | **0.033** | 0.190 | **0.169** | 2.747 | **2.118** |
| | 50% | 0.070 | **0.052** | 0.032 | **0.031** | 0.033 | **0.032** | 0.612 | **0.489** | 0.040 | **0.038** | 0.235 | **0.212** | 2.670 | **2.274** |
| Weather | 12.5% | 0.122 | **0.112** | 0.103 | **0.102** | **0.102** | 0.103 | 0.089 | **0.085** | 0.088 | **0.087** | 0.095 | **0.089** | 0.499 | **0.312** |
| | 25% | 0.137 | **0.125** | 0.112 | **0.110** | **0.105** | 0.107 | 0.089 | **0.084** | 0.097 | **0.096** | 0.102 | **0.095** | 0.476 | **0.340** |
| | 37.5% | 0.162 | **0.146** | 0.126 | **0.124** | **0.116** | 0.117 | 0.094 | **0.089** | 0.107 | **0.106** | 0.116 | **0.107** | 0.481 | **0.350** |
| | 50% | 0.174 | **0.160** | 0.135 | **0.134** | 0.129 | **0.128** | 0.100 | **0.095** | 0.121 | **0.120** | 0.125 | **0.115** | 0.499 | **0.398** |
| Electricity | 12.5% | **0.101** | 0.103 | 0.128 | 0.129 | 0.156 | 0.161 | 0.183 | **0.181** | 0.114 | **0.112** | 0.215 | **0.198** | 0.379 | **0.340** |
| | 25% | **0.108** | 0.110 | 0.140 | 0.140 | **0.164** | 0.165 | 0.191 | **0.185** | 0.120 | **0.118** | 0.219 | **0.202** | 0.381 | **0.341** |
| | 37.5% | **0.117** | 0.118 | 0.155 | 0.155 | 0.179 | **0.175** | 0.202 | **0.191** | 0.127 | **0.125** | 0.223 | **0.206** | 0.393 | **0.352** |
| | 50% | **0.127** | 0.128 | 0.171 | **0.170** | 0.195 | **0.191** | 0.216 | **0.203** | 0.135 | **0.133** | 0.232 | **0.215** | 0.381 | **0.345** |

## D.3 Full results for forecasting with longer look-back windows

In principle, a longer look-back window expands the receptive field of the model and may therefore lead to improved forecasting performance. To verify whether ITS remains effective under longer historical dependencies, we extend the look-back window on representative models and datasets. The complete results are reported in Table 12.

Table 12: Full MSE results for forecasting with prediction lengths $S \in \{96, 192, 336, 720\}$ are reported on representative models and datasets, using $K = 64$ and a fixed lookback length of $L = 720$. Vanilla denotes the original methods, while **+ITS** represents our proposed framework.

| Method | | iTransformer | | PatchTST | | Crossformer | |
|---|---|---|---|---|---|---|---|
| Dataset | | Vanilla | +ITS | Vanilla | +ITS | Vanilla | +ITS |
| ETTh1 | 96 | 0.402 | **0.385** | 0.384 | **0.383** | 0.463 | **0.433** |
| | 192 | 0.435 | **0.419** | 0.426 | **0.412** | 0.631 | **0.607** |
| | 336 | 0.458 | **0.440** | 0.467 | **0.453** | 0.646 | **0.620** |
| | 720 | 0.602 | **0.575** | 0.512 | **0.497** | 1.402 | **1.131** |
| ETTh2 | 96 | 0.316 | **0.307** | 0.287 | 0.287 | 0.840 | **0.783** |
| | 192 | 0.379 | **0.370** | 0.365 | **0.353** | 1.247 | **1.095** |
| | 336 | 0.402 | **0.389** | 0.385 | **0.373** | 1.387 | **1.112** |
| | 720 | 0.439 | **0.426** | 0.424 | **0.415** | 1.738 | **1.575** |
| Weather | 96 | 0.165 | **0.164** | 0.146 | **0.137** | 0.148 | **0.141** |
| | 192 | **0.208** | 0.209 | 0.189 | **0.180** | 0.212 | **0.204** |
| | 336 | 0.264 | **0.260** | 0.244 | **0.231** | 0.268 | **0.259** |
| | 720 | 0.327 | **0.323** | 0.316 | **0.302** | 0.326 | **0.312** |
| Electricity | 96 | 0.134 | **0.125** | 0.141 | **0.132** | 0.194 | 0.194 |
| | 192 | 0.161 | **0.156** | 0.157 | **0.148** | 0.227 | **0.213** |
| | 336 | 0.175 | **0.169** | 0.173 | **0.154** | 0.775 | **0.650** |
| | 720 | 0.192 | **0.184** | 0.207 | **0.188** | 0.803 | **0.621** |

## D.4 Full results for imputation with longer sequences and specific baselines

In this section, we present the complete imputation results under longer input sequence settings. Increasing the sequence length enables a more thorough evaluation of ITS in scenarios with stronger temporal dependencies. In addition, we evaluate several dedicated imputation baselines including SAITS (Du et al., 2023), CSDI (Tashiro et al., 2021), and BRITS (Cao et al., 2018), thereby providing a more comprehensive and rigorous comparison. The full results are reported in Table 13, and these results demonstrate the robustness and effectiveness of ITS across varying sequence lengths and model architectures.

Table 13: Full MSE results for imputation are reported on representative models and datasets. We randomly mask $\{12.5\%, 25\%, 37.5\%, 50\%\}$ of subsequences in time series with length $L = 720$. The results use $K = 64$ and a fixed subsequence length of $N = 12$. Vanilla denotes the original methods, while **+ITS** represents our proposed framework.

| Method | | iTransformer | | PatchTST | | Crossformer | | SAITS | | CSDI | | BRITS | |
|---|---|---|---|---|---|---|---|---|---|---|---|---|---|
| Dataset | | Vanilla | +ITS | Vanilla | +ITS | Vanilla | +ITS | Vanilla | +ITS | Vanilla | +ITS | Vanilla | +ITS |
| ETTh1 | 12.5% | 0.347 | **0.328** | 0.294 | **0.277** | 0.319 | **0.304** | 0.167 | **0.149** | 0.352 | **0.327** | 0.259 | **0.254** |
| | 25% | 0.363 | **0.343** | 0.304 | **0.288** | 0.327 | **0.313** | 0.199 | **0.182** | 0.360 | **0.341** | 0.387 | **0.372** |
| | 37.5% | 0.392 | **0.368** | 0.318 | **0.302** | 0.339 | **0.325** | 0.250 | **0.237** | 0.447 | **0.424** | 0.374 | **0.366** |
| | 50% | 0.438 | **0.406** | 0.333 | **0.319** | 0.355 | **0.343** | 0.295 | **0.278** | 0.375 | **0.354** | 0.458 | **0.439** |
| ETTh2 | 12.5% | 0.305 | **0.254** | 0.119 | **0.104** | 0.265 | **0.229** | 0.362 | **0.322** | 0.414 | **0.391** | 0.402 | **0.400** |
| | 25% | 0.432 | **0.346** | 0.128 | **0.113** | 0.318 | **0.273** | 0.432 | **0.413** | 0.399 | **0.382** | 0.401 | **0.395** |
| | 37.5% | 0.549 | **0.511** | 0.142 | **0.133** | 0.385 | **0.328** | 0.554 | **0.521** | 0.397 | **0.382** | 0.436 | **0.427** |
| | 50% | 0.638 | **0.552** | 0.156 | **0.146** | 0.500 | **0.420** | 0.635 | **0.574** | 0.455 | **0.443** | 0.458 | **0.449** |
| Weather | 12.5% | 0.077 | **0.076** | 0.068 | **0.067** | 0.078 | **0.071** | 0.243 | **0.229** | 0.272 | **0.267** | **0.311** | 0.314 |
| | 25% | 0.087 | **0.085** | 0.074 | **0.072** | 0.076 | **0.073** | 0.274 | **0.263** | 0.261 | **0.250** | **0.335** | 0.334 |
| | 37.5% | 0.093 | **0.092** | 0.078 | **0.075** | 0.081 | **0.077** | 0.331 | **0.320** | 0.306 | **0.301** | 0.362 | 0.362 |
| | 50% | 0.103 | **0.102** | 0.087 | **0.086** | 0.089 | **0.084** | 0.358 | **0.344** | 0.327 | **0.322** | 0.407 | **0.405** |
| Electricity | 12.5% | **0.082** | 0.084 | 0.086 | **0.084** | 0.120 | **0.104** | 0.380 | **0.364** | 0.294 | **0.285** | 0.381 | 0.381 |
| | 25% | **0.091** | 0.093 | 0.095 | **0.091** | 0.133 | **0.113** | 0.496 | **0.477** | 0.312 | **0.303** | 0.403 | **0.400** |
| | 37.5% | 0.099 | 0.099 | 0.106 | **0.105** | 0.151 | **0.140** | 0.559 | **0.523** | 0.335 | **0.324** | 0.442 | **0.441** |
| | 50% | 0.107 | **0.105** | 0.118 | **0.114** | 0.157 | **0.151** | 0.725 | **0.675** | 0.406 | **0.396** | 0.506 | **0.502** |

# E  BACKWARD MODEL

## E.1  TRAINING DETAILS

In this section, we provide more details on the pretraining and fine-tuning process of the backward model. We adopt the Timer architecture (Liu et al., 2024c) as the backbone of our backward model. During pretraining, the model is trained on the temporally reversed 12G-UTSD dataset (Liu et al., 2024c) in an autoregressive manner. The token length is set to 96, and the dropout rate is fixed at 0.1. We use the AdamW optimizer (Kingma & Ba, 2014) and apply a cosine annealing scheduler to adjust the learning rate. The pretraining process is conducted on 2 NVIDIA A100 80G GPUs. Although this pretraining process is relatively costly and takes roughly one month to complete, we will release the pretrained weights publicly, which will eliminate this burden for future users. Table 14 presents the detailed hyperparameter settings used during training.

Table 14: Pretraining hyperparameter settings for the backward model on the UTSD dataset.

| Dataset | Layers | Head | Model Dim |
|---|---|---|---|
| 12G-UTSD | 8 | 8 | 1024 |
| **FFN Dim** | **Batch** | **Epoch** | **Lr** |
| 2048 | 8192 | 3 | $5 \times 10^{-5}$ |

For both forecasting and imputation tasks, the pretrained backward model is further fine-tuned on each target dataset. Specifically, for forecasting, the model is fine-tuned on temporally reversed datasets, using a lookback length of $L = 96$ and a prediction length of $S = 96$. For imputation, the model is fine-tuned on the original datasets using sequences of length $L = 96$, where random subsequences are masked at missing rates of $\{12.5\%, 25\%, 37.5\%, 50\%\}$ with a fixed subsequence length of $N = 12$. During fine-tuning, we use a dropout rate of 0.1, apply a cosine annealing scheduler for learning rate adjustment, and adopt early stopping. The fine-tuning process is performed on a single NVIDIA A100 80G GPU and incurs relatively low computational cost. The detailed hyperparameter settings and training cost are summarized in Table 15.

Table 15: Fine-tuning hyperparameters and training cost of the backward model for forecasting and imputation.

| Task | Dataset | Layers | Head | Model Dim | FFN Dim | Batch | Epoch | Lr | Time |
|---|---|---|---|---|---|---|---|---|---|
| Forecasting | ETTh1 | 8 | 8 | 1024 | 2048 | 1024 | 30 | $5 \times 10^{-6}$ | 358.24s |
| | ETTh2 | 8 | 8 | 1024 | 2048 | 1024 | 30 | $5 \times 10^{-6}$ | 364.15s |
| | ETTm1 | 8 | 8 | 1024 | 2048 | 1024 | 30 | $5 \times 10^{-6}$ | 28.24min |
| | ETTm2 | 8 | 8 | 1024 | 2048 | 1024 | 30 | $5 \times 10^{-6}$ | 27.48min |
| | Electricity | 8 | 8 | 1024 | 2048 | 32 | 30 | $5 \times 10^{-6}$ | 2.14hour |
| | Exchange | 8 | 8 | 1024 | 2048 | 1024 | 30 | $5 \times 10^{-6}$ | 294.58s |
| | Weather | 8 | 8 | 1024 | 2048 | 512 | 30 | $5 \times 10^{-6}$ | 43.62min |
| Imputation | ETTh1 | 8 | 8 | 1024 | 2048 | 1024 | 30 | $1 \times 10^{-4}$ | 122.47s |
| | ETTh2 | 8 | 8 | 1024 | 2048 | 1024 | 30 | $1 \times 10^{-4}$ | 124.12s |
| | ETTm1 | 8 | 8 | 1024 | 2048 | 1024 | 30 | $1 \times 10^{-4}$ | 172.39s |
| | ETTm2 | 8 | 8 | 1024 | 2048 | 1024 | 30 | $1 \times 10^{-4}$ | 325.28s |
| | Electricity | 8 | 8 | 1024 | 2048 | 512 | 30 | $1 \times 10^{-4}$ | 39.21min |
| | Exchange | 8 | 8 | 1024 | 2048 | 1024 | 30 | $1 \times 10^{-4}$ | 72.69s |
| | Weather | 8 | 8 | 1024 | 2048 | 1024 | 30 | $1 \times 10^{-4}$ | 406.31s |

## E.2  RECONSTRUCTION PERFORMANCE ANALYSIS

We compare the reconstruction performance of our Fine-Tuned Backward Model (FT-BM) with that of the SOTA methods used for evaluation in our framework. Specifically, for forecasting, we train each SOTA method on each target dataset after temporal reversal. For imputation, we use the models trained on the original target datasets. The detailed results are presented in Table 16 and Table 17. Finally, we evaluate the effectiveness of using the time-series foundation model (TSFM),

pretrained on large-scale datasets, as the backward model component in our ITS framework. The results, compared against four representative SOTA method used as the backward model component, denoted as TSLM in the figures, are presented in Fig. 8–15.

Table 16: Comparison of reconstruction performance between the fine-tuned backward model (FT-BM) and the SOTA methods on forecasting tasks, with lookback length $L = 96$ and prediction length $S = 96$.

| Method | FT-BM | | iTransformer | | TimeMixer | | TimeXer | | PatchTST | | Crossformer | | TimesNet | | MICN | | Autoformer | |
|---|---|---|---|---|---|---|---|---|---|---|---|---|---|---|---|---|---|---|
| Dataset | MSE | MAE | MSE | MAE | MSE | MAE | MSE | MAE | MSE | MAE | MSE | MAE | MSE | MAE | MSE | MAE | MSE | MAE |
| ETTh1 | 0.192 | 0.308 | 0.239 | 0.312 | 0.373 | 0.396 | **0.176** | **0.219** | 0.238 | 0.305 | 0.178 | 0.240 | 0.290 | 0.356 | 0.382 | 0.415 | 0.445 | 0.452 |
| ETTh2 | **0.075** | **0.193** | 0.172 | 0.265 | 0.313 | 0.356 | 0.125 | 0.193 | 0.146 | 0.231 | 0.168 | 0.274 | 0.173 | 0.263 | 0.372 | 0.418 | 0.349 | 0.395 |
| ETTm1 | **0.040** | **0.137** | 0.184 | 0.254 | 0.322 | 0.361 | 0.144 | 0.185 | 0.152 | 0.209 | 0.159 | 0.222 | 0.186 | 0.257 | 0.085 | 0.212 | 0.370 | 0.408 |
| ETTm2 | **0.026** | **0.103** | 0.086 | 0.174 | 0.176 | 0.260 | 0.066 | 0.130 | 0.069 | 0.133 | 0.101 | 0.197 | 0.129 | 0.240 | 0.060 | 0.170 | 0.148 | 0.256 |
| Exchange | 0.041 | 0.150 | 0.025 | 0.099 | 0.086 | 0.204 | 0.022 | 0.082 | 0.023 | 0.083 | 0.235 | 0.341 | 0.031 | 0.115 | **0.003** | **0.033** | 0.060 | 0.173 |
| Weather | **0.015** | **0.047** | 0.069 | 0.097 | 0.161 | 0.209 | 0.065 | 0.095 | 0.074 | 0.103 | 0.059 | 0.100 | 0.099 | 0.146 | 0.037 | 0.121 | 0.156 | 0.238 |
| Electricity | **0.005** | **0.046** | 0.069 | 0.127 | 0.154 | 0.249 | 0.067 | 0.151 | 0.103 | 0.185 | 0.135 | 0.240 | 0.129 | 0.240 | 0.047 | 0.147 | 0.171 | 0.291 |

Table 17: Comparison of reconstruction performance between the fine-tuned backward model (FT-BM) and the SOTA model on the imputation task. We randomly mask $\{12.5\%, 25\%, 37.5\%, 50\%\}$ of subsequences within time series of length $L = 96$, with the subsequence length set to $N = 12$.

| Method | | FT-BM | | iTransformer | | TimeMixer | | PatchTST | | Crossformer | | TimesNet | | MICN | | Autoformer | |
|---|---|---|---|---|---|---|---|---|---|---|---|---|---|---|---|---|---|---|
| Dataset | | MSE | MAE | MSE | MAE | MSE | MAE | MSE | MAE | MSE | MAE | MSE | MAE | MSE | MAE | MSE | MAE |
| ETTh1 | 12.5% | **0.201** | **0.297** | 0.263 | 0.347 | 0.274 | 0.339 | 0.320 | 0.367 | 0.300 | 0.373 | 0.232 | 0.322 | 0.275 | 0.384 | 0.398 | 0.461 |
| | 25% | **0.228** | **0.317** | 0.303 | 0.372 | 0.296 | 0.349 | 0.370 | 0.391 | 0.335 | 0.395 | 0.268 | 0.341 | 0.325 | 0.421 | 0.494 | 0.514 |
| | 37.5% | **0.253** | **0.334** | 0.337 | 0.392 | 0.316 | 0.361 | 0.401 | 0.406 | 0.362 | 0.413 | 0.304 | 0.360 | 0.397 | 0.472 | 0.498 | 0.529 |
| | 50% | **0.284** | **0.353** | 0.379 | 0.413 | 0.347 | 0.377 | 0.444 | 0.425 | 0.398 | 0.433 | 0.359 | 0.388 | 0.472 | 0.515 | 0.565 | 0.563 |
| ETTh2 | 12.5% | **0.105** | **0.209** | 0.230 | 0.332 | 0.118 | 0.225 | 0.134 | 0.244 | 0.290 | 0.359 | 0.108 | 0.223 | 0.278 | 0.342 | 1.030 | 0.636 |
| | 25% | **0.113** | **0.217** | 0.292 | 0.374 | 0.128 | 0.234 | 0.142 | 0.251 | 0.343 | 0.397 | 0.116 | 0.231 | 0.390 | 0.424 | 1.416 | 0.764 |
| | 37.5% | **0.125** | **0.228** | 0.347 | 0.405 | 0.140 | 0.244 | 0.155 | 0.261 | 0.383 | 0.422 | 0.129 | 0.241 | 0.532 | 0.510 | 1.612 | 0.834 |
| | 50% | **0.135** | **0.238** | 0.384 | 0.428 | 0.156 | 0.257 | 0.169 | 0.271 | 0.429 | 0.453 | 0.146 | 0.254 | 0.683 | 0.599 | 1.878 | 0.931 |
| ETTm1 | 12.5% | 0.132 | **0.230** | 0.159 | 0.262 | 0.142 | 0.237 | 0.154 | 0.245 | 0.266 | 0.337 | 0.143 | 0.237 | **0.116** | 0.223 | 0.339 | 0.396 |
| | 25% | 0.169 | **0.255** | 0.205 | 0.297 | 0.179 | 0.261 | 0.190 | 0.270 | 0.319 | 0.384 | 0.200 | 0.277 | **0.160** | 0.270 | 0.426 | 0.446 |
| | 37.5% | 0.219 | **0.284** | 0.267 | 0.338 | 0.233 | 0.293 | 0.253 | 0.306 | 0.179 | 0.296 | 0.264 | 0.313 | **0.212** | 0.315 | 0.515 | 0.496 |
| | 50% | 0.291 | **0.320** | 0.327 | 0.369 | 0.305 | 0.328 | 0.330 | 0.345 | **0.217** | 0.327 | 0.353 | 0.357 | 0.279 | 0.366 | 0.610 | 0.549 |
| ETTm2 | 12.5% | 0.062 | **0.153** | 0.114 | 0.224 | 0.064 | 0.153 | 0.067 | 0.159 | 0.111 | 0.229 | **0.056** | 0.157 | 0.130 | 0.238 | 0.854 | 0.535 |
| | 25% | 0.068 | **0.163** | 0.138 | 0.251 | 0.072 | 0.163 | 0.073 | 0.169 | 0.128 | 0.248 | **0.063** | 0.166 | 0.188 | 0.292 | 1.021 | 0.615 |
| | 37.5% | 0.074 | **0.172** | 0.160 | 0.272 | 0.081 | 0.175 | 0.083 | 0.181 | 0.146 | 0.264 | **0.072** | 0.177 | 0.249 | 0.346 | 1.209 | 0.706 |
| | 50% | **0.081** | **0.182** | 0.169 | 0.281 | 0.096 | 0.190 | 0.099 | 0.198 | 0.166 | 0.277 | 0.086 | 0.192 | 0.397 | 0.430 | 1.288 | 0.770 |
| Exchange | 12.5% | 0.010 | **0.060** | 0.078 | 0.190 | 0.010 | 0.061 | 0.010 | 0.063 | 0.521 | 0.503 | 0.018 | 0.089 | 0.116 | 0.233 | 1.112 | 0.671 |
| | 25% | **0.011** | 0.065 | 0.095 | 0.214 | 0.010 | **0.064** | 0.012 | 0.070 | 0.540 | 0.518 | 0.019 | 0.092 | 0.157 | 0.282 | 1.404 | 0.758 |
| | 37.5% | 0.014 | 0.072 | 0.105 | 0.229 | **0.013** | **0.072** | 0.014 | 0.076 | 0.576 | 0.539 | 0.022 | 0.098 | 0.190 | 0.318 | 1.759 | 0.885 |
| | 50% | 0.018 | **0.081** | 0.124 | 0.249 | **0.017** | 0.081 | 0.018 | 0.085 | 0.618 | 0.563 | 0.026 | 0.106 | 0.242 | 0.364 | 2.120 | 1.003 |
| Weather | 12.5% | 0.065 | **0.091** | 0.100 | 0.154 | 0.069 | 0.095 | 0.067 | 0.093 | 0.087 | 0.159 | **0.063** | 0.105 | 0.078 | 0.144 | 0.299 | 0.339 |
| | 25% | **0.067** | **0.098** | 0.117 | 0.187 | 0.079 | 0.110 | 0.072 | 0.098 | 0.072 | 0.131 | 0.069 | 0.112 | 0.081 | 0.151 | 0.346 | 0.376 |
| | 37.5% | **0.072** | **0.105** | 0.133 | 0.205 | 0.087 | 0.119 | 0.079 | 0.105 | 0.076 | 0.140 | 0.077 | 0.122 | 0.090 | 0.164 | 0.426 | 0.441 |
| | 50% | **0.077** | **0.111** | 0.093 | 0.136 | 0.097 | 0.128 | 0.093 | 0.124 | 0.081 | 0.147 | 0.089 | 0.133 | 0.098 | 0.172 | 0.411 | 0.430 |
| Electricity | 12.5% | 0.102 | 0.200 | **0.092** | **0.195** | 0.138 | 0.228 | 0.138 | 0.249 | 0.174 | 0.293 | 0.117 | 0.233 | 0.179 | 0.292 | 0.353 | 0.433 |
| | 25% | 0.110 | **0.206** | **0.103** | 0.206 | 0.149 | 0.235 | 0.155 | 0.261 | 0.188 | 0.304 | 0.129 | 0.247 | 0.195 | 0.304 | 0.372 | 0.443 |
| | 37.5% | 0.131 | 0.224 | **0.116** | **0.219** | 0.164 | 0.245 | 0.180 | 0.280 | 0.206 | 0.319 | 0.146 | 0.263 | 0.207 | 0.313 | 0.384 | 0.449 |
| | 50% | 0.161 | 0.246 | **0.132** | **0.234** | 0.180 | 0.257 | 0.213 | 0.303 | 0.226 | 0.335 | 0.171 | 0.285 | 0.218 | 0.321 | 0.394 | 0.454 |

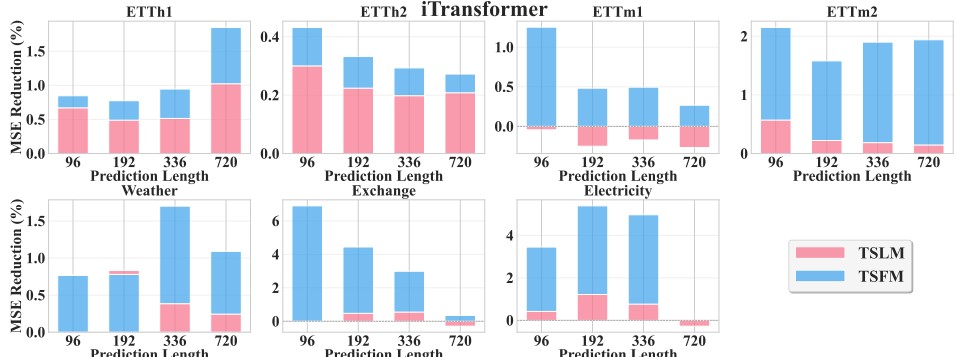

Figure 8: MSE reductions on forecasting under different backward model settings for iTransformer.

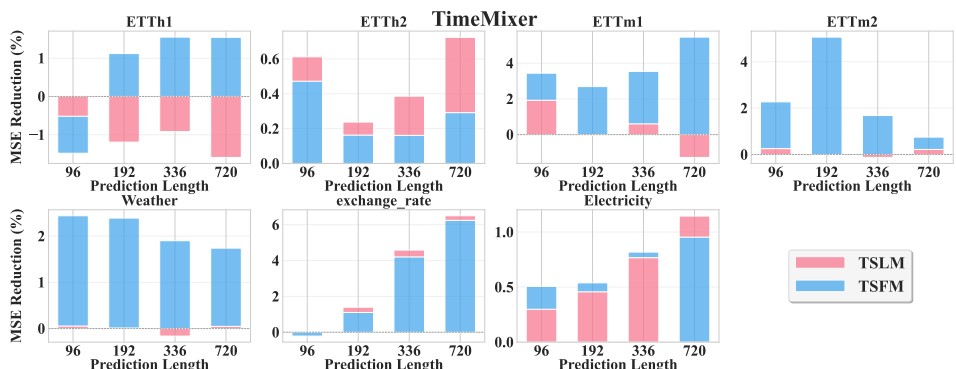

Figure 9: MSE reductions on forecasting under different backward model for TimeMixer.

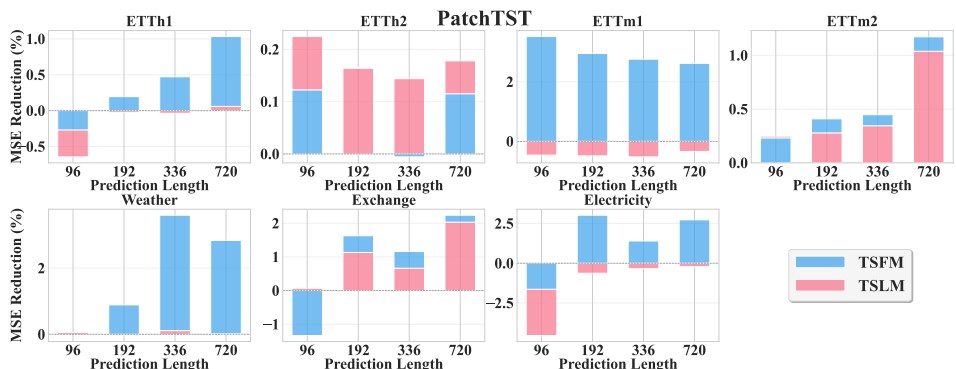

Figure 10: MSE reductions on forecasting under different backward model for PatchTST.

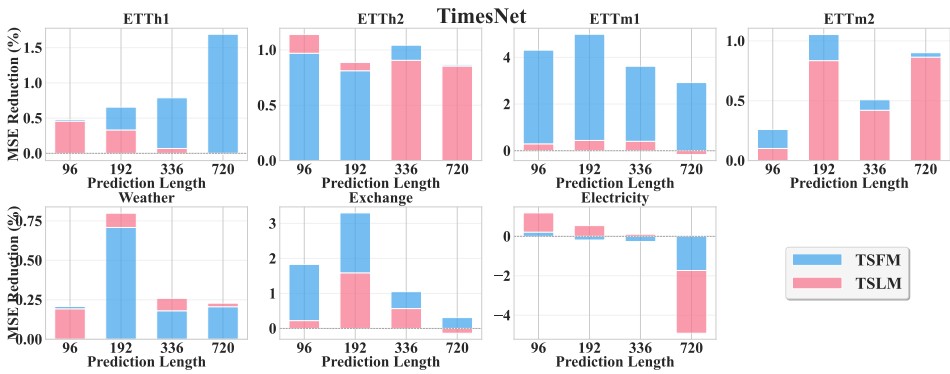

Figure 11: MSE reductions on forecasting under different backward model for TimesNet.

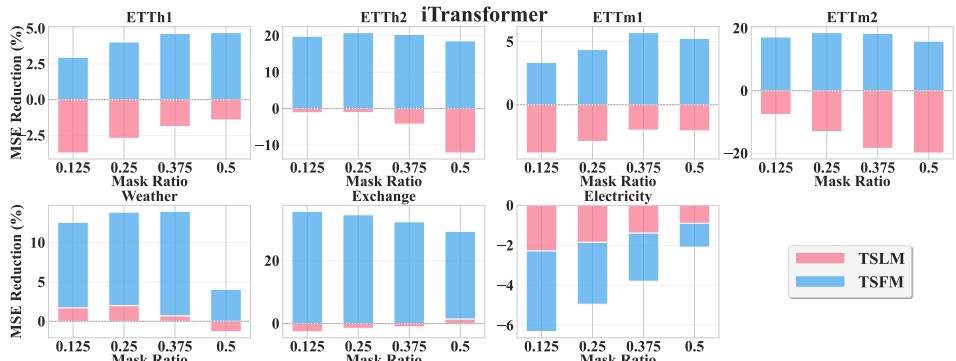

Figure 12: MSE reductions on imputation under different backward model for iTransformer.

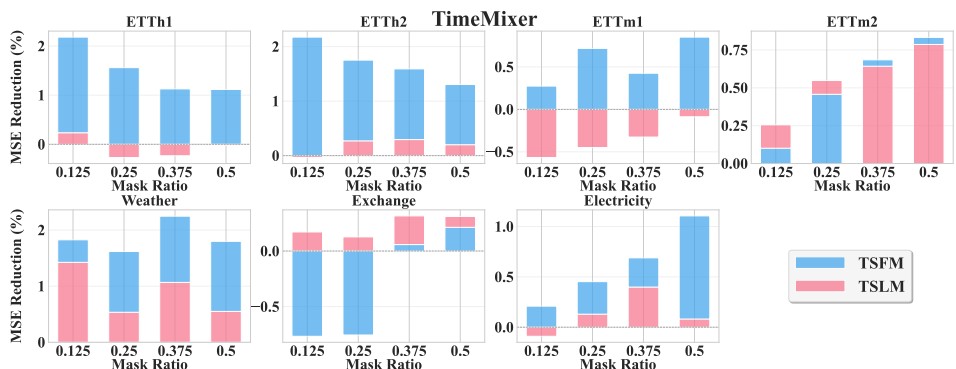

Figure 13: MSE reductions on imputation under different backward model for TimeMixer.

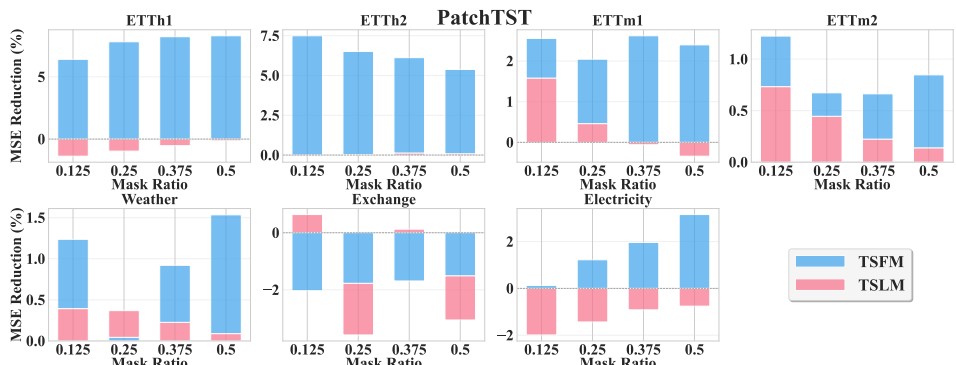

Figure 14: MSE reductions on imputation under different backward model for PatchTST.

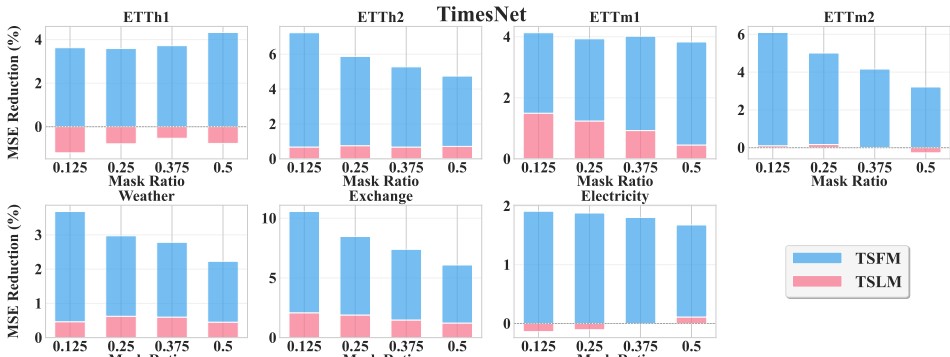

Figure 15: MSE reductions on imputation under different backward model for TimesNet.

# F DISCUSSION

## F.1 FUSION STRATEGY ANALYSIS

In the theoretical analysis, we show that the parameter $\sigma$ governs the sharpness of the weighting distribution used to aggregate candidate outputs based on reconstruction errors. As $\sigma$ approaches 0, the distribution becomes highly peaked, and the strategy converges to a Winner-Take-All (WTA) scheme that exclusively selects the candidate with the lowest reconstruction error. In contrast, as $\sigma$ tends to infinity, the distribution becomes uniform and corresponds to a Majority Voting (Majority) strategy in which all candidates receive equal weights and their outputs are simply averaged, regardless of their reconstruction errors. To assess the practical implications of these extremes, we choose four representative methods compare their performance against our adopted strategy with $\sigma = \sigma_e$, which softly weighs candidates based on their reconstruction errors. The results, shown in Fig. 16–23, indicate that both WTA and Majority lead to suboptimal performance on forecasting and imputation.

WTA is sensitive to noise and often results in unstable predictions, especially when the selected candidate is unreliable. In contrast, Majority averaging tends to dilute useful signals, as it ignores reconstruction quality and treats all candidates equally. Our approach with $\sigma = \sigma_e$ achieves the best performance by a balanced weighting that leverages reconstruction errors without overfitting to a single candidate. For imputation tasks, both WTA and our method outperform uniform averaging, demonstrating the value of reconstruction error as a proxy for candidate quality. However, WTA still exhibits instability across different models and datasets due to its sensitivity to outliers. These findings highlight the robustness and generalization benefits of our adaptive weighting scheme with $\sigma = \sigma_e$ in both forecasting and imputation scenarios.

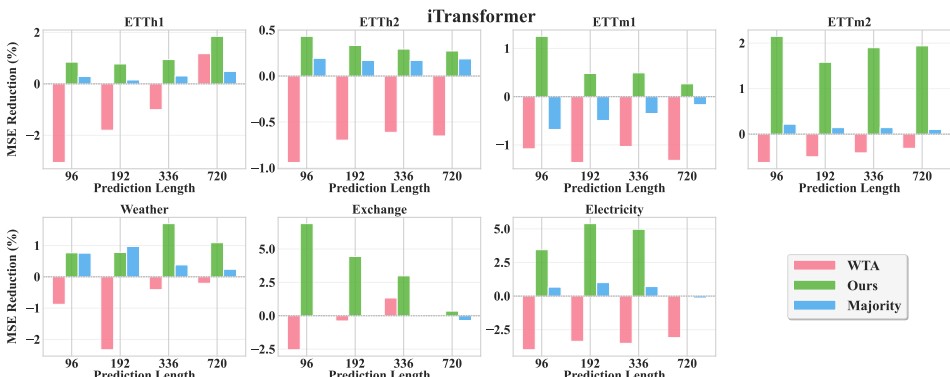

Figure 16: MSE reductions on forecasting under three schemes for iTransformer.

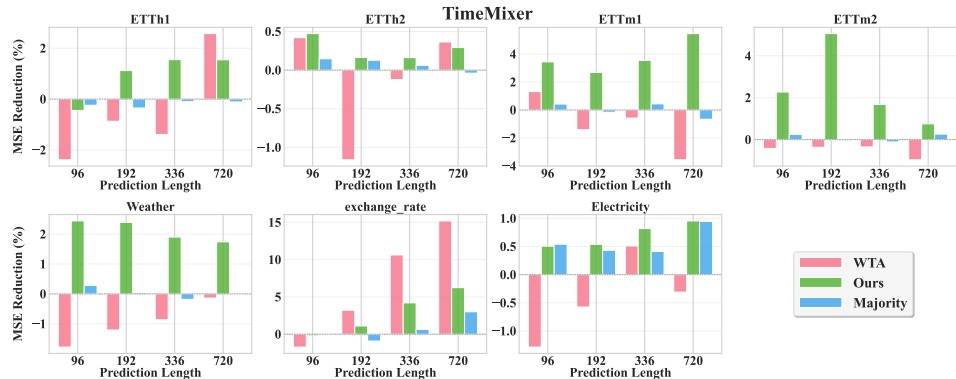

Figure 17: MSE reductions on forecasting under three schemes for TimeMixer.

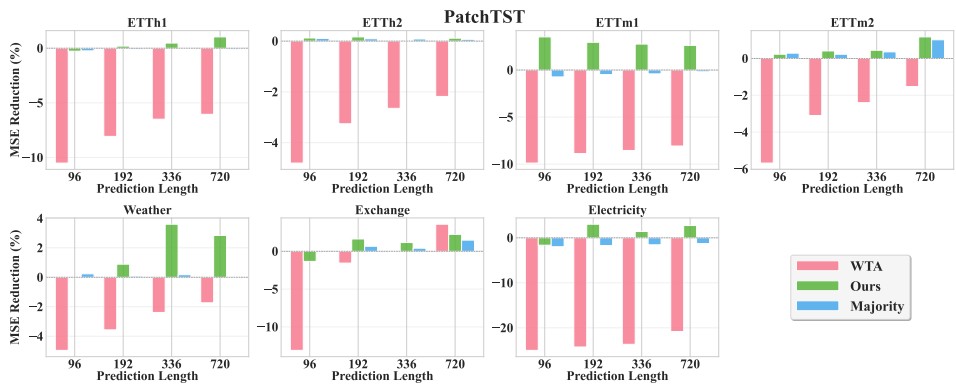

Figure 18: MSE reductions on forecasting under three schemes for PatchTST.

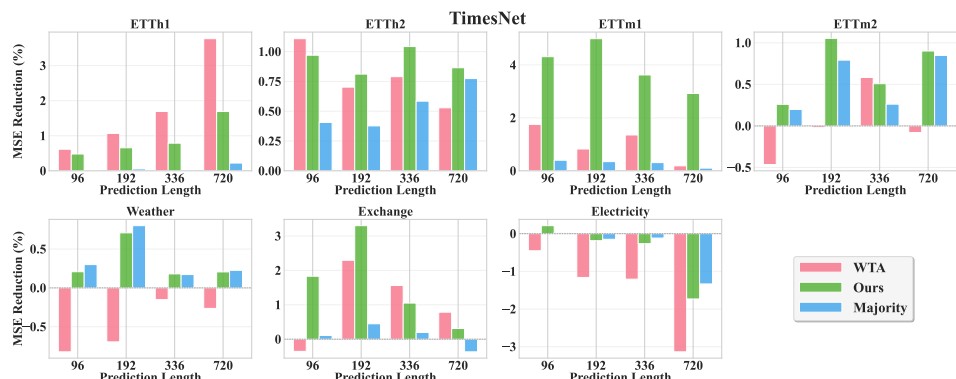

Figure 19: MSE reductions on forecasting under three schemes for TimesNet.

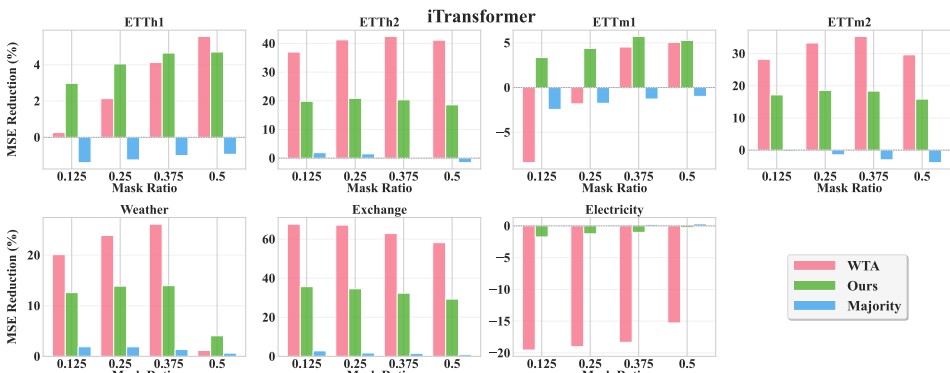

Figure 20: MSE reductions on imputation under three schemes for iTransformer.

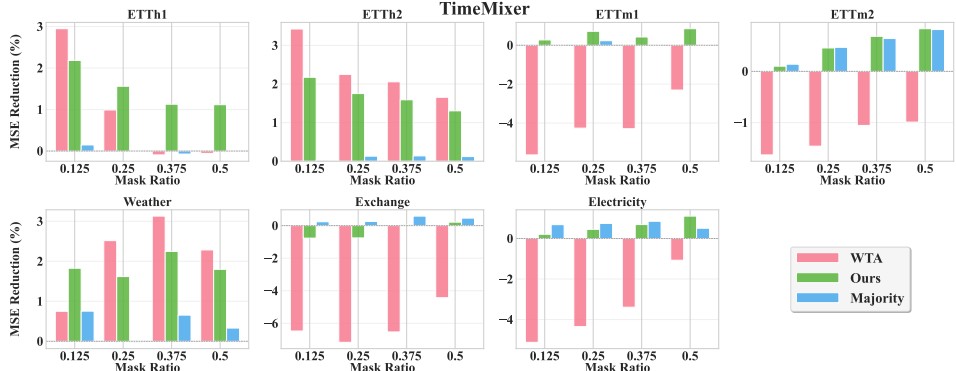

Figure 21: MSE reductions on imputation under three schemes for TimeMixer.

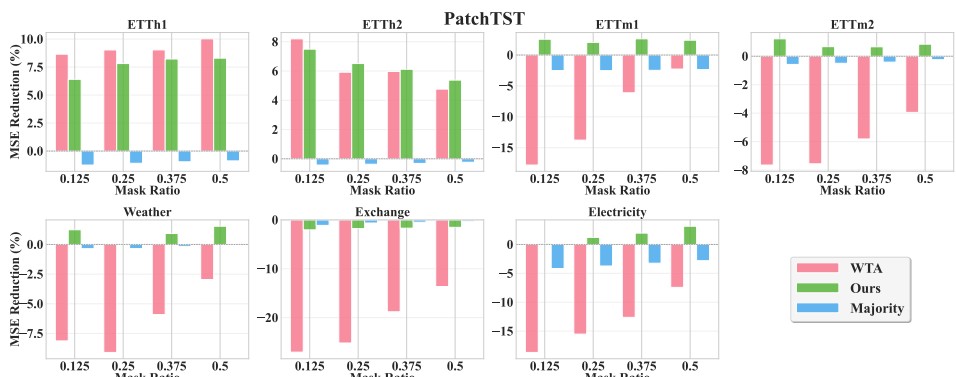

Figure 22: MSE reductions on imputation under three schemes for PatchTST.

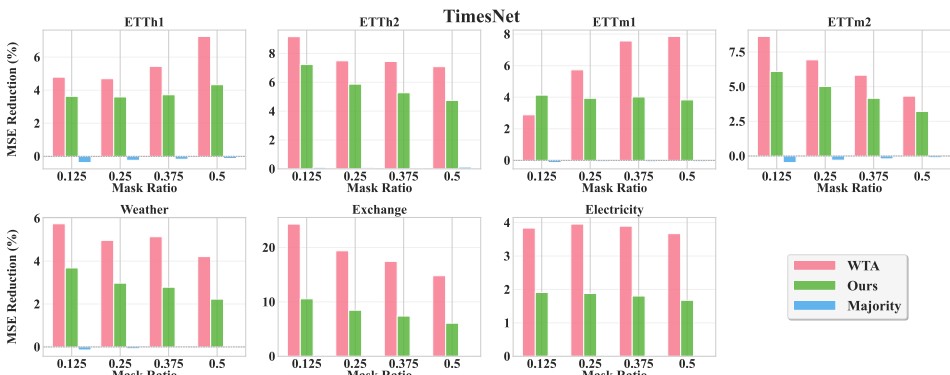

Figure 23: MSE reductions on imputation under three schemes for TimesNet.

Furthermore, we also compare the three aforementioned schemes with a weighting strategy based on the candidates' own uncertainty to more clearly demonstrate the effectiveness of our backward model. We conduct this analysis on representative datasets and backbone models, with the results summarized in Table 18. Our reconstruction-based weighting scheme consistently outperforms all baselines by a substantial margin, highlighting the crucial role and effectiveness of the backward model within the ITS framework. We briefly summarize the four weighting schemes:

- **Ours**: reconstruction-based weighting,

$$p(y \mid \boldsymbol{x}, D) = \sum_k w_k \, p(y \mid \boldsymbol{x}, \boldsymbol{W}_k),$$

  where $w_k$ is a weight derived from the reconstruction error of candidate $k$, which is obtained using the backward model.

- **Entropy**: entropy-based weighting using the predictive uncertainty of each candidate,

$$p(y \mid \boldsymbol{x}, D) = \sum_k \hat{w}_k \, p(y \mid \boldsymbol{x}, \boldsymbol{W}_k),$$

  where $\hat{w}_k$ is derived from the predictive uncertainty of candidate $k$.

- **Majority Voting (Majority)**: uniform averaging over all candidates,

$$p(y \mid \boldsymbol{x}, D) = \frac{1}{K} \sum_k p(y \mid \boldsymbol{x}, \boldsymbol{W}_k).$$

- **Winner-Take-All (WTA)**: selecting the single candidate with the lowest reconstruction error,

$$p(y \mid \boldsymbol{x}, D) = p(y \mid \boldsymbol{x}, \hat{\boldsymbol{W}}^*),$$

  where $\hat{\boldsymbol{W}}^*$ denotes the parameters of the candidate model with the lowest reconstruction error.

Table 18: Comparison of different weighting schemes for forecasting with a lookback length of $L = 96$.

| Method | | iTransformer | | | | | Crossformer | | | | | PatchTST | | | | |
|---|---|---|---|---|---|---|---|---|---|---|---|---|---|---|---|---|
| Dataset | | Vanilla | Ours | Entropy | Majority | WTA | Vanilla | Ours | Entropy | Majority | WTA | Vanilla | Ours | Entropy | Majority | WTA |
| ETTh1 | 96 | 0.411 | **0.407** | 0.409 | 0.410 | 0.423 | 0.395 | **0.391** | 0.399 | 0.396 | 0.412 | **0.408** | 0.409 | 0.414 | 0.409 | 0.451 |
| | 192 | 0.462 | **0.459** | 0.461 | 0.461 | 0.470 | 0.628 | **0.614** | 0.628 | 0.631 | 0.629 | 0.460 | **0.459** | 0.469 | 0.461 | 0.497 |
| | 336 | 0.492 | **0.487** | 0.488 | 0.490 | 0.497 | 0.665 | **0.657** | 0.673 | 0.669 | 0.674 | 0.494 | **0.492** | 0.503 | 0.495 | 0.526 |
| | 720 | 0.512 | **0.503** | 0.508 | 0.510 | 0.506 | 0.887 | 0.825 | 0.861 | 0.854 | **0.789** | 0.495 | **0.489** | 0.499 | 0.495 | 0.525 |
| Electricity | 96 | 0.150 | **0.145** | 0.147 | 0.149 | 0.156 | 0.219 | **0.218** | 0.226 | 0.219 | 0.248 | **0.193** | 0.196 | 0.199 | 0.197 | 0.241 |
| | 192 | 0.167 | **0.158** | 0.164 | 0.165 | 0.172 | **0.237** | 0.238 | 0.239 | 0.238 | 0.265 | 0.199 | **0.193** | 0.197 | 0.203 | 0.248 |
| | 336 | 0.181 | **0.172** | 0.178 | 0.180 | 0.188 | 0.785 | 0.778 | 0.781 | 0.786 | **0.775** | 0.214 | **0.207** | 0.229 | 0.217 | 0.264 |
| | 720 | 0.222 | **0.210** | 0.219 | 0.222 | 0.229 | 0.813 | **0.810** | 0.811 | 0.813 | 0.815 | 0.256 | **0.249** | 0.251 | 0.259 | 0.309 |

Table 19: Comparison of different weighting schemes for imputation with a sequence length of $L = 96$.

| Method | | iTransformer | | | | | Crossformer | | | | | PatchTST | | | | |
|---|---|---|---|---|---|---|---|---|---|---|---|---|---|---|---|---|
| Dataset | | Vanilla | Ours | Entropy | Majority | WTA | Vanilla | Ours | Entropy | Majority | WTA | Vanilla | Ours | Entropy | Majority | WTA |
| ETTh1 | 12.5% | 0.262 | **0.255** | 0.267 | 0.269 | 0.262 | 0.300 | **0.285** | 0.302 | 0.305 | 0.287 | 0.320 | 0.300 | 0.320 | 0.324 | **0.293** |
| | 25% | 0.303 | **0.290** | 0.304 | 0.306 | 0.296 | 0.335 | **0.315** | 0.335 | 0.340 | 0.316 | 0.370 | 0.342 | 0.368 | 0.375 | **0.337** |
| | 37.5% | 0.337 | **0.321** | 0.343 | 0.340 | 0.323 | 0.362 | **0.339** | 0.368 | 0.367 | 0.342 | 0.401 | 0.368 | 0.397 | 0.405 | **0.365** |
| | 50% | 0.379 | 0.361 | 0.381 | 0.383 | **0.358** | 0.398 | **0.373** | 0.394 | 0.403 | 0.375 | 0.444 | 0.407 | 0.449 | 0.448 | **0.399** |
| Electricity | 12.5% | **0.092** | 0.094 | 0.093 | **0.092** | 0.110 | 0.174 | **0.167** | 0.174 | 0.177 | 0.171 | **0.138** | **0.138** | 0.142 | 0.144 | 0.164 |
| | 25% | **0.103** | 0.104 | 0.104 | **0.103** | 0.123 | 0.188 | **0.178** | 0.195 | 0.192 | 0.182 | 0.155 | **0.153** | 0.156 | 0.161 | 0.179 |
| | 37.5% | **0.116** | 0.117 | 0.117 | **0.116** | 0.137 | 0.206 | **0.195** | 0.201 | 0.210 | 0.200 | 0.180 | **0.176** | 0.180 | 0.186 | 0.203 |
| | 50% | **0.132** | 0.133 | 0.134 | **0.132** | 0.153 | 0.226 | **0.214** | 0.223 | 0.229 | 0.220 | 0.213 | **0.206** | 0.214 | 0.219 | 0.229 |

## F.2 SENSITIVITY ANALYSIS FOR THE NUMBER OF CANDIDATES

Since our candidates are generated via MC Dropout, a natural concern is whether they might be too narrow to satisfy the diversity requirements of ITS. To assess this concern, we conduct a diversity analysis on the iTransformer, comparing MC Dropout–generated candidates with those produced by a flow-based generative model. We quantify diversity using Norm Pairwise L2, which measures the average normalized distance between different candidates, and Norm Ensemble Variance, which measures the normalized variance across the $K$ candidates. As shown in Tables 20 and 21, the diversity metrics of MC Dropout are only slightly lower than those of the flow-based model, indicating that MC Dropout candidates are sufficiently diverse in our setting. In particular, increasing the number of samples $K$ and appropriately tuning the dropout rate $p_d$ provide ample diversity in practice for supporting ITS.

Table 20: Diversity metrics of MC Dropout candidates under different $K$ and dropout rates $p_d$.

| $K$ | Dropout $p_d$ | Norm Pairwise L2 | Norm Ensemble Variance |
|---|---|---|---|
| 8 | 0.1 | 0.2238 | 0.0219 |
| 32 | 0.1 | 0.2441 | 0.0263 |
| 64 | 0.1 | 0.2812 | 0.0389 |
| 64 | 0.2 | 0.3322 | 0.0543 |

Table 21: Diversity metrics of flow-based model candidates under different $K$.

| $K$ | Norm Pairwise L2 | Norm Ensemble Variance |
|---|---|---|
| 8 | 0.2397 | 0.0256 |
| 32 | 0.2653 | 0.0315 |
| 64 | 0.3128 | 0.0447 |

We also investigate the impact of the number of candidate outputs $K$ on the performance of our ITS across four representative methods, with results presented in Fig. 24–31. The experiments show that increasing $K$ consistently leads to reductions in MSE across all evaluated datasets. This improvement can be attributed to the availability of more diverse candidates, which allows the ITS to make more reliable use of the reconstruction signals provided by the backward model. As $K$ increases, the ITS becomes better at distinguishing between high- and low-quality predictions, leading to improved overall accuracy. The performance gains are particularly notable when $K$ increases from 8 to 64, while further increases beyond that yield diminishing improvements. This suggests that a moderately large candidate set is sufficient to capture predictive uncertainty and support robust inference under the reconstruction-guided evaluation process.

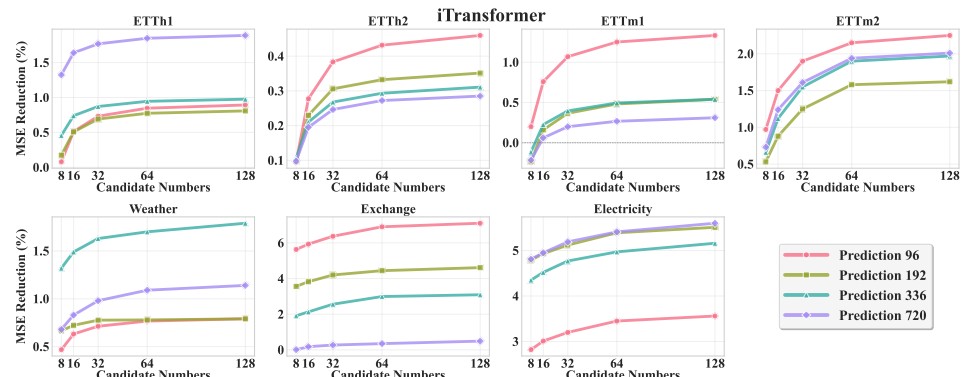

Figure 24: MSE reductions on forecasting with different numbers of candidates for iTransformer.

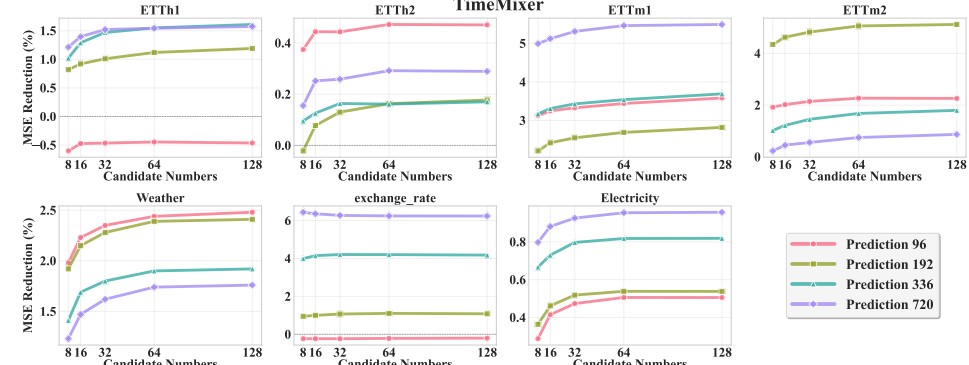

Figure 25: MSE reductions on forecasting with different numbers of candidates for TimeMixer.

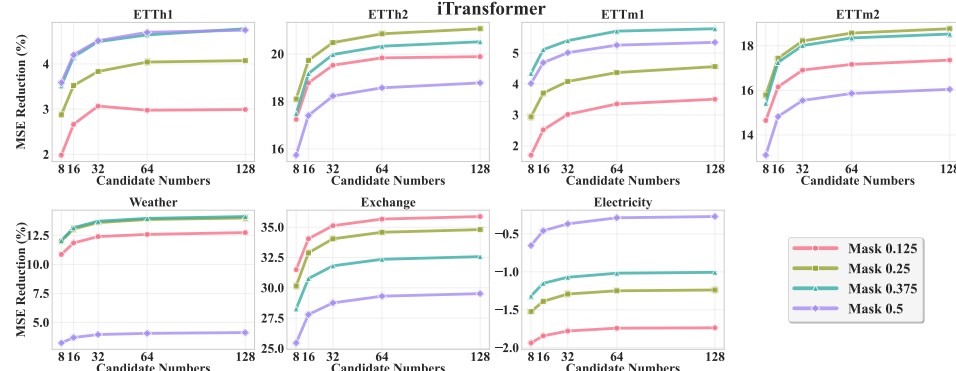

Figure 28: MSE reductions on imputation with different numbers of candidates for iTransformer.

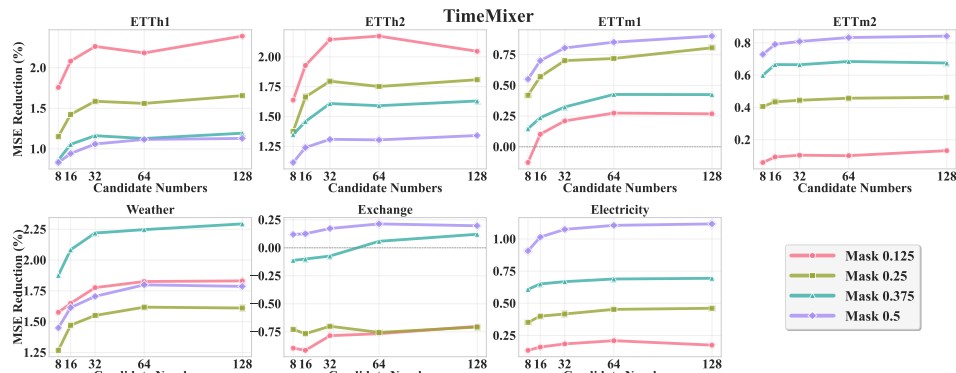

Figure 29: MSE reductions on imputation with different numbers of candidates for TimeMixer.

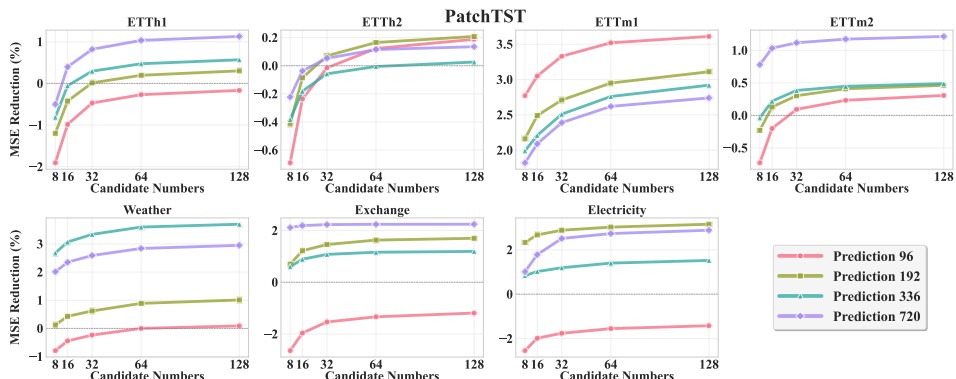

Figure 26: MSE reductions on forecasting with different numbers of candidates for PatchTST.

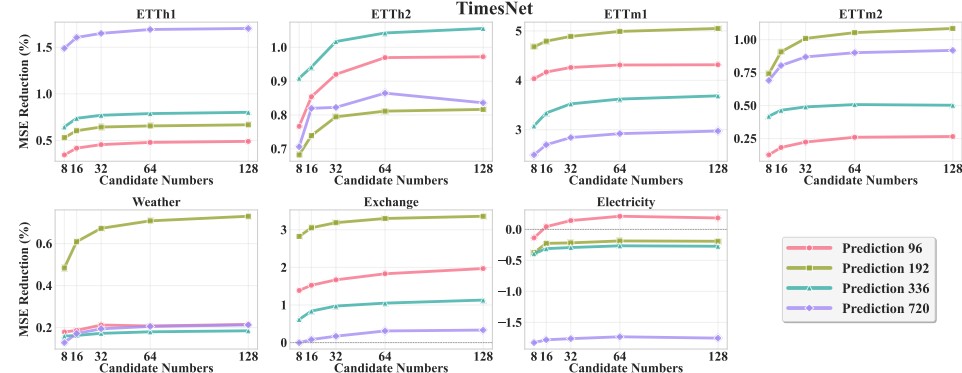

Figure 27: MSE reductions on forecasting with different numbers of candidates for TimesNet.

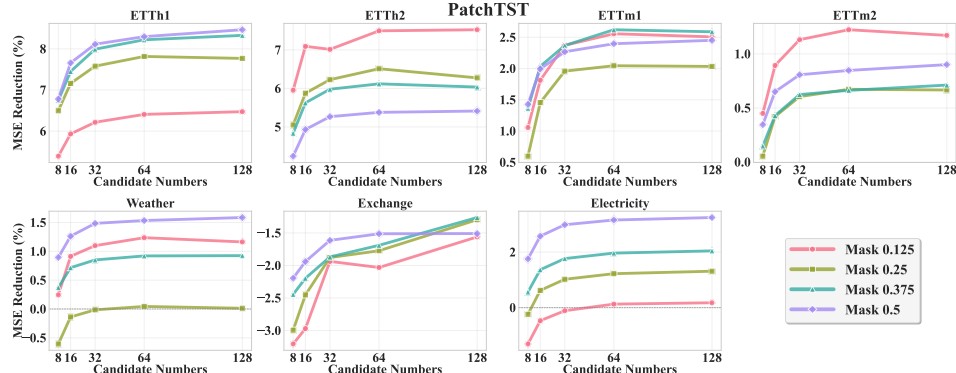

Figure 30: MSE reductions on imputation with different numbers of candidates for PatchTST.

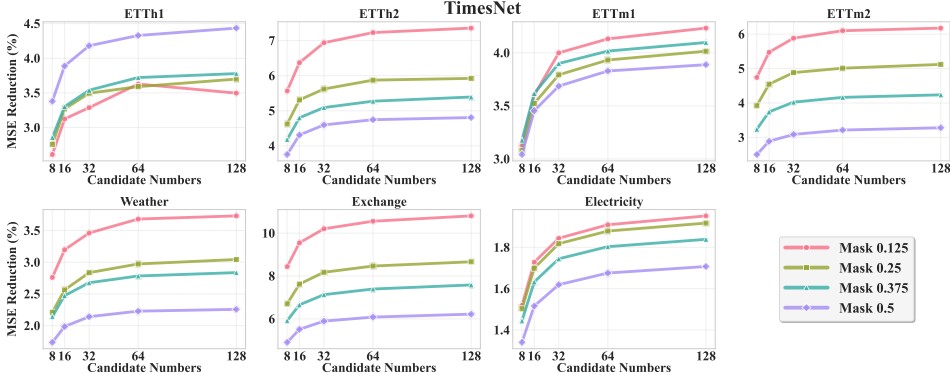

Figure 31: MSE reductions on imputation with different numbers of candidates for TimesNet.

### F.3 SENSITIVITY ANALYSIS FOR DROPOUT RATIO $p_d$

ITS leverages MC Dropout to introduce stochasticity during candidate generation, enabling predictive diversity for reconstruction-based evaluation. To investigate the impact of different levels of stochasticity, we conduct experiments with dropout ratios $p_d \in \{0.05, 0.1, 0.2\}$ on four representative methods, and report the corresponding results in Fig. 32–39. The results show that both extremely low and excessively high dropout ratios lead to suboptimal performance. When the dropout ratio is too low, the forward model tends to generate overly similar candidate outputs, reducing diversity and limiting the effectiveness of reconstruction-based selection. In contrast, a high dropout ratio injects too much noise, which compromises prediction accuracy and causes unstable inference. Among the evaluated settings, a dropout ratio of 0.1 consistently yields the most stable and accurate performance across most models and datasets. Notably, this configuration aligns with the dropout setting used during the pretraining phase of the forward model in the majority of methods, helping to maintain a balance between candidate diversity and variance, and ensuring that the generated candidates remain both informative and reliable.

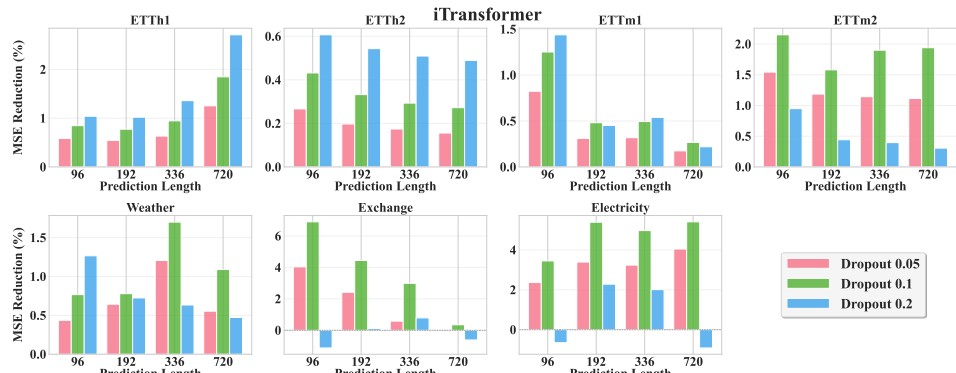

Figure 32: MSE reductions on forecasting under different dropout ratio settings for iTransformer.

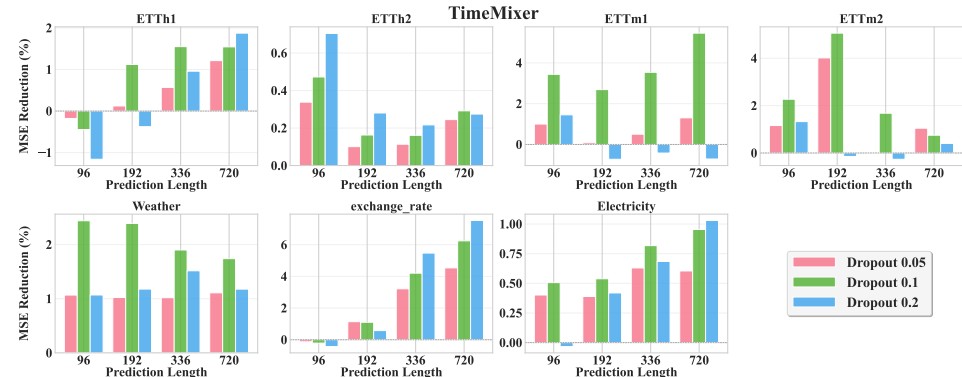

Figure 33: MSE reductions on forecasting under different dropout ratio settings for TimeMixer.

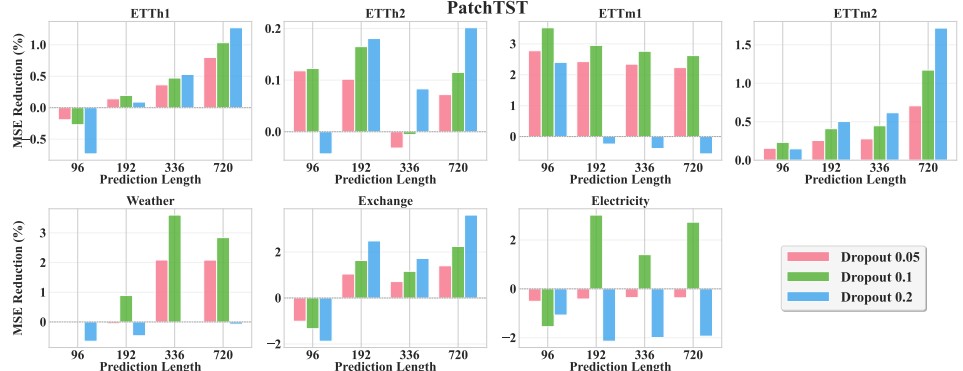

Figure 34: MSE reductions on forecasting under different dropout ratio settings for PatchTST.

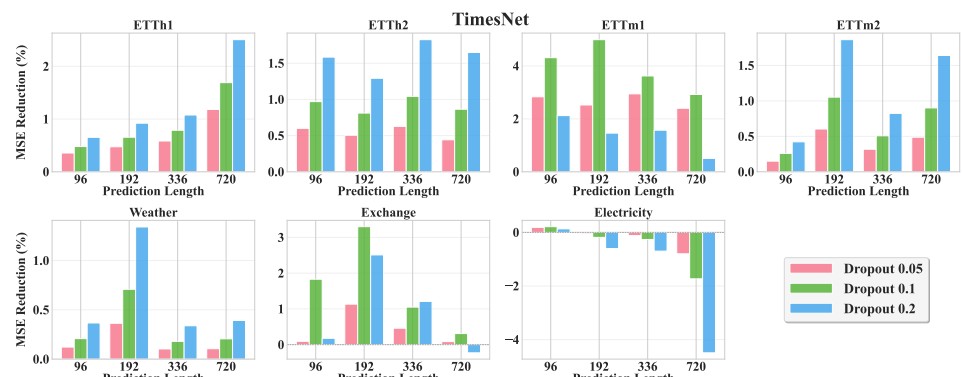

Figure 35: MSE reductions on forecasting under different dropout ratio settings for TimesNet.

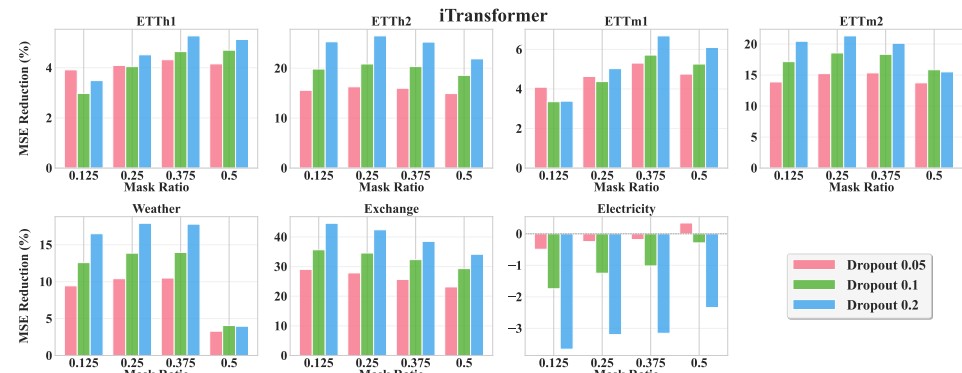

Figure 36: MSE reductions on imputation under different dropout ratio settings for iTransformer.

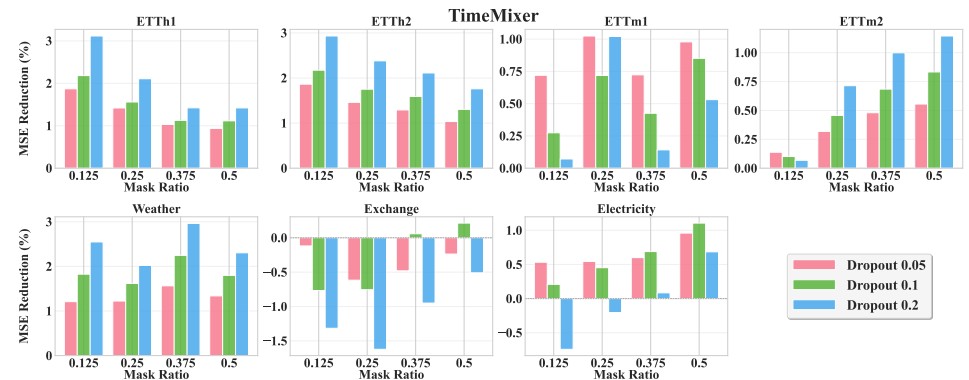

Figure 37: MSE reductions on imputation under different dropout ratio settings for TimeMixer.

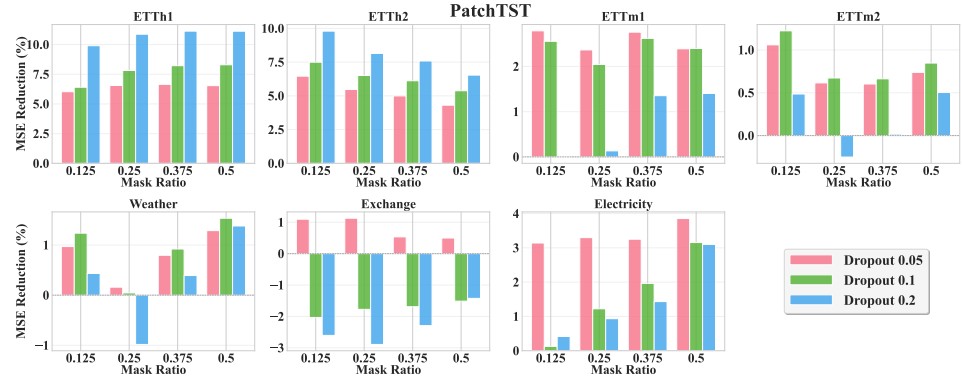

Figure 38: MSE reductions on imputation under different dropout ratio settings for PatchTST.

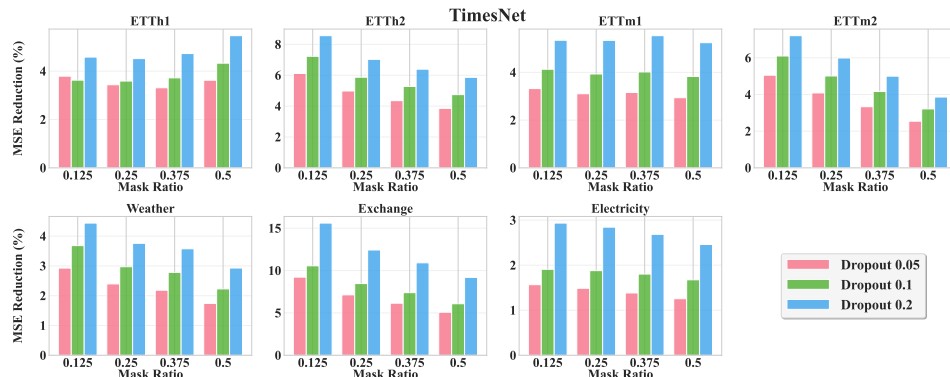

Figure 39: MSE reductions on imputation under different dropout ratio settings for TimesNet.

## F.4 APPLICATION TO LLMS-BASED MODEL

With the immense advancement of large language models (LLMs) infrastructure, LLMs-based methods for time series forecasting have witnessed rapid progress in recent years. To assess the applicability of our ITS method in this emerging paradigm, we conduct experiments on two representative and influential frameworks: GPT4TS (Zhou et al., 2023) and AutoTimes (Liu et al., 2024b), both of which adapt LLMs to time series forecasting tasks. GPT4TS provides a unified framework for various time-series tasks by leveraging a frozen GPT-2. In contrast, AutoTimes projects time series into the LLM's token embedding space and exploits the autoregressive nature of decoder-only LLMs for future prediction, supporting flexible integration with different LLM backbones.

In our experiments, we follow the settings from the original papers and adopt GPT-2 as the LLM component for both frameworks. We apply ITS to both GPT4TS and AutoTimes and evaluate its effectiveness on forecasting and imputation tasks. For forecasting, models are trained with a lookback window of $L = 672$ and prediction lengths of $S \in \{96, 192, 336, 720\}$. For imputation, the input sequence length is $L = 96$ with fixed subsequence length of $N = 12$, and mask rates are set to $\{12.5\%, 25\%, 37.5\%, 50\%\}$. All training configurations strictly follow those in the original papers. During inference, we set $K = 32$ candidate outputs, and all other settings are kept consistent with those used in the main experiments. The dropout ratio $p_d$ for both methods remain aligned with their respective pretraining setups. Table 22 presents the full MSE results for forecasting, demonstrating consistent improvements across datasets and prediction lengths. Table 23 provides the full MSE results for imputation using GPT4TS, as AutoTimes does not provide an official implementation for imputation in its released codebase. Our results demonstrate that ITS can further enhance the performance of these already strong LLM-based models. It consistently yields improvements in both forecasting and imputation tasks across diverse datasets and a range of prediction horizons or masking rates, highlighting its robustness and broad applicability.

Table 22: Full MSE results for forecasting with prediction lengths $S \in \{96, 192, 336, 720\}$ on seven datasets, using $K = 32$ and a fixed lookback length of $L = 672$. Vanilla denotes the original methods, while **+ITS** represents our proposed framework.

| Method | Dataset | ETTh1 Vanilla | +ITS | ETTh2 Vanilla | +ITS | ETTm1 Vanilla | +ITS | ETTm2 Vanilla | +ITS | Exchange Vanilla | +ITS | Weather Vanilla | +ITS | Electricity Vanilla | +ITS |
|---|---|---|---|---|---|---|---|---|---|---|---|---|---|---|---|
| GPT4TS | 96 | 0.373 | 0.373 | 0.294 | **0.285** | 0.299 | **0.288** | 0.172 | **0.168** | **0.107** | 0.110 | **0.149** | 0.150 | 0.137 | **0.129** |
| | 192 | 0.407 | **0.404** | 0.357 | **0.349** | 0.335 | **0.325** | 0.240 | **0.229** | 0.222 | **0.220** | 0.154 | 0.154 | 0.154 | **0.146** |
| | 336 | 0.446 | **0.440** | 0.376 | **0.366** | 0.363 | **0.352** | 0.287 | **0.277** | 0.399 | **0.386** | 0.249 | **0.246** | 0.169 | **0.161** |
| | 720 | 0.495 | **0.488** | 0.438 | **0.429** | 0.416 | **0.404** | 0.367 | **0.361** | **0.974** | 0.975 | 0.324 | **0.317** | 0.211 | **0.204** |
| AutoTimes | 96 | 0.358 | **0.353** | 0.291 | **0.286** | 0.282 | **0.279** | 0.173 | **0.169** | 0.113 | **0.104** | 0.160 | 0.160 | 0.201 | **0.199** |
| | 192 | 0.388 | **0.383** | 0.351 | **0.347** | 0.332 | **0.327** | 0.235 | **0.231** | **0.231** | 0.236 | 0.210 | **0.209** | 0.248 | **0.246** |
| | 336 | 0.405 | **0.397** | 0.375 | **0.371** | 0.368 | **0.363** | 0.291 | **0.288** | **0.410** | 0.422 | 0.266 | **0.261** | **0.183** | 0.184 |
| | 720 | 0.422 | **0.407** | 0.438 | **0.427** | 0.426 | **0.425** | 0.373 | **0.370** | **1.062** | 1.118 | 0.341 | **0.331** | **0.223** | 0.225 |

## F.5 ANALYSIS FOR THE COMPUTATIONAL COST

We evaluate the computational cost introduced by our ITS to assess its practicality in real-world applications. For fairness, we adopt the batch sizes specified by the original authors for each method.

Table 23: Full MSE results for imputation on seven datasets. We randomly mask $\{12.5\%, 25\%, 37.5\%, 50\%\}$ of subsequences in time series with length $L = 96$. The results use $K = 32$ and a fixed subsequence length of $N = 12$. Vanilla denotes the original methods, while **+ITS** represents our proposed framework.

| Method | | ETTh1 | | ETTh2 | | ETTm1 | | ETTm2 | | Exchange | | Weather | | Electricity | |
|---|---|---|---|---|---|---|---|---|---|---|---|---|---|---|---|---|
| Dataset | | Vanilla | +ITS | Vanilla | +ITS | Vanilla | +ITS | Vanilla | +ITS | Vanilla | +ITS | Vanilla | +ITS | Vanilla | +ITS |
| GPT4TS | 12.5% | 0.116 | **0.111** | 0.086 | **0.084** | 0.102 | **0.096** | 0.046 | 0.046 | 0.021 | **0.020** | 0.055 | **0.054** | 0.146 | **0.139** |
| | 25% | 0.157 | **0.148** | 0.093 | **0.090** | 0.143 | **0.133** | 0.053 | **0.052** | 0.023 | **0.022** | 0.064 | **0.063** | 0.159 | **0.150** |
| | 37.5% | 0.222 | **0.203** | 0.109 | **0.105** | 0.197 | **0.181** | 0.063 | **0.061** | 0.025 | **0.024** | 0.074 | **0.072** | 0.179 | **0.168** |
| | 50% | 0.285 | **0.262** | 0.125 | **0.121** | 0.270 | **0.246** | 0.076 | **0.075** | 0.029 | **0.028** | 0.086 | **0.084** | 0.208 | **0.195** |

Specifically, we measure the computational overhead with $K = 64$ and compare the results against those of the corresponding vanilla models. Although ITS introduces additional computations, such as generating candidate outputs, performing reconstruction, and applying a weighting scheme, the overall increase in computational cost remains modest. As shown in Fig. 40-41 and Tables 24-25, the architectural complexity of different models leads to substantial variations in computational cost.

Among the evaluated methods, iTransformer and TimeMixer, which rely on attention-based and MLP-based designs respectively, exhibit the lowest inference-time cost. In contrast, TimeXer incorporates variable embeddings, while PatchTST employs channel-wise processing; both lead to moderate increases in inference-time cost. Crossformer and MICN involve more complex attention or convolutional mechanisms and therefore incur higher costs. TimesNet relies on frequency-domain transformations, whereas Autoformer integrates decomposition modules for trend and seasonal component separation; both exhibit the highest computational cost due to their architectural complexity. Timer is a foundation model pretrained on large-scale time-series data that adopts an autoregressive architecture and thus incurs a moderate inference-time cost. While ITS introduces additional computations during inference, it consistently improves performance across a wide range of models and datasets with only modest overall overhead, demonstrating strong robustness and practical applicability. The inference-time cost tends to increase with the dimensionality of the time series, especially in multivariate settings, as ITS performs separate computations for each dimension. Among all components, candidate generation accounts for the majority of the overhead, with more than 85% of the total cost attributed to the use of MC Dropout. This is primarily due to the repeated stochastic forward passes required to produce diverse candidate outputs. Improving the efficiency of candidate generation, particularly by optimizing MC Dropout, can lead to a substantial reduction in the overall inference-time cost of ITS.

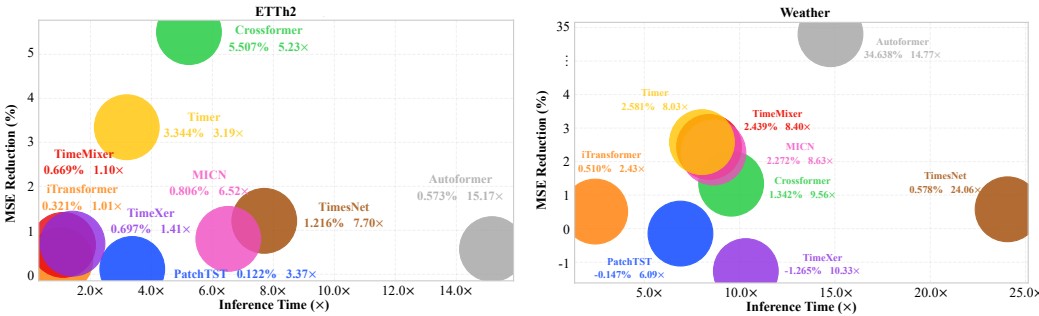

Figure 40: Computational cost of ITS for forecasting with input-96-predict-96 on ETTh2 and Weather.

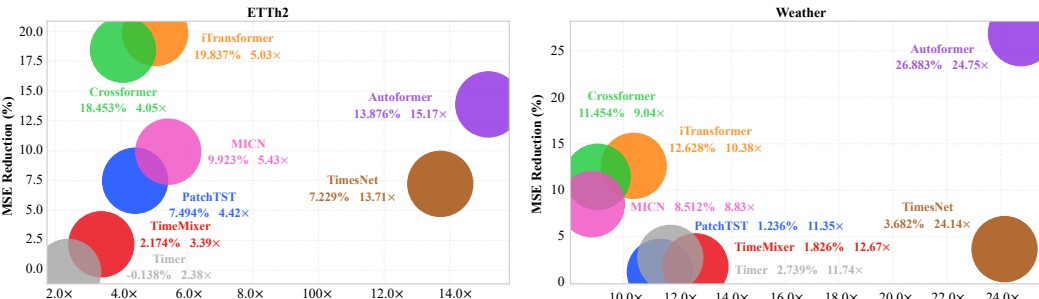

Figure 41: Computational cost of ITS for imputation with input-96 and randomly mask 12.5% on ETTh2 and Weather.

Table 24: Inference time comparison of Vanilla and ITS with a lookback length of $L = 96$.

| Method | iTransformer | | PatchTST | |
|---|---|---|---|---|
| Dataset | Vanilla (ms / example) | +ITS (ms / example) | Vanilla (ms / example) | +ITS (ms / example) |
| ETTh2 96 | 0.094 | 0.099 | 0.112 | 0.351 |
| ETTh2 192 | 0.128 | 0.193 | 0.124 | 0.403 |
| ETTh2 336 | 0.143 | 0.283 | 0.133 | 0.534 |
| ETTh2 720 | 0.184 | 0.423 | 0.221 | 0.948 |
| Weather 96 | 0.257 | 0.598 | 0.112 | 0.684 |
| Weather 192 | 0.271 | 0.626 | 0.128 | 0.898 |
| Weather 336 | 0.313 | 0.892 | 0.165 | 1.235 |
| Weather 720 | 0.374 | 0.998 | 0.222 | 1.753 |
| Electricity 96 | 0.482 | 2.882 | 0.838 | 8.472 |
| Electricity 192 | 0.602 | 3.654 | 0.911 | 9.315 |
| Electricity 336 | 0.705 | 4.490 | 0.944 | 11.987 |
| Electricity 720 | 1.774 | 11.125 | 2.081 | 22.150 |

Table 25: Inference time comparison of Vanilla and ITS for imputation with a sequence length of $L = 96$.

| Method | iTransformer | | PatchTST | |
|---|---|---|---|---|
| Dataset | Vanilla (ms / example) | +ITS (ms / example) | Vanilla (ms / example) | +ITS (ms / example) |
| ETTh2 12.5% | 0.107 | 0.546 | 0.119 | 0.489 |
| ETTh2 25% | 0.158 | 0.780 | 0.108 | 0.446 |
| ETTh2 37.5% | 0.161 | 0.813 | 0.089 | 0.410 |
| ETTh2 50% | 0.169 | 0.846 | 0.106 | 0.454 |
| Weather 12.5% | 0.117 | 1.145 | 0.218 | 2.443 |
| Weather 25% | 0.111 | 1.139 | 0.182 | 1.936 |
| Weather 37.5% | 0.112 | 1.180 | 0.223 | 2.457 |
| Weather 50% | 0.107 | 1.088 | 0.191 | 2.181 |
| Electricity 12.5% | 0.495 | 7.033 | 1.143 | 15.019 |
| Electricity 25% | 0.431 | 6.331 | 1.388 | 19.196 |
| Electricity 37.5% | 0.449 | 6.752 | 1.451 | 20.357 |
| Electricity 50% | 0.433 | 6.352 | 1.391 | 19.014 |

### F.6 SENSITIVITY TO RANDOM SEED

To evaluate the impact of random seeds, we conducted five runs with different seed values on representative datasets and models, and the corresponding standard deviations are reported in Tables 26 and 27. We observe that ITS is minimally sensitive to random seed variations.

Table 26: Sensitivity of ITS to random seeds for forecasting with a lookback length of $L = 96$.

| Method | iTransformer | | PatchTST | |
|---|---|---|---|---|
| Dataset | Vanilla | +ITS | Vanilla | +ITS |
| ETTh1 96 | $0.412_{\pm 0.001}$ | $\mathbf{0.406}_{\pm 0.001}$ | $0.409_{\pm 0.000}$ | $\mathbf{0.408}_{\pm 0.000}$ |
| ETTh1 192 | $0.464_{\pm 0.001}$ | $\mathbf{0.457}_{\pm 0.000}$ | $0.463_{\pm 0.003}$ | $\mathbf{0.457}_{\pm 0.001}$ |
| ETTh1 336 | $0.491_{\pm 0.001}$ | $\mathbf{0.482}_{\pm 0.001}$ | $0.495_{\pm 0.001}$ | $\mathbf{0.490}_{\pm 0.001}$ |
| ETTh1 720 | $0.517_{\pm 0.001}$ | $\mathbf{0.505}_{\pm 0.000}$ | $0.496_{\pm 0.002}$ | $\mathbf{0.488}_{\pm 0.001}$ |
| Electricity 96 | $0.151_{\pm 0.000}$ | $\mathbf{0.145}_{\pm 0.001}$ | $0.195_{\pm 0.000}$ | $0.195_{\pm 0.001}$ |
| Electricity 192 | $0.164_{\pm 0.001}$ | $\mathbf{0.152}_{\pm 0.002}$ | $0.196_{\pm 0.001}$ | $\mathbf{0.188}_{\pm 0.000}$ |
| Electricity 336 | $0.178_{\pm 0.002}$ | $\mathbf{0.168}_{\pm 0.000}$ | $0.216_{\pm 0.001}$ | $\mathbf{0.204}_{\pm 0.002}$ |
| Electricity 720 | $0.224_{\pm 0.001}$ | $\mathbf{0.211}_{\pm 0.002}$ | $0.253_{\pm 0.001}$ | $\mathbf{0.240}_{\pm 0.000}$ |

Table 27: Sensitivity of ITS to random seeds for imputation with sequence length $L = 96$.

| Method | | iTransformer | | PatchTST | |
|---|---|---|---|---|---|
| Dataset | | Vanilla | +ITS | Vanilla | +ITS |
| ETTh1 | 12.5% | $0.266_{\pm 0.003}$ | $\mathbf{0.255}_{\pm 0.001}$ | $0.325_{\pm 0.002}$ | $\mathbf{0.299}_{\pm 0.001}$ |
| | 0.25 | $0.301_{\pm 0.001}$ | $\mathbf{0.287}_{\pm 0.000}$ | $0.368_{\pm 0.003}$ | $\mathbf{0.336}_{\pm 0.001}$ |
| | 0.375 | $0.336_{\pm 0.001}$ | $\mathbf{0.319}_{\pm 0.001}$ | $0.401_{\pm 0.001}$ | $\mathbf{0.366}_{\pm 0.000}$ |
| | 0.5 | $0.378_{\pm 0.002}$ | $\mathbf{0.360}_{\pm 0.001}$ | $0.442_{\pm 0.001}$ | $\mathbf{0.404}_{\pm 0.001}$ |
| Electricity | 12.5% | $0.092_{\pm 0.001}$ | $\mathbf{0.091}_{\pm 0.000}$ | $0.138_{\pm 0.002}$ | $\mathbf{0.133}_{\pm 0.001}$ |
| | 0.25 | $0.103_{\pm 0.000}$ | $\mathbf{0.102}_{\pm 0.001}$ | $0.156_{\pm 0.000}$ | $\mathbf{0.150}_{\pm 0.000}$ |
| | 0.375 | $\mathbf{0.116}_{\pm 0.002}$ | $0.117_{\pm 0.001}$ | $0.179_{\pm 0.001}$ | $\mathbf{0.173}_{\pm 0.000}$ |
| | 0.5 | $\mathbf{0.132}_{\pm 0.001}$ | $0.133_{\pm 0.001}$ | $0.210_{\pm 0.001}$ | $\mathbf{0.203}_{\pm 0.000}$ |

## G VISUALIZATION

To clearly compare the performance differences before and after applying ITS across different models and datasets, we present the forecasting and imputation results for all models on various benchmark datasets. The visualizations are shown in Fig. 42–58.

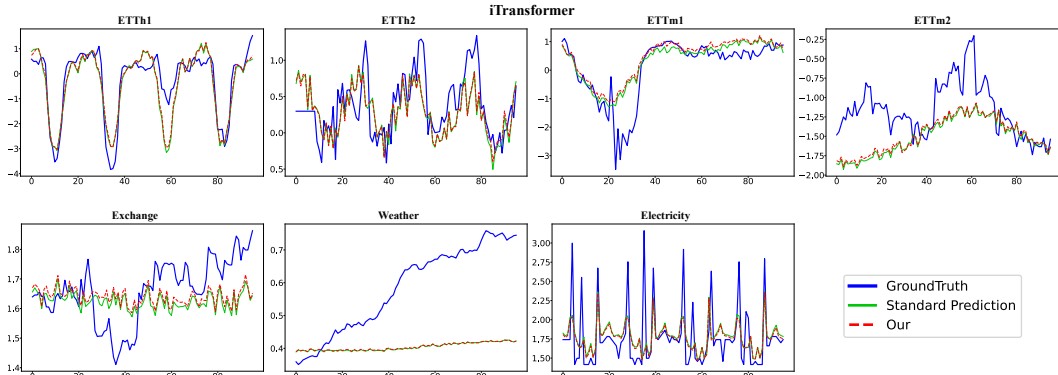

Figure 42: Visualization of input-96-predict-96 results for iTransformer.

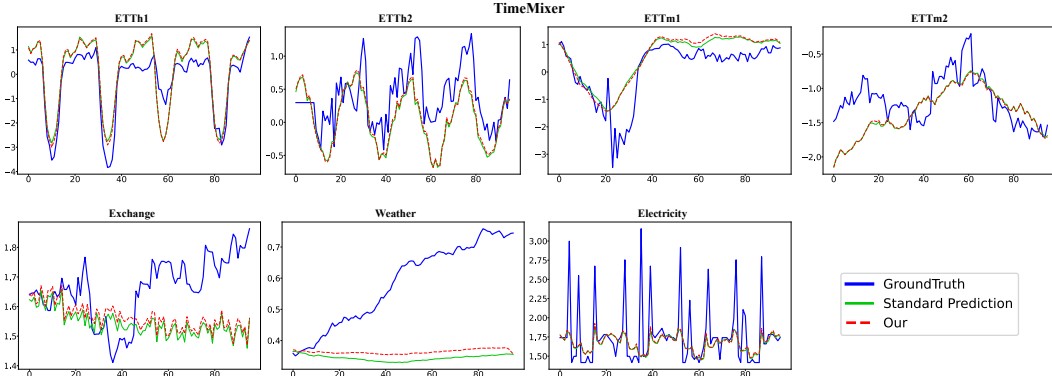

Figure 43: Visualization of input-96-predict-96 results for TimeMixer.

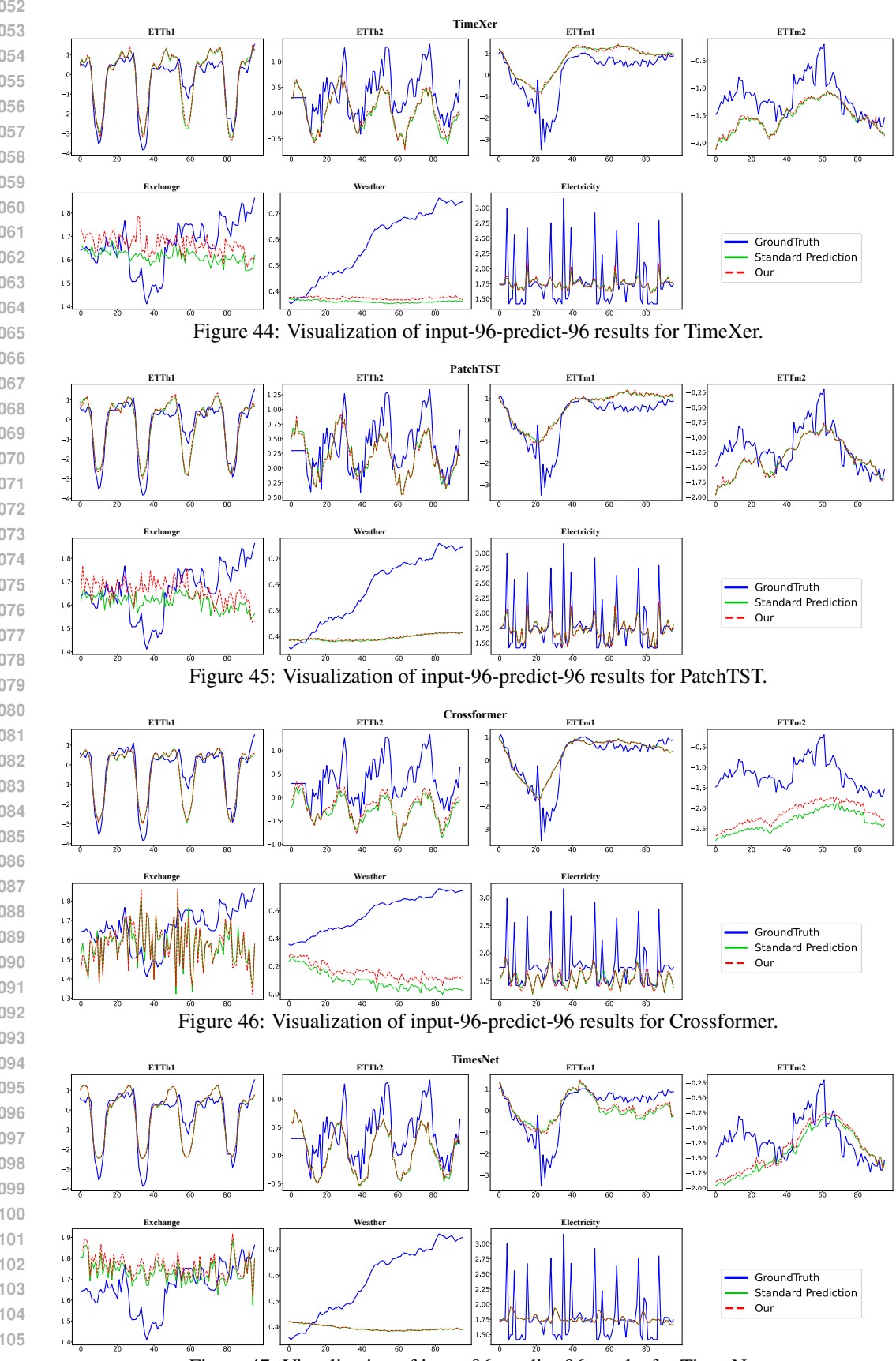

Figure 44: Visualization of input-96-predict-96 results for TimeXer.

Figure 45: Visualization of input-96-predict-96 results for PatchTST.

Figure 46: Visualization of input-96-predict-96 results for Crossformer.

Figure 47: Visualization of input-96-predict-96 results for TimesNet.

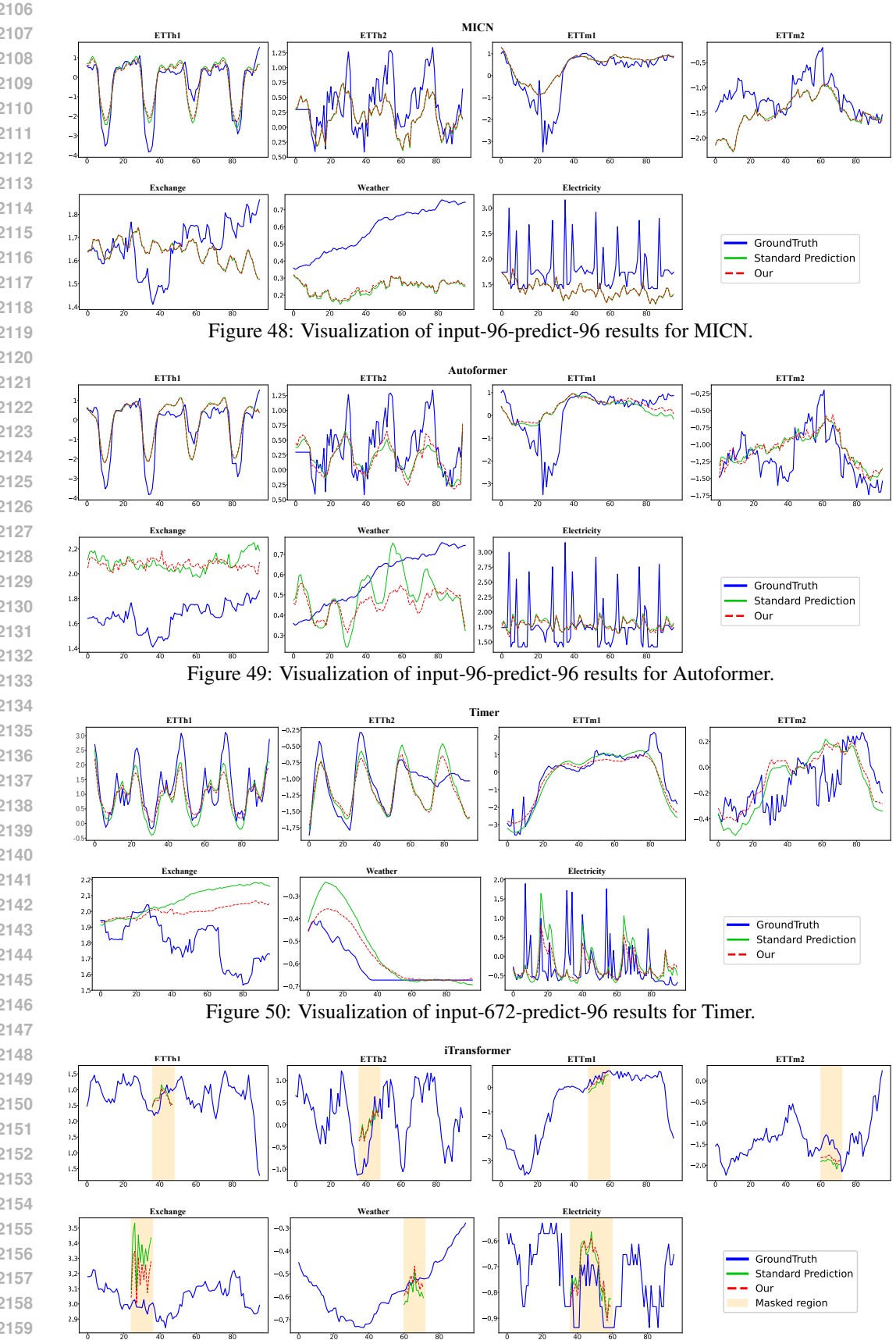

Figure 48: Visualization of input-96-predict-96 results for MICN.

Figure 49: Visualization of input-96-predict-96 results for Autoformer.

Figure 50: Visualization of input-672-predict-96 results for Timer.

Figure 51: Visualization of input-96 and randomly mask 12.5% results for iTransformer.

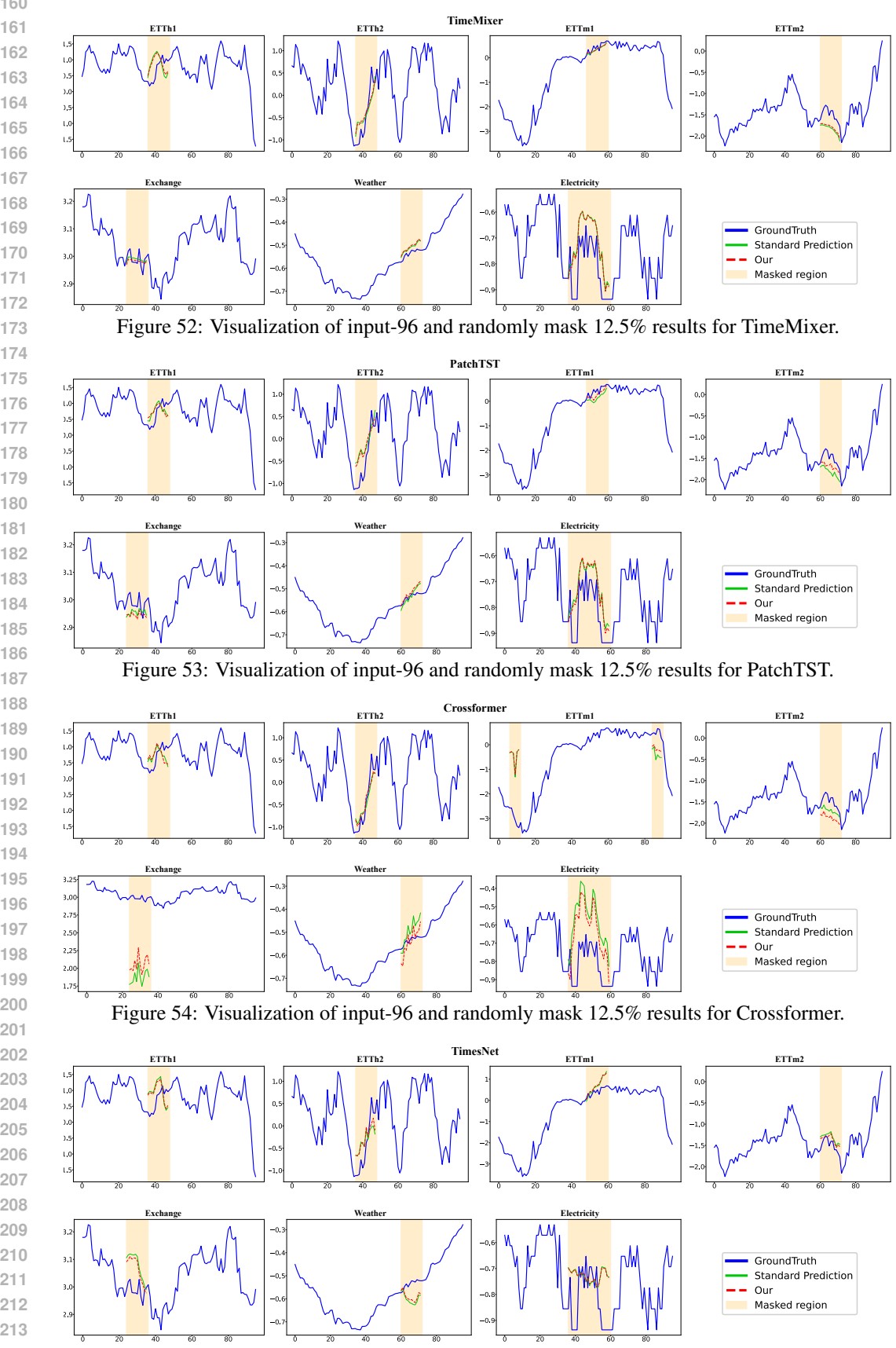

Figure 52: Visualization of input-96 and randomly mask 12.5% results for TimeMixer.

Figure 53: Visualization of input-96 and randomly mask 12.5% results for PatchTST.

Figure 54: Visualization of input-96 and randomly mask 12.5% results for Crossformer.

Figure 55: Visualization of input-96 and randomly mask 12.5% results for TimesNet.

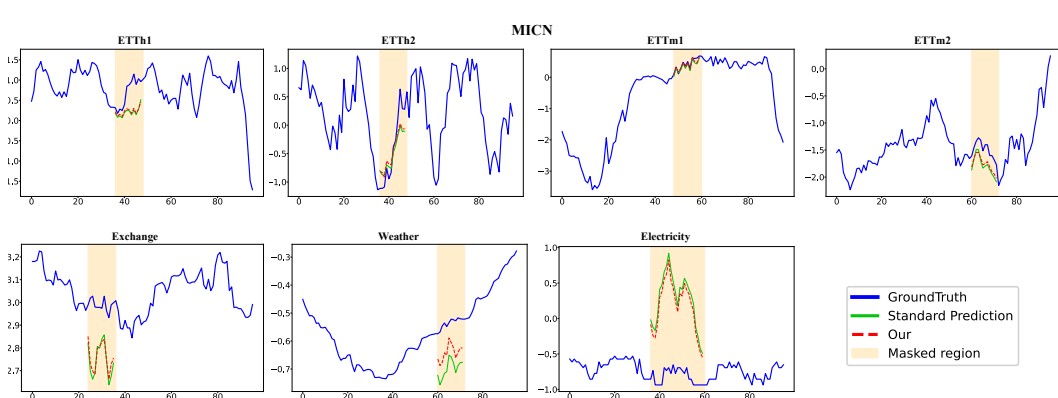

Figure 56: Visualization of input-96 and randomly mask 12.5% results for MICN.

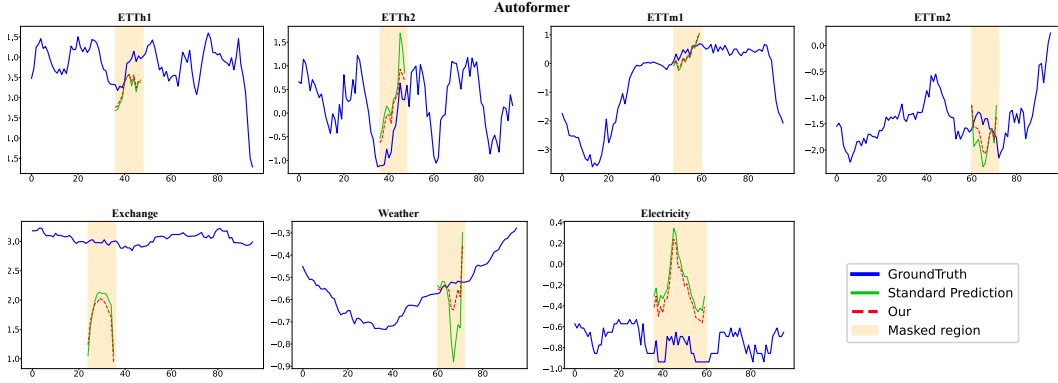

Figure 57: Visualization of input-96 and randomly mask 12.5% results for Autoformer.

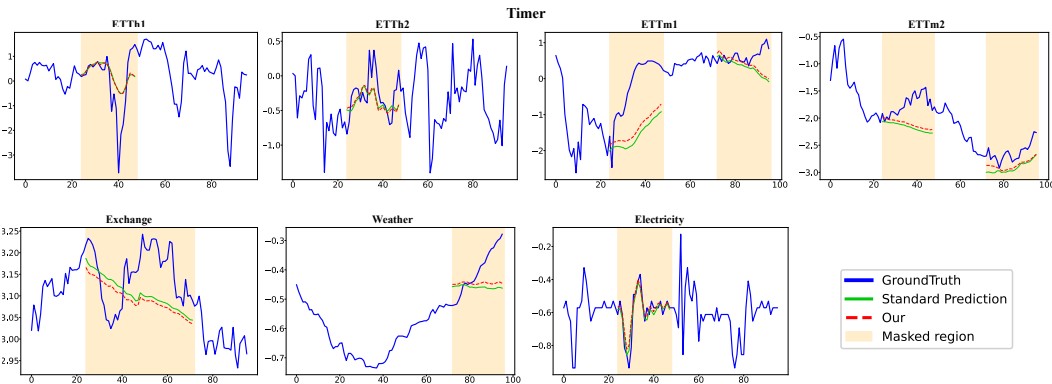

Figure 58: Visualization of input-96 and randomly mask 12.5% results for Timer.

