# OpenReview forum: "Inference-time Scaling for Time-series Processing"
_ICLR.cc/2026/Conference — Submitted to ICLR 2026_

### Official Review · Reviewer_yT6j · 2025-10-15

**Soundness:** 3
**Presentation:** 4
**Contribution:** 3
**Rating:** 6
**Confidence:** 3

**Summary:**

This paper proposes an ensembling method of time series forecasting/imputation models. Basically, the idea is, given a piece of time series to-be-forecasted/imputated and a forecasting model, the authors use MC drop-out to generate multiple forecasting/imputation candidates. After generating, a backward model backcasts the original time series (in forecasting cases); or some original pieces are masked and a backward model backcasts the original time series (in imputation cases). Authors carry out theory analysis based on Bayesian uncertainty theory. Authors also provide experiment results for forecasting and imputation, showing that the proposed method performs better than WTA or average. Higher mse reduction is shown for larger number of candidates.

**Strengths:**

1. The paper studies the interesting idea to use multiple predictions from same model.
2. Authors provide reasonable theory analysis and support for the proposed method.
3. Authors provide comprehensive experiments to validate that using weighted average is better than simple average or best of N.

**Weaknesses:**

1. Could you please provide more ablation study on the choice of $\sigma_e$? Especially, please specify is $\sigma_e$ globally chosen or is it specially tuned for each model each dataset on each prediction horizon/imputation mask percentage. As shown in Figure 4, Figure 16, 17, etc., WTA in some cases would lead to worse performance. Please specify how you choose $\sigma_e$ in more detail, and perhaps carry out sensitivity study of $\sigma_e$.

**Questions:**

Please see weakness. Also questions that are not considered as Cons:

1. The reverse model here is somewhat similar to reward model for LLM. The authors train a backward model with same architecutre/design for each foreward model for consistency, which I think is indeed reasonable. Given that said, according to my understanding and common sense, a stronger reward model would lead to better inference time scaling result for LLM, and perhaps a stronger backward model would lead to better inference time scaling result for time series forecasting. I notice in FIgure 18 and Figure 26 that in the PatchTST as foreward \& backward model case, the WTA and weighted-avg-of-8 lead to worse performance in some cases. Would this be caused by, that the PatchTST backward model is not a nice backward model? Could you use some other backward model with PatchTST foreward model, to see if this would improve PatchTST inference time scaling performance?
2. Following 1., what's your considered best backward model? Would the backward model performance related to the forward forecasting performance? (I know to answer this question in great detail would require a lot of experiments. I'm not requiring the authors to conduct this amount of thorough experiments just to answer this question. Perhaps some intuitive answer would be enough for this question. My main concern is still written in Weakness section.)

One further question for discussion:
Recent observations (e.g. https://www.arxiv.org/abs/2510.02729) show that these time series forecasting benchmarks have almost been saturated. Other concerns come from the fact that it doesn't seem to make sense that one simple neural network can reach sota for all time series tasks without any context (e.g. https://neurips.cc/virtual/2024/108471), so perhaps we should combine more things together (e.g. features, NNs, strategies, etc.). What's your opinion on these thoughts? Compared to proposing methods that risk overfitting these datasets, what future direction do you think would be good for further research of our time series community?

---

> ### Author Response · Authors · 2025-11-21
>
> Dear Reviewer yT6j,
>
> We sincerely appreciate your thorough review and constructive comments. We greatly appreciate your **recognition of the novelty and soundness of our method**, as well as your positive evaluation of the insights contributed by our work.
>
> >**W1-1** Could you please provide more ablation study on the choice of $\sigma_e$? Especially, please specify is $\sigma_e$ globally chosen or is it specially tuned for each model each dataset on each prediction horizon/imputation mask percentage. As shown in Figure 4, Figure 16, 17, etc.,
>
> Thank you very much for raising this question. We apologize for the ambiguity in the original submission that caused some misunderstandings among readers. In fact, **the core scheme used in the main text and Appendix F.1 in the original submission are the weighting based on $\sigma_e$ (the green bars in Fig. 4), and all other weighting strategies serve as comparison baselines. During inference, $\sigma_e$ is a globally fixed choice that does NOT change across models, datasets, forecasting horizons and mask ratios.** It corresponds to the weighting form in Eq. (1) rather than a tunable hyperparameter, and each candidate's weight is entirely determined by its reconstruction error from the backward model.
>
> We have analyzed experimental comparisons with different weighting methods to more clearly demonstrate the effectiveness of the backward model.
> We would like to clarify that **Entropy, Majority, and WTA are ablation baselines obtained by replacing the weighting scheme in ITS, while our proposed method corresponds to the reconstruction-based weighting**:
> * **Ours**: reconstruction–based weighting, $p(y|\boldsymbol{x},D) = \sum_{k}w_kp(y|\boldsymbol{x}, \boldsymbol{W}_k)$, where $w_k$ is a weight derived from the reconstruction error of candidate $k$, which is obtained using the backward model;
> * **Entropy**: entropy-based weighting using the predictive uncertainty of each candidate, $p(y|\boldsymbol{x},D) = \sum_{k}\hat{w}_kp(y|\boldsymbol{x}, \boldsymbol{W}_k)$, where $\hat{w}_k$ is derived from the predictive uncertainty of candidate $k$;
> * **Majority Voting (Majority)**: uniform averaging over all candidates, $p(y|\boldsymbol{x},D) = \frac{1}{K}\sum_{k}p(y|\boldsymbol{x}, \boldsymbol{W}_k)$;
> * **Winners-Take-All (WTA)**: selecting the single candidate with the lowest reconstruction error, $p(y|\boldsymbol{x},D) =   p(y|\boldsymbol{x}, \boldsymbol{\hat{W}^\ast})$, where $\hat{\boldsymbol{W}}^{\ast}$ denotes the parameters of the candidate model with the lowest reconstruction error.
>
> For forecasting:
>
> | Model       | Dataset | Look_back | Pred_len  | Vanilla | Ours  | Entropy| Majority | WTA   |
> |:------------:|:---------:|:-----------:|:------------:|:---------:|:-------:|:----------------:|:---------:|:-------:|
> | iTransformer | ETTh1  | $96$         | $96$         | $0.411$   | $\textbf{0.407}$ | $0.409$          | $0.410$   | $0.423$ |
> |              |        | $96$        | $192$         | $0.462$   | $\textbf{0.459}$ | $0.461$          | $0.461$   | $0.470$ |
> |              |        | $96$        | $336$         | $0.492$   | $\textbf{0.487}$ | $0.488$          | $0.490$   | $0.497$ |
> |              |        | $96$        | $720$         | $0.512$   | $\textbf{0.503}$ | $0.508$          | $0.510$   | $0.506$ |
> |              |Electricity| $96$      | $96$         | $0.150$   | $\textbf{0.145}$ | $0.147$          | $0.149$   | $0.156$ |
> |              |        | $96$        | $192$         | $0.167$   | $\textbf{0.158}$ | $0.164$          | $0.165$   | $0.172$ |
> |              |        | $96$        | $336$         | $0.181$   | $\textbf{0.172}$ | $0.178$          | $0.180$   | $0.188$ |
> |              |        | $96$        | $720$         | $0.222$   | $\textbf{0.210}$ | $0.219$          | $0.222$   | $0.229$ |
> | Crossformer  | ETTh1   | $96$        | $96$         | $0.395$   | $\textbf{0.391}$ | $0.399$          | $0.396$   | $0.412$ |
> |              |         | $96$       | $192$         | $0.628$   | $\textbf{0.614}$ | $0.628$          | $0.631$   | $0.629$ |
> |              |         | $96$       | $336$         | $0.665$   | $\textbf{0.657}$ | $0.673$          | $0.669$   | $0.674$ |
> |              |         | $96$       | $720$         | $0.887$   | $0.825$ | $0.861$          | $0.854$   | $\textbf{0.789}$ |
> |              |Electricity| $96$      | $96$         | $0.219$   | $\textbf{0.218}$ | $0.226$          | $0.219$   | $0.248$ |
> |              |         | $96$       | $192$         | $\textbf{0.237}$   | $0.238$ | $0.239$          | $0.238$   | $0.265$ |
> |              |         | $96$       | $336$         | $0.785$   | $0.778$ | $0.781$          | $0.786$   | $\textbf{0.775}$ |
> |              |         | $96$       | $720$         | $0.813$   | $\textbf{0.810}$ | $0.811$          | $0.813$   | $0.815$ |
>
> [Continued below]

---

> > ### Author Response · Authors · 2025-11-21
> >
> > | Model       | Dataset | Look_back | Pred_len  | Vanilla | Ours  | Entropy| Majority | WTA   |
> > |:------------:|:---------:|:-----------:|:------------:|:---------:|:-------:|:----------------:|:---------:|:-------:|
> > | PatchTST     | ETTh1   | $96$        | $96$         | $\textbf{0.408}$   | $0.409$ | $0.414$          | $0.409$   | $0.451$ |
> > |              |         | $96$       | $192$         | $0.460$   | $\textbf{0.459}$ | $0.469$          | $0.461$   | $0.497$ |
> > |              |         | $96$       | $336$         | $0.494$   | $\textbf{0.492}$ | $0.503$          | $0.495$   | $0.526$ |
> > |              |         | $96$       | $720$         | $0.495$   | $\textbf{0.489}$ | $0.499$          | $0.495$   | $0.525$ |
> > |              |Electricity| $96$      | $96$         | $\textbf{0.193}$   | $0.196$ | $0.199$          | $0.197$   | $0.241$ |
> > |              |         | $96$       | $192$         | $0.199$   | $\textbf{0.193}$ | $0.197$          | $0.203$   | $0.248$ |
> > |              |         | $96$       | $336$         | $0.214$   | $\textbf{0.207}$ | $0.229$          | $0.217$   | $0.264$ |
> > |              |         | $96$       | $720$         | $0.256$   | $\textbf{0.249}$ | $0.251$          | $0.259$   | $0.309$ |
> >
> > For imputation:
> >
> > | Model       | Dataset | Seq len | mask ratio  | Vanilla | Ours  | Entropy| Majority | WTA   |
> > |:------------:|:---------:|:-----------:|:------------:|:---------:|:-------:|:----------------:|:---------:|:-------:|
> > | iTransformer | ETTh1  | $96$        | $0.125$       | $0.262$   | $\textbf{0.255}$ | $0.267$          | $0.269$   | $0.262$ |
> > |              |        | $96$        | $0.25$        | $0.303$   | $\textbf{0.290}$ | $0.304$          | $0.306$   | $0.296$ |
> > |              |        | $96$        | $0.375$       | $0.337$   | $\textbf{0.321}$ | $0.343$          | $0.340$   | $0.323$ |
> > |              |        | $96$        | $0.5$         | $0.379$   | $0.361$ | $0.381$          | $0.383$   | $\textbf{0.358}$ |
> > |              |Electricity| $96$     | $0.125$       | $\textbf{0.092}$   | $0.094$ | $0.093$          | $\textbf{0.092}$   | $0.110$ |
> > |              |        | $96$        | $0.25$        | $\textbf{0.103}$   | $0.104$ | $0.104$          | $\textbf{0.103}$   | $0.123$ |
> > |              |        | $96$        | $0.375$       | $\textbf{0.116}$   | $0.117$ | $0.117$          | $\textbf{0.116}$   | $0.137$ |
> > |              |        | $96$        | $0.5$         | $\textbf{0.132}$   | $0.133$ | $0.134$          | $\textbf{0.132}$   | $0.153$ |
> > | Crossformer  | ETTh1   | $96$       | $0.125$       | $0.300$   | $\textbf{0.285}$ | $0.302$          | $0.305$   | $0.287$ |
> > |              |         | $96$       | $0.25$        | $0.335$   | $\textbf{0.315}$ | $0.335$          | $0.340$   | $0.316$ |
> > |              |         | $96$       | $0.375$       | $0.362$   | $\textbf{0.339}$ | $0.368$          | $0.367$   | $0.342$ |
> > |              |         | $96$       | $0.5$         | $0.398$   | $\textbf{0.373}$ | $0.394$          | $0.403$   | $0.375$ |
> > |              |Electricity| $96$     | $0.125$       | $0.174$   | $\textbf{0.167}$ | $0.174$          | $0.177$   | $0.171$ |
> > |              |         | $96$       | $0.25$        | $0.188$   | $\textbf{0.178}$ | $0.195$          | $0.192$   | $0.182$ |
> > |              |         | $96$       | $0.375$       | $0.206$   | $\textbf{0.195}$ | $0.201$          | $0.210$   | $0.200$ |
> > |              |         | $96$       | $0.5$         | $0.226$   | $\textbf{0.214}$ | $0.223$          | $0.229$   | $0.220$ |
> > | PatchTST     | ETTh1   | $96$       | $0.125$       | $0.320$   | $0.300$ | $0.320$          | $0.324$   | $\textbf{0.293}$ |
> > |              |         | $96$       | $0.25$        | $0.370$   | $0.342$ | $0.368$          | $0.375$   | $\textbf{0.337}$ |
> > |              |         | $96$       | $0.375$       | $0.401$   | $0.368$ | $0.397$          | $0.405$   | $\textbf{0.365}$ |
> > |              |         | $96$       | $0.5$         | $0.444$   | $0.407$ | $0.449$          | $0.448$   | $\textbf{0.399}$ |
> > |              |Electricity| $96$     | $0.125$       | $\textbf{0.138}$   | $\textbf{0.138}$  | $0.142$          | $0.144$   | $0.164$ |
> > |              |         | $96$       | $0.25$        | $0.155$   | $\textbf{0.153}$ | $0.156$          | $0.161$   | $0.179$ |
> > |              |         | $96$       | $0.375$       | $0.180$   | $\textbf{0.176}$ | $0.180$          | $0.186$   | $0.203$ |
> > |              |         | $96$       | $0.5$         | $0.213$   | $\textbf{0.206}$ | $0.214$          | $0.219$   | $0.229$ |
> >
> >
> > To better highlight our contributions, we have expanded and refined the discussion of weighting strategies in Appendix F.1 of the revised version.
> >
> > [Continued below]

---

> ### Author Response · Authors · 2025-11-21
>
> >**W1-2** WTA in some cases would lead to worse performance. Please specify how you choose $\sigma_e$ in more detail, and perhaps carry out sensitivity study of $\sigma_e$.
>
> Thank you for your question. As explained in our response to W1-1, **WTA is used only as a comparison variant within the ITS, where it replaces our reconstruction-based weighting. It serves as an ablation baseline rather than part of our proposed weighting strategy.** Our weighting scheme is globally fixed. The WTA strategy selects the single candidate with the smallest reconstruction error. From its formulation $p(y|\boldsymbol{x},D) = p(y|\boldsymbol{x}, \boldsymbol{\hat{W}^{\ast}}) $, we can see that $\boldsymbol{\hat{W}^{\ast}}$ corresponds to the candidate $\arg\min_{k} E(W_k)$, that is, the one with the minimal reconstruction error. However, even a small amount of noise or estimation bias in $E(W_k)$ can cause WTA to pick a “pseudo-optimal’’ candidate. We also performed a statistical analysis of $\boldsymbol{\hat{W}^*}$ and found that in most cases, relying solely on the backward model’s reconstruction error leads to a rather low probability of selecting the truly optimal candidate.
>
> | Metric                   | Result |
> | :----------------------- |:--------------:|
> | **Total Number of Experiments**  | 100    |
> | **Consistency Count**            | 34     |
>
> >**Q1-1** The reverse model here is somewhat similar to reward model for LLM. The authors train a backward model with same architecutre/design for each foreward model for consistency, which I think is indeed reasonable. Given that said, according to my understanding and common sense, a stronger reward model would lead to better inference time scaling result for LLM, and perhaps a stronger backward model would lead to better inference time scaling result for time series forecasting.
>
> Thank you very much for this insightful question. Our weighting idea shares certain formal similarities with group-wise methods such as GRPO, but it differs in both its goals and its usage. In our framework, **the backward model is designed as a general time-series foundation model**, and in most cases its reconstruction capability is stronger than using the forward model as a backward model. Therefore, **backward model does not change with the choice of the forward model**. We have verified this in Appendix E.2 of the original submission by comparing several backward model.
>
> >**Q1-2** Given that said, according to my understanding and common sense, a stronger reward model would lead to better inference time scaling result for LLM, and perhaps a stronger backward model would lead to better inference time scaling result for time series forecasting.
>
> Yes, you are right! In our framework, the backward model plays an important role as the component that evaluates candidate quality. **Theoretically, the stronger the reconstruction ability of the backward model, the better**:
> * As shown in our theoretical analysis in Sec. 3.4 of the original submission, the model’s output distribution can be linked to its performance through $p(\mathbf{W}\mid D) \sim p(\mathbf{z})\, f(E(\boldsymbol{W}))$, where $E(\boldsymbol{W})$ denotes the model error estimated using the backward model. In particular, $f[E(\mathbf{W})] \sim \frac{1}{\sigma}\exp\left[-\frac{E(\mathbf{W})-E^{\ast}}{\sigma}\right]$ if $E(\boldsymbol{W}) \le E^{\ast}$, and 0 otherwise, with $E^{\ast}$ representing the lowest error achieved on the training corpus $D$. This formulation assigns larger posterior probability mass to models whose errors are closer to $E^{\ast}$. We can denote the reconstruction error for the current input sample as an assessment of the involved model, namely $p(y\mid \mathbf{x},D)=\sum_{k} w_k\, p(y\mid \mathbf{x},\mathbf{W}_k)$.
> * Under this view, when the reconstruction is more accurate, the reconstruction-error signal has higher discriminative power across different candidates, enabling it to more effectively distinguish those that better follow the historical evolution of the time series from those that deviate significantly.
> * At the same time, stronger reconstruction ability reduces the amount of random noise in this error signal, which allows it to exhibit a more stable and exploitable correlation with the true forecasting error, thereby leading to more reliable weighting results overall.
>
> [Continued below]

---

> > ### Author Response · Authors · 2025-11-21
> >
> > >**Q1-3** I notice in FIgure 18 and Figure 26 that in the PatchTST as foreward \& backward model case, the WTA and weighted-avg-of-8 lead to worse performance in some cases. Would this be caused by, that the PatchTST backward model is not a nice backward model? Could you use some other backward model with PatchTST foreward model, to see if this would improve PatchTST inference time scaling performance?
> >
> > Thank you for your sharp observation and question. We humbly respond to your concerns from the following aspects:
> > * As clarified in our response to W1-1, **WTA and Majority are used only as comparison baselines within the ITS framework for ablation purposes. They are NOT part of the default ITS design nor are they methods we propose. The green bars correspond to the weighting scheme (reconstruction-based weighting) we actually adopt**, and Fig. 18 in the original submission shows that this scheme brings performance improvements in most settings.
> > * The WTA strategy always selects the candidate with the lowest reconstruction error. However, as we clarified in our response to W1-2, unavoidable noise or estimation errors in the backward model may cause it to pick a “pseudo-optimal’’ candidate. This issue is also reflected in the experimental results discussed in W1-1. In contrast, the Majority strategy applies simple uniform averaging and does not use the reconstruction signal provided by the backward model at all. Its performance is therefore determined mainly by the forward model itself, failing to reflect the role of the backward model.
> >
> > For this reason, **we consider the PatchTST backward model to be effective**, and the observed performance degradation mainly arises from the weighting strategy employed rather than from any inherent problem with the PatchTST model. In addition, as shown in Fig. 10 in the original submission, **PatchTST backward model still has substantial room for further improvement when compared with stronger backward models**. We have also analyzed an experiment where a single forward model is combined with multiple backward models to assess whether stronger backward models lead to further performance improvements.
> >
> > | model        | dataset | Look back | Pred len | Vanilla  | FT-BM(Ours) | PatchTST | TimeMixer | TimeXer | iTransformer | Crossformer |
> > |:--------------:|:---------:|:-----------:|:----------:|:----------:|:-------:|:--------------:|:-----------:|:---------:|:----------:|:-------------:|
> > | PatchTST     | ETTm2   | $96$      | $96$     | $0.177$  | $\textbf{0.166}$ | $0.176$    | $0.175$   | $0.176$ | $0.174$  | $0.177$     |
> > |              |         |           | $192$    | $0.243$  | $\textbf{0.233}$ | $0.242$    | $0.241$   | $0.244$ | $0.240$  | $0.242$     |
> > |              |         |           | $336$    | $0.305$  | $\textbf{0.293}$ | $0.303$    | $0.306$   | $0.302$ | $0.301$  | $0.303$     |
> > |              |         |           | $720$    | $0.410$  | $\textbf{0.395}$ | $0.406$    | $0.405$   | $0.407$ | $0.406$  | $0.407$     |
> >
> > >**Q2** Following 1., what's your considered best backward model? Would the backward model performance related to the forward forecasting performance? (I know to answer this question in great detail would require a lot of experiments. I'm not requiring the authors to conduct this amount of thorough experiments just to answer this question. Perhaps some intuitive answer would be enough for this question. My main concern is still written in Weakness section.)
> >
> > Thank you for this particularly meaningful question. We humbly respond to your concerns from the following aspects:
> > * We consider that the key to a good backward model is not its ability to reconstruct all candidates back to the same history as closely as possible. Instead, **it should be able to reconstruct different candidates with sufficient discriminative power, ensuring that genuinely more plausible candidates and clearly implausible ones produce clearly distinguishable reconstruction signals.**
> > * We also consider that **the backward model’s performance is typically related to its forward forecasting performance.** Our backward model is still a time-series model. Models that perform better on forward forecasting tasks usually possess stronger representation capacity and more expressive temporal-structure modeling ability. As a result, when trained on the reconstruction task, they are more likely to learn finer-grained and more reliable reconstruction-error signals.
> >
> > [Continued below]

---

> ### Author Response · Authors · 2025-11-21
>
> >**Q3** One further question for discussion: Recent observations (e.g. https://www.arxiv.org/abs/2510.02729) show that these time series forecasting benchmarks have almost been saturated. Other concerns come from the fact that it doesn't seem to make sense that one simple neural network can reach sota for all time series tasks without any context (e.g. https://neurips.cc/virtual/2024/108471), so perhaps we should combine more things together (e.g. features, NNs, strategies, etc.). What's your opinion on these thoughts? Compared to proposing methods that risk overfitting these datasets, what future direction do you think would be good for further research of our time series community?
>
> Thank you very much for your recognition and for giving us the opportunity to continue this discussion. As you pointed out, on the one hand, the widely used time-series forecasting benchmarks are already close to saturation, which makes further improvements increasingly challenging. On the other hand, the fact that relatively simple neural networks can achieve SOTA performance across multiple datasets may suggest that such results are not truly meaningful progress in time-series modeling, and may instead reflect limitations or biases in existing benchmarks.
>
> Against this background, the concern raised by Reviewer wRpv that the gains of our ITS framework appear small on certain backbones is entirely understandable. **When benchmarks are near saturation, further reductions in error become increasingly marginal and less informative about real methodological progress. As a result, raw error metrics alone are no longer sufficient to capture whether a new method offers meaningful advances,** and the community needs to place greater emphasis on robustness, stability, and generalization rather than solely optimizing a single numerical score.
>
> Building on this perspective, we feel that future progress in the time-series community may require moving beyond purely architectural innovations. One promising direction is to integrate domain knowledge into large models so that they can understand the semantics underlying time-series signals, rather than treating them as raw numerical sequences [1-4]. This is also a line of research we are actively pursuing. For example, in health diagnostics, different physiological variables such as ECG, respiration, blood oxygen, and hormonal rhythms carry fundamentally different meanings and interact through latent physiological mechanisms. However, most current time-series models, including strong deep learning methods, treat them as nearly interchangeable numeric channels, leaving much of the semantic structure unexplored. Recent studies have begun to investigate this direction and show that embedding structured domain knowledge such as physiological priors, causal mechanisms, or symbolic medical rules into foundation models can lead to predictions that are more interpretable and more generalizable.
>
> In light of these observations, continued focus on squeezing marginal MSE gains from saturated benchmarks may yield diminishing returns. Instead, it is important to explore how to equip models with richer domain knowledge, and how to enable large models to truly understand time-series rather than merely fit them. **It is also valuable to design post-training and inference-time techniques that enhance generalization beyond specific benchmarks, and to develop evaluation suites that better reflect real-world complexity, heterogeneity, and semantic structure.** We are actively exploring these directions, especially how to incorporate domain knowledge into large models so that they can capture meaningful temporal semantics rather than only numerical patterns, and we hope that this line of work can contribute, in a small but constructive way, to the next stage of time-series research.
>
> [1] Peng H, et al. BearLLM: A Prior Knowledge-Enhanced Bearing Health Management Framework with Unified Vibration Signal Representation. AAAI 2025.\
> [2] Chan N, et al. Medtsllm: Leveraging llms for multimodal medical time series analysis. arXiv preprint arXiv:2408.07773, 2024.\
> [3] Cai Y, et al. Jolt: Jointly learned representations of language and time-series. NeurIPS 2023.\
> [4] Liu L J, et al. KIT-LSTM: knowledge-guided time-aware LSTM for continuous clinical risk prediction. BIBM 2022.
>
> We sincerely appreciate the constructive feedback you have provided and hope that our responses and the modifications to the manuscript adequately address your insightful feedback and increases your impression and confidence in our work.  We are happy to provide any further clarifications if needed.

---

> > ### Comment · Reviewer_yT6j · 2025-11-21
> >
> > I would like to thank the authors for the very detailed and timely reply. I think they have answered my questions clearly.

---

> > > ### Author Response · Authors · 2025-11-22
> > >
> > > Dear Reviewer yT6j,
> > >
> > > We sincerely appreciate the time you devoted to reviewing our work. Your insightful and constructive comments have been invaluable to us!
> > >
> > > We noticed that our score has not changed and are wondering whether there might still be any questions or unresolved concerns that we could help address. If there is anything you would like us to explain in more detail, we would be very happy to provide additional clarification. We genuinely hope that our responses have addressed your concerns and that this may support a more favorable evaluation of our manuscript.
> > >
> > > Your thoughtful feedback has greatly contributed to improving the quality and clarity of our work, and we have fully incorporated your suggestions into the revised version. Once again, thank you very much for your careful review and for the effort you invested in our submission!
> > >
> > > Kind regards,
> > >
> > > Authors

---

> ### Comment · Reviewer_yT6j · 2025-11-22
>
> I would like to thank the authors for their very detailed rebuttal reply. For me, I also agree with you that when benchmarks are near saturation, further reduction in error would be difficult and small margin of improvement could also show effectiveness of your method; I think many papers accepted to top conferences on time series forecasting also shows similar extend of improvement, hence this would be an issue to solve for the whole time series community. Moreover, I indeed think the idea to explore inference-time-scaling is meaningful for time series forecasting tasks. Overall, I am more confident now that your work meets the threshold of acceptance; hence I would like to keep my rating as 6: weak accept while increase my confidence from 3 to 4.

---

> > ### Author Response · Authors · 2025-11-24
> >
> > Dear Reviewer yT6j,
> >
> > Thank you sincerely for your generous and constructive follow-up comments. Your recognition of the ITS framework means a great deal to our team!
> >
> > We also appreciate your decision to maintain a positive rating and to increase your confidence in the evaluation. Your thoughtful feedback has directly helped us improve the clarity and rigor of our work, and we will continue refining the manuscript with your insights in mind.
> >
> > Thank you again for your careful reading, constructive guidance, and open-minded engagement. We deeply value your contribution to strengthening our paper!
> >
> > Kind regards,
> >
> > Authors

---

### Official Review · Reviewer_wRpv · 2025-10-18

**Soundness:** 2
**Presentation:** 2
**Contribution:** 2
**Rating:** 2
**Confidence:** 3

**Summary:**

This paper introduces a general inference-time scaling (ITS) framework for enhancing time-series processing tasks, including forecasting and imputation. The key idea is to enhance model performance at inference time without retraining by generating multiple candidate outputs via Monte Carlo Dropout, evaluating each candidate through reconstruction errors using a backward model, and aggregating them through error-based weighting. The authors justify the approach with a Bayesian uncertainty interpretation, showing that the weighting scheme corresponds to a probabilistic ensemble that reduces epistemic uncertainty. Experiments are performed across seven benchmark datasets and nine models.

**Strengths:**

1. The paper tackles an underexplored direction — inference-time scaling for time-series models — and presents a clear, well-motivated framework. Most current work focuses on squeezing more data into pre-training to obtain better foundation models; shifting attention to smarter inference is an important and complementary perspective. I would, however, have appreciated a deeper discussion comparing ITS to other inference-focused approaches (e.g., meta-learning based ([1], [2]) test-time adaptation or context-based methods [3] / in-context learning).
2. The Bayesian interpretation of the weighting scheme gives theoretical grounding to what would otherwise be a heuristic ensemble technique.
3. The authors validate their procedure across multiple tasks (forecasting and imputation) and over a large set of methods, which supports the claimed generality of ITS.

[1] Learning deep time-index models for time series forecasting, Woo et al, ICML 2023

[2] Time Series Continuous Modeling for Imputation and Forecasting with Implicit Neural Representations, Le Naour et al, TMLR 2024

[3] From Tables to Time: How TabPFN-v2 Outperforms Specialized Time Series Forecasting Models, Hoo et al, 2025

**Weaknesses:**

1. The relative improvements are small. For forecasting, the average relative gain for methods like PatchTST and iTransformer (Table 1 and Appendix F.2) is under 2%, which is modest. Moreover, some experimental choices for forecasting are questionable: using a look-back window of 96 to predict horizons of 96, 192, 336, and 720 is not convincing in my view (longer history windows are typically required for long horizons).
2. For imputation, relative gains are larger, but the experimental design weakens the claim. Most baselines are forecasting models rather than imputation-specific methods; including dedicated imputation baselines (e.g., BRITS, SAITS, CSDI) would be more convincing. Also, the sequences used for imputation are very short (L = 96), which reduces realism and makes it harder to judge real-world efficacy.
3. The computational-cost discussion is underdeveloped and not fully convincing. Section 4.3 and Appendix F.5 only report runtime analyses on two small datasets and for the smallest settings (forecasting input=96 then predict=96, imputation input=96 with 12.5% masking). Because the performance gains are relatively mild, a thorough runtime comparison is essential to justify the extra inference cost. I would expect a comparative table showing end-to-end inference time (in seconds) on a single GPU across multiple datasets and settings.

**Questions:**

Here are my suggestions :

1. Please substantially expand the inference-time computational analysis (Weakness 3). Provide a comparative inference-time table (seconds per example) on the same GPU across representative datasets and multiple settings (varying L and S, and different mask rates). This will clarify the trade-off between accuracy and cost.
2. Strengthen the discussion comparing ITS with other inference-focused approaches, such as meta-learning-based test-time adaptation and context/in-context learning methods (see Strength 1). A short empirical or conceptual comparison would help place ITS in the broader landscape.
3. Improve experimental quality: use longer look-back windows for long-horizon forecasting, include imputation-specific baselines (BRITS, SAITS, CSDI), and evaluate on longer sequences to better reflect practical imputation scenarios.

---

> ### Author Response · Authors · 2025-11-21
>
> Dear Reviewer wRpv,
>
> We sincerely appreciate your thorough review and constructive comments. We are also grateful for your **recognition of our method’s novelty and the soundness of our theoretical analysis**, as well as your positive evaluation of the insights provided by our work.
>
> >**W1.1** The relative improvements are small. For forecasting, the average relative gain for methods like PatchTST and iTransformer (Table 1 and Appendix F.2) is under 2\%, which is modest.
>
> We truly appreciate you raising this concern, and we humbly respond to your concerns from the following aspects:
> * We acknowledge that the relative performance gains of our framework are not substantial for some models. However, **our contribution is a general inference-time scaling framework that is designed to be complementary to existing backbone improvements rather than in competition with them**. As shown in our experiments, **it yields consistent additional gains across almost all models**, each of which was already a SOTA method at the time of its publication. In other words, **the substantial improvements brought by these strong backbones and the further gains from our framework are cumulative** rather than mutually exclusive.
> * As also noted in the discussion raised by Reviewer yT6j, **mainstream time-series baselines are already very strong and many models are close to saturation on multiple benchmarks**, so further improvements become increasingly challenging. **To our knowledge, inference-time scaling in the time-series domain has not been explored before.** We think that obtaining stable and reproducible extra improvements through a plug-and-play inference-time scaling framework without modifying model architectures or retraining remains practically meaningful for time-series tasks.
>
> >**W1.2** Moreover, some experimental choices for forecasting are questionable: using a look-back window of 96 to predict horizons of 96, 192, 336, and 720 is not convincing in my view (longer history windows are typically required for long horizons).
>
> >**Q3-1** Improve experimental quality: use longer look-back windows for long-horizon forecasting,
>
> Thank you for this insightful comment and very helpful reminder. We apologize that our focus on maintaining fairness with baseline settings led us to overlook the completeness of research in the time-series domain. **In the original submission, we strictly followed the look-back window settings used by the corresponding baselines across all datasets. The choice of $L = 96$ is fully consistent with their configurations and can be verified in their original papers [1–8].** We have conducted experiments with extended look-back windows to further validate the stability and effectiveness of ITS under longer history windows. The corresponding results and discussions have been added in Appendix D.3 of the revised version.
>
> | Dataset | Look back | Pred len | iTransformer |                        | PatchTST    |            | Crossformer    |           |
> |:-------:|:--------:|:--------:|:------------:|:---------:|:-----------:|:---------:|:-----------:|:---------:|
> |         |          |             | Vanilla    | +ITS                    | Vanilla    | +ITS                    | Vanilla    | +ITS                    |
> | ETTh1   | $720$     | $96$       | $0.402$    | $\textbf{0.385}$        | $0.384$    | $\textbf{0.383}$        | $0.463$    | $\textbf{0.433}$        |
> |         |           | $192$      | $0.435$    | $\textbf{0.419}$        | $0.426$    | $\textbf{0.412}$        | $0.631$    | $\textbf{0.607}$        |
> |         |           | $336$      | $0.458$    | $\textbf{0.440}$        | $0.467$    | $\textbf{0.453}$        | $0.646$    | $\textbf{0.620}$        |
> |         |           | $720$      | $0.602$    | $\textbf{0.575}$        | $0.512$    | $\textbf{0.497}$        | $1.402$    | $\textbf{1.131}$        |
> | ETTh2   | $720$     | $96$       | $0.316$    | $\textbf{0.307}$        | $0.287$    | $0.287$                 | $0.840$    | $\textbf{0.783}$        |
> |         |           | $192$      | $0.379$    | $\textbf{0.370}$        | $0.365$    | $\textbf{0.353}$        | $1.247$    | $\textbf{1.095}$        |
> |         |           | $336$      | $0.402$    | $\textbf{0.389}$        | $0.385$    | $\textbf{0.373}$        | $1.387$    | $\textbf{1.112}$        |
> |         |           | $720$      | $0.439$    | $\textbf{0.426}$        | $0.424$    | $\textbf{0.415}$        | $1.738$    | $\textbf{1.575}$        |
>
> [Continued below]

---

> > ### Author Response · Authors · 2025-11-21
> >
> > | Dataset | Look back | Pred len | iTransformer |                        | PatchTST    |            | Crossformer    |           |
> > |:-------:|:--------:|:--------:|:------------:|:---------:|:-----------:|:---------:|:-----------:|:---------:|
> > | Weather | $720$     | $96$       | $0.165$    | $\textbf{0.164}$        | $0.146$    | $\textbf{0.137}$        | $0.148$    | $\textbf{0.141}$        |
> > |         |           | $192$      | $\textbf{0.208}$    | $0.209$        | $0.189$    | $\textbf{0.180}$        | $0.212$    | $\textbf{0.204}$        |
> > |         |           | $336$      | $0.264$    | $\textbf{0.260}$        | $0.244$    | $\textbf{0.231}$        | $0.268$    | $\textbf{0.259}$        |
> > |         |           | $720$      | $0.327$    | $\textbf{0.323}$        | $0.316$    | $\textbf{0.302}$        | $0.326$    | $\textbf{0.312}$        |
> > | Electricity| $720$  | $96$       | $0.134$    | $\textbf{0.125}$        | $0.141$    | $\textbf{0.132}$        | $0.194$    | $0.194$                 |
> > |         |           | $192$      | $0.161$    | $\textbf{0.156}$        | $0.157$    | $\textbf{0.148}$        | $0.227$    | $\textbf{0.213}$        |
> > |         |           | $336$      | $0.175$    | $\textbf{0.169}$        | $0.173$    | $\textbf{0.154}$        | $0.775$    | $\textbf{0.650}$        |
> > |         |           | $720$      | $0.192$	| $\textbf{0.184}$        | $0.207$    | $\textbf{0.188}$        | $0.803$    | $\textbf{0.621}$        |
> >
> > [1] Liu Y, et al. iTransformer: Inverted Transformers Are Effective for Time Series Forecasting. ICLR 2024.\
> > [2] Wang S, et al. TimeMixer: Decomposable Multiscale Mixing for Time Series Forecasting. ICLR 2024.\
> > [3] Wang Y, et al. Timexer: Empowering transformers for time series forecasting with exogenous variables. NeurIPS 2024.\
> > [4] Nie Y, et al. A Time Series is Worth 64 Words: Long-term Forecasting with Transformers. ICLR 2023.\
> > [5] Zhang Y, et al. Crossformer: Transformer utilizing cross-dimension dependency for multivariate time series forecasting. ICLR 2023.\
> > [6] Wu, et al. TimesNet: Temporal 2D-Variation Modeling for General Time Series Analysis. ICLR 2023.\
> > [7] Wang, et al. Micn: Multi-scale local and global context modeling for long-term series forecasting. ICLR 2023. \
> > [8] Wu, et al. Autoformer: Decomposition transformers with auto-correlation for long-term series forecasting. NeurIPS 2021.
> >
> > >**W2** For imputation, relative gains are larger, but the experimental design weakens the claim. Most baselines are forecasting models rather than imputation-specific methods; including dedicated imputation baselines (e.g., BRITS, SAITS, CSDI) would be more convincing. Also, the sequences used for imputation are very short (L = 96), which reduces realism and makes it harder to judge real-world efficacy.
> >
> > >**Q3-2** Imputation-specific baselines (BRITS, SAITS, CSDI), and evaluate on longer sequences to better reflect practical imputation scenarios.
> >
> > This is a great point that we had to think seriously about, and we sincerely appreciate your reminder. We have added dedicated imputation baseline models and conducted additional experiments evaluating imputation on longer sequences, and these have been included in Appendix D.4 of the revised version.
> >
> > | Dataset     | Seq len | Mask ratio | iTransformer[1] |            | PatchTST[2]    |            | Crossformer[3] |            |SAITS[4]|        | CSDI[5]    |        | BRITS[6]    |        |
> > |:-----------:|:-------:|:----------:|:------------:|:----------:|:-----------:|:----------:|:-----------:|:----------:|:------:|:------:|:------:|:------:|:------:|:------:|
> > |             |         |            | Vanilla      | +ITS       | Vanilla     | +ITS       | Vanilla     | +ITS       | Vanilla     | +ITS       | Vanilla     | +ITS       | Vanilla     | +ITS       |
> > | ETTh1       | $720$   | $0.125$    | $0.347$      | $\textbf{0.328}$ | $0.294$      | $\textbf{0.277}$ | $0.319$      | $\textbf{0.304}$ | $0.167$   | $\textbf{0.149}$   | $0.352$   | $\textbf{0.327}$   | $0.259$   | $\textbf{0.254}$  |
> > |             |         | $0.25$     | $0.363$      | $\textbf{0.343}$ | $0.304$      | $\textbf{0.288}$ | $0.327$      | $\textbf{0.313}$ | $0.199$   | $\textbf{0.182}$   | $0.360$   | $\textbf{0.341}$   | $0.387$   | $\textbf{0.372}$   |
> > |             |         | $0.375$    | $0.392$      | $\textbf{0.368}$ | $0.318$      | $\textbf{0.302}$ | $0.339$      | $\textbf{0.325}$ | $0.250$   | $\textbf{0.237}$   | $0.447$   | $\textbf{0.424}$   | $0.374$   | $\textbf{0.366}$   |
> > |             |         | $0.5$      | $0.438$      | $\textbf{0.406}$ | $0.333$      | $\textbf{0.319}$ | $0.355$      | $\textbf{0.343}$ | $0.295$   | $\textbf{0.278}$   | $0.375$   | $\textbf{0.354}$   | $0.458$   | $\textbf{0.439}$   |
> >
> > [Continued below]

---

> > > ### Author Response · Authors · 2025-11-21
> > >
> > > | Dataset     | Seq len | Mask ratio | iTransformer[1] |            | PatchTST[2]    |            | Crossformer[3] |            |SAITS[4]|        | CSDI[5]    |        | BRITS[6]    |        |
> > > |:-----------:|:-------:|:----------:|:------------:|:----------:|:-----------:|:----------:|:-----------:|:----------:|:------:|:------:|:------:|:------:|:------:|:------:|
> > > | ETTh2       | $720$   | $0.125$    | $0.305$      | $\textbf{0.254}$ | $0.119$      | $\textbf{0.104}$ | $0.265$      | $\textbf{0.229}$ | $0.362$   | $\textbf{0.322}$   | $0.414$   | $\textbf{0.391}$   | $0.402$   | $\textbf{0.400}$   |
> > > |             |         | $0.25$     | $0.432$      | $\textbf{0.346}$ | $0.128$      | $\textbf{0.113}$ | $0.318$      | $\textbf{0.273}$ | $0.432$   | $\textbf{0.413}$   | $0.399$   | $\textbf{0.382}$   | $0.401$   | $\textbf{0.395}$   |
> > > |             |         | $0.375$    | $0.549$      | $\textbf{0.511}$ | $0.142$      | $\textbf{0.133}$ | $0.385$      | $\textbf{0.328}$ | $0.554$   | $\textbf{0.521}$   | $0.397$   | $\textbf{0.382}$   | $0.436$   | $\textbf{0.427}$   |
> > > |             |         | $0.5$      | $0.638$      | $\textbf{0.552}$ | $0.156$      | $\textbf{0.146}$ | $0.500$      | $\textbf{0.420}$ | $0.635$   | $\textbf{0.574}$   | $0.455$   | $\textbf{0.443}$   | $0.458$   | $\textbf{0.449}$   |
> > > | Weather     | $720$   | $0.125$    | $0.077$      | $\textbf{0.076}$ | $0.068$      | $\textbf{0.067}$ | $0.078$      | $\textbf{0.071}$ | $0.243$   | $\textbf{0.229}$   | $0.272$   | $\textbf{0.267}$   | $\textbf{0.311}$   | $0.314$   |
> > > |             |         | $0.25$     | $0.087$      | $\textbf{0.085}$ | $0.074$      | $\textbf{0.072}$ | $0.076$      | $\textbf{0.073}$ | $0.274$   | $\textbf{0.263}$   | $0.261$   | $\textbf{0.250}$   | $\textbf{0.335}$   | $0.334$   |
> > > |             |         | $0.375$    | $0.093$      | $\textbf{0.092}$ | $0.078$      | $\textbf{0.075}$ | $0.081$      | $\textbf{0.077}$ | $0.331$   | $\textbf{0.320}$   | $0.306$   | $\textbf{0.301}$   | $0.362$   | $0.362$   |
> > > |             |         | $0.5$      | $0.103$      | $\textbf{0.102}$ | $0.087$      | $\textbf{0.086}$ | $0.089$      | $\textbf{0.084}$ | $0.358$   | $\textbf{0.344}$   | $0.327$   | $\textbf{0.322}$   | $0.407$   | $\textbf{0.405}$   |
> > > | Electricity | $720$   | $0.125$    | $\textbf{0.082}$ | $0.084$      | $0.086$      | $\textbf{0.084}$ | $0.120$      | $\textbf{0.104}$ | $0.380$      | $\textbf{0.364}$      | $0.294$    | $\textbf{0.285}$        | $0.381$    | $0.381$                 |
> > > |             |         | $0.25$     | $\textbf{0.091}$ | $0.093$      | $0.095$      | $\textbf{0.091}$ | $0.133$      | $\textbf{0.113}$ | $0.496$      | $\textbf{0.477}$        | $0.312$    | $\textbf{0.303}$        | $0.403$    | $\textbf{0.400}$        |
> > > |             |         | $0.375$    | $0.099$      | $0.099$          | $0.106$      | $\textbf{0.105}$ | $0.151$      | $\textbf{0.140}$ | $0.559$      | $\textbf{0.523}$       | $0.335$    | $\textbf{0.324}$        | $0.442$    | $\textbf{0.441}$        |
> > > |             |         | $0.5$      | $0.107$      | $\textbf{0.105}$ | $0.118$      | $\textbf{0.114}$ | $0.157$      | $\textbf{0.151}$ | $0.725$      | $\textbf{0.675}$        | $0.406$    | $\textbf{0.396}$        | $0.506$    | $\textbf{0.502}$        |
> > >
> > > [1] Liu Y, et al. iTransformer: Inverted Transformers Are Effective for Time Series Forecasting. ICLR 2024.\
> > > [2] Nie Y, et al. A Time Series is Worth 64 Words: Long-term Forecasting with Transformers. ICLR 2023.\
> > > [3] Zhang Y, et al. Crossformer: Transformer utilizing cross-dimension dependency for multivariate time series forecasting. ICLR 2023.\
> > > [4] Du W, et al. Saits: Self-attention-based imputation for time series. Expert Systems with Applications 2023.\
> > > [5] Tashiro Y, et al. Csdi: Conditional score-based diffusion models for probabilistic time series imputation. NeurIPS 2021.\
> > > [6] Cao W, et al. Brits: Bidirectional recurrent imputation for time series. NeurIPS  2018.
> > >
> > > [Continued below]

---

> > > > ### Author Response · Authors · 2025-11-21
> > > >
> > > > >**W3** The computational-cost discussion is underdeveloped and not fully convincing. Section 4.3 and Appendix F.5 only report runtime analyses on two small datasets and for the smallest settings (forecasting input=96 then predict=96, imputation input=96 with 12.5\% masking). Because the performance gains are relatively mild, a thorough runtime comparison is essential to justify the extra inference cost. I would expect a comparative table showing end-to-end inference time (in seconds) on a single GPU across multiple datasets and settings.
> > > >
> > > > >**Q1** Please substantially expand the inference-time computational analysis (Weakness 3). Provide a comparative inference-time table (seconds per example) on the same GPU across representative datasets and multiple settings (varying L and S, and different mask rates). This will clarify the trade-off between accuracy and cost.
> > > >
> > > > Thank you very much for this thoughtful suggestion. As these two questions are similar, they are addressed together. We have added a detailed table comparing inference time on the same GPU across representative datasets and multiple settings, and these have been added to Appendix F.5 of the revised version.
> > > >
> > > >
> > > > | Dataset     | Model        | L      | S      | Vanilla (ms / example) | +ITS (ms / example) |
> > > > |:-----------:|:------------:|:------:|:------:|:-----------------------:|:--------------------:|
> > > > | ETTh2       | iTransformer | $96$   | $96$   | $0.094$                 | $0.099$              |
> > > > |             |              |        | $192$  | $0.128$                 | $0.193$              |
> > > > |             |              |        | $336$  | $0.143$                 | $0.283$              |
> > > > |             |              |        | $720$  | $0.184$                 | $0.423$              |
> > > > |             | PatchTST     | $96$   | $96$   | $0.112$                 | $0.351$              |
> > > > |             |              |        | $192$  | $0.124$                 | $0.403$              |
> > > > |             |              |        | $336$  | $0.133$                 | $0.534$              |
> > > > |             |              |        | $720$  | $0.221$                 | $0.948$              |
> > > > | Weather     | iTransformer | $96$   | $96$   | $0.257$                 | $0.598$              |
> > > > |             |              |        | $192$  | $0.271$                 | $0.626$              |
> > > > |             |              |        | $336$  | $0.313$                 | $0.892$              |
> > > > |             |              |        | $720$  | $0.374$                 | $0.998$              |
> > > > |             | PatchTST     | $96$   | $96$   | $0.112$                 | $0.684$              |
> > > > |             |              |        | $192$  | $0.128$                 | $0.898$              |
> > > > |             |              |        | $336$  | $0.165$                 | $1.235$              |
> > > > |             |              |        | $720$  | $0.222$                 | $1.753$              |
> > > > | Electricity | iTransformer | $96$   | $96$   | $0.482$                 | $2.882$              |
> > > > |             |              |        | $192$  | $0.602$                 | $3.654$              |
> > > > |             |              |        | $336$  | $0.705$                 | $4.490$              |
> > > > |             |              |        | $720$  | $1.774$                 | $11.125$             |
> > > > |             | PatchTST     | $96$   | $96$   | $0.838$                 | $8.472$              |
> > > > |             |              |        | $192$  | $0.911$                 | $9.315$              |
> > > > |             |              |        | $336$  | $0.944$                 | $11.987$             |
> > > > |             |              |        | $720$  | $2.081$                 | $22.150$             |
> > > >
> > > > | Dataset     | Model        | L      | Mask ratio | Vanilla (ms / example) | +ITS (ms / example) |
> > > > |:-----------:|:------------:|:------:|:----------:|:-----------------------:|:--------------------:|
> > > > | ETTh2       | iTransformer | $96$   | $0.125$    | $0.107$                 | $0.546$              |
> > > > |             |              |        | $0.25$     | $0.158$                 | $0.780$              |
> > > > |             |              |        | $0.375$    | $0.161$                 | $0.813$              |
> > > > |             |              |        | $0.5$      | $0.169$                 | $0.846$              |
> > > > |             | PatchTST     | $96$   | $0.125$    | $0.119$                 | $0.489$              |
> > > > |             |              |        | $0.25$     | $0.108$                 | $0.446$              |
> > > > |             |              |        | $0.375$    | $0.089$                 | $0.410$              |
> > > > |             |              |        | $0.5$      | $0.106$                 | $0.454$              |
> > > > | Weather     | iTransformer | $96$   | $0.125$    | $0.117$                 | $1.145$              |
> > > > |             |              |        | $0.25$     | $0.111$                 | $1.139$              |
> > > >
> > > > [Continued below]

---

> > > > > ### Author Response · Authors · 2025-11-21
> > > > >
> > > > > | Dataset     | Model        | L      | Mask ratio | Vanilla (ms / example) | +ITS (ms / example) |
> > > > > |:-----------:|:------------:|:------:|:----------:|:-----------------------:|:--------------------:|
> > > > > |             |              |        | $0.375$    | $0.112$                 | $1.180$              |
> > > > > |             |              |        | $0.5$      | $0.107$                 | $1.088$              |
> > > > > |             | PatchTST     | $96$   | $0.125$    | $0.218$                 | $2.443$              |
> > > > > |             |              |        | $0.25$     | $0.182$                 | $1.936$              |
> > > > > |             |              |        | $0.375$    | $0.223$                 | $2.457$              |
> > > > > |             |              |        | $0.5$      | $0.191$                 | $2.181$              |
> > > > > | Electricity | iTransformer | $96$   | $0.125$    | $0.495$                 | $7.033$              |
> > > > > |             |              |        | $0.25$     | $0.431$                 | $6.331$              |
> > > > > |             |              |        | $0.375$    | $0.449$                 | $6.752$              |
> > > > > |             |              |        | $0.5$      | $0.433$                 | $6.352$              |
> > > > > |             | PatchTST     | $96$   | $0.125$    | $1.143$                 | $15.019$             |
> > > > > |             |              |        | $0.25$     | $1.388$                 | $19.196$             |
> > > > > |             |              |        | $0.375$    | $1.451$                 | $20.357$             |
> > > > > |             |              |        | $0.5$      | $1.391$                 | $19.014$             |
> > > > >
> > > > > >**Q2**.Strengthen the discussion comparing ITS with other inference-focused approaches, such as meta-learning-based test-time adaptation and context/in-context learning methods (see Strength 1). A short empirical or conceptual comparison would help place ITS in the broader landscape.
> > > > >
> > > > > We sincerely appreciate this highly valuable suggestion, which indeed helps clarify the positioning and contribution of our work. We conceptually compare our proposed ITS with several representative inference-focused approaches:
> > > > > * **Test-time adaptation (TTA)** refers to dynamic optimization at inference time, where model parameters are slightly updated to gradually align the model with the test-time distribution. Woo et al. [1] mitigate long-horizon distribution shift by updating time-index representations online, while Le Naour et al. [2] optimize latent codes in an implicit neural representation framework to better capture local dynamics during test time.
> > > > > * **Test-time training (TTT)** similarly introduces self-supervised auxiliary objectives during inference and jointly optimizes the model without requiring future labels, enabling better adaptation to the current input. For instance, Christou et al.[3] employ test-time training modules to adjust weights during inference, thereby improving adaptivity and dependency modeling.
> > > > > * **In-context learning (ICL)** leverages a model’s pretrained contextual learning capabilities by concatenating additional examples, instructions, or task descriptions to the input, enabling rapid adaptation without any gradient updates. A representative example is Hoo et al. [4], who reformulate time-series forecasting as a tabular regression task and use TabPFN’s attention mechanisms to condition on historical observations as context for future prediction.
> > > > > * **Inference-time scaling (ITS)** neither updates model parameters at test time nor relies on constructing additional contextual sequences. With the model kept fully frozen, ITS allocates extra compute at inference time to generate a diverse set of candidate predictions, and then evaluates their temporal consistency with historical observations to reweight these candidates. In this way, it provides a plug-and-play mechanism that systematically improves the performance of existing time-series models.
> > > > >
> > > > > We have added Appendix A.3 in the revised version to discuss our ITS comparisons with other inference-focused approaches.
> > > > >
> > > > > [1] Woo et al, Learning deep time-index models for time series forecasting. ICML 2023\
> > > > > [2] Le Naour et al. Time Series Continuous Modeling for Imputation and Forecasting with Implicit Neural Representations. TMLR 2024.\
> > > > > [3] Christou P, et al. Test time learning for time series forecasting. arXiv preprint arXiv:2409.14012, 2024.\
> > > > > [4] Hoo et al, From Tables to Time: How TabPFN-v2 Outperforms Specialized Time Series Forecasting Models. arXiv preprint arXiv:2501.02945, 2025.
> > > > >
> > > > > We sincerely appreciate the constructive feedback you have provided and hope that our responses and the modifications to the manuscript adequately address your insightful feedback and increases your impression and confidence in our work.  We are happy to provide any further clarifications if needed.

---

> > > > > > ### Comment · Reviewer_wRpv · 2025-11-23
> > > > > > **Answer**
> > > > > >
> > > > > > I would like to thank the authors for their thorough rebuttal. They answered all of my questions carefully.
> > > > > >
> > > > > > I agree that these benchmarks are close to saturation for univariate modeling without covariates. However, when you consider the look-back windows in forecasting, as I suggested, the performance gain is relatively consistent and significant. If I'm reading the table correctly, it's sometimes more significant than the ITS method. But in all the settings, the ITS method appears to improve the results metrics.
> > > > > >
> > > > > > Honestly, I don't understand why SAITS performed so well for ETTh1 and poorly for the other datasets in the new imputation experiments. SAITS is supposed to be state-of-the-art (or nearly so) in imputation. For example, the electricity dataset is supposed to be easier than the ETTh1 dataset (from my experience and the literature).
> > > > > >
> > > > > > In conclusion, I will raise my score because the authors carefully answered all my questions and were convincing for most of them. However, since the gains are not usually substantial and I am surprised by the results of some experiments, I cannot raise my score any higher.

---

> ### Author Response · Authors · 2025-11-24
>
> Dear Reviewer wRpv,
>
> Thank you very much for your prompt reply! We are glad to hear that our responses have addressed most of your concerns and have contributed to a more positive evaluation. We sincerely appreciate all the valuable comments and constructive suggestions you provided—these have greatly improved the quality of our paper, especially in terms of completeness and rigor in experimental evaluation.
>
> We humbly respond to your questions from the following aspects:
> * Your interpretation of the table is correct. However, we would like to note that many prior studies have suggested that longer look-back windows do not necessarily lead to stable or significant performance improvements [1–5]. In our supplementary experiments, iTransformer and PatchTST include architectural components that explicitly alleviate degradation under long look-back windows, whereas Crossformer does not, and the observed behaviors are consistent with these design differences. Therefore, we consider the conclusion that “longer look-back windows yield relatively consistent and significant gains” does not hold universally. While we do not deny that a longer look-back window may sometimes produce improvements that are even larger than those brought by ITS, we argue that ITS provides consistent gains across different look-back lengths, and this robustness is an important strength of our method.
> * SAITS is indeed a strong imputation model, and this is beyond question. However, it is important to note that it is not consistently superior in all scenarios [6–10]. Our supplementary experiments were conducted under a challenging setting: long imputation sequences of L = 720 and missing subsequences of length N = 12. This difficulty level is significantly higher than the setups used in the original SAITS paper and in most existing studies [11, 12]. We are currently running additional experiments with settings $L \in (24, 96, 192, 336)$, using either point-wise missingness or missing subsequences of length N = 12, to investigate whether the performance discrepancies arise from the difficulty of our evaluation setup.  These experiments will take about two days to complete, and we will update the results as soon as they become available.
>
> Thank you again for your careful reading, constructive guidance, and open-minded engagement. We deeply value your contribution to strengthening our paper!
>
> Kind regards,
>
> Authors
>
> [1] Zeng A, et al. Are transformers effective for time series forecasting?. AAAI 2023.\
> [2] Nie Y, et al. A Time Series is Worth 64 Words: Long-term Forecasting with Transformers. ICLR 2023.\
> [3] Wen Q, et al. Transformers in time series: a survey. IJCAI. 2023.\
> [4] Liu Y, et al. iTransformer: Inverted Transformers Are Effective for Time Series Forecasting. ICLR 2024.\
> [5] Kim D, et al. Are self-attentions effective for time series forecasting?. NeuIPS 2024.\
> [6] Yang J, et al. Revisiting multivariate time series forecasting with missing values. arXiv preprint arXiv:2509.23494, 2025.\
> [7] Du W, et al. Tsi-bench: Benchmarking time series imputation. arXiv preprint arXiv:2406.12747, 2024.\
> [8] Jing P, et al. S4M: S4 for multivariate time series forecasting with Missing values. ICLR 2025.\
> [9] Cheng J, et al. NuwaTS: a Foundation Model Mending Every Incomplete Time Series. arXiv preprint arXiv:2405.15317, 2024.\
> [10] Zhou J, et al. Mtsci: A conditional diffusion model for multivariate time series consistent imputation. CIKM 2024.\
> [11] Liu Y, et al. Timer: generative pre-trained transformers are large time series models. ICML 2024.\
> [12] Du W, et al. Saits: Self-attention-based imputation for time series. Expert Systems with Applications 2023.

---

> ### Author Response · Authors · 2025-11-27
>
> Dear Reviewer wRpv,
>
> Thank you again for your valuable comments and constructive suggestions. We provide here the imputation results under different settings (where $L$ denotes the sequence length and $N$ denotes the length of the missing subsequence). Our conclusion is as follows: we found that **the unexpected behavior observed in our previous supplementary experiments was primarily caused by the value of $N$, rather than any issue in our experimental results**. By comparing eight different $(L, N)$ configurations (as shown in the table below), we observed that the overall sequence length $L$ does not play a dominant role in determining model performance. Instead, **the value of $N$ is the key factor**. When $N = 1$, corresponding to the standard **point-missing scenario, SAITS performs very strongly and BRITS also shows solid performance**. However, once point missingness is extended to longer **subsequence missingness ($N=12$), both SAITS and BRITS experience significant performance degradation, and we have also observed similar phenomena in some outstanding prior works[1]**. We consider that this phenomenon arises from the design philosophy and typical application scenarios of SAITS and BRITS, which are not fully aligned with the **more realistic, real-world evaluation** setting used in our study.
>
> We hope that the above additional explanations help address your concerns more thoroughly. If our responses have given you a more positive overall impression of our work, we would kindly appreciate it if you could consider raising the score again.
>
> Kind regards,
>
> Authors
>
> [1] Naour E L, et al. Are Time-Indexed Foundation Models the Future of Time Series Imputation?[J]. arXiv preprint arXiv:2511.05980, 2025.
>
> ***
> $L=24$, $N=1$
>
> | Dataset     | Mask ratio | iTransformer |            | PatchTST    |            |SAITS|        | BRITS    |        |
> |:-----------:|:----------:|:------------:|:----------:|:-----------:|:----------:|:------:|:------:|:------:|:------:|
> |             |            | Vanilla      | +ITS       | Vanilla     | +ITS       | Vanilla     | +ITS       | Vanilla     | +ITS       |
> | ETTh2       | $0.125$    | $0.121$      | $\textbf{0.103}$      | $0.073$      | $\textbf{0.069}$ | $0.065$      | $\textbf{0.054}$      | $0.143$    | $\textbf{0.130}$   |
> |             | $0.25$     | $0.152$      | $\textbf{0.130}$      | $0.074$      | $\textbf{0.063}$ | $0.068$      | $\textbf{0.061}$      | $0.145$    | $\textbf{0.135}$   |
> |             | $0.375$    | $0.184$      | $\textbf{0.159}$      | $0.076$      | $\textbf{0.068}$ | $0.080$      | $\textbf{0.072}$      | $0.161$    | $\textbf{0.147}$   |
> |             | $0.5$      | $0.214$      | $\textbf{0.188}$      | $0.082$      | $\textbf{0.074}$ | $0.098$      | $\textbf{0.087}$      | $0.253$    | $\textbf{0.239}$   |
> | Electricity | $0.125$    | $0.072$      | $\textbf{0.064}$      | $0.104$      | $\textbf{0.094}$ | $0.068$      | $\textbf{0.064}$      | $0.122$    | $\textbf{0.113}$   |
> |             | $0.25$     | $0.082$      | $\textbf{0.073}$      | $0.124$      | $\textbf{0.113}$ | $0.092$      | $\textbf{0.089}$      | $0.121$    | $\textbf{0.116}$        |
> |             | $0.375$    | $0.099$      | $\textbf{0.089}$      | $0.179$      | $\textbf{0.160}$ | $0.093$      | $\textbf{0.088}$      | $0.141$    | $\textbf{0.131}$        |
> |             | $0.5$      | $0.112$      | $\textbf{0.105}$      | $0.118$      | $\textbf{0.111}$ | $0.115$      | $\textbf{0.112}$      | $0.158$    | $\textbf{0.152}$        |

---

> > ### Author Response · Authors · 2025-11-27
> >
> > $L=24$, $N=12$
> >
> > | Dataset     | Mask ratio | iTransformer |            | PatchTST    |            |SAITS|        | BRITS    |        |
> > |:-----------:|:----------:|:------------:|:----------:|:-----------:|:----------:|:------:|:------:|:------:|:------:|
> > |             |            | Vanilla      | +ITS       | Vanilla     | +ITS       | Vanilla     | +ITS       | Vanilla     | +ITS       |
> > | ETTh2       | $0.125$    | $0.191$      | $\textbf{0.184}$      | $0.191$      | $\textbf{0.189}$ | $0.342$   | $\textbf{0.322}$   | $0.886$   | $\textbf{0.804}$   |
> > |             | $0.25$     | $0.195$      | $\textbf{0.186}$      | $0.192$      | $\textbf{0.183}$ | $0.295$   | $\textbf{0.283}$   | $0.842$   | $\textbf{0.795}$   |
> > |             | $0.375$    | $0.190$      | $\textbf{0.181}$      | $0.187$      | $\textbf{0.178}$ | $0.299$   | $\textbf{0.291}$   | $0.772$   | $\textbf{0.727}$   |
> > |             | $0.5$      | $0.189$      | $\textbf{0.182}$      | $0.186$      | $\textbf{0.180}$ | $0.310$   | $\textbf{0.304}$   | $0.713$   | $\textbf{0.649}$   |
> > | Electricity | $0.125$    | $0.282$      | $\textbf{0.274}$      | $0.588$      | $\textbf{0.544}$ | $0.405$   | $\textbf{0.364}$      | $0.334$    | $\textbf{0.321}$                 |
> > |             | $0.25$     | $0.280$      | $\textbf{0.273}$      | $0.588$      | $\textbf{0.553}$ | $0.404$   | $\textbf{0.377}$      | $0.336$    | $\textbf{0.320}$        |
> > |             | $0.375$    | $\textbf{0.278}$      | $0.279$      | $0.587$      | $\textbf{0.540}$ | $0.404$   | $\textbf{0.383}$      | $0.334$    | $\textbf{0.321}$        |
> > |             | $0.5$      | $0.277$      | $\textbf{0.275}$      | $0.583$      | $\textbf{0.571}$ | $0.403$   | $\textbf{0.375}$      | $0.337$    | $\textbf{0.332}$        |
> >
> > $L=96$, $N=1$
> >
> > | Dataset     | Mask ratio | iTransformer |            | PatchTST    |            |SAITS|        | BRITS    |        |
> > |:-----------:|:----------:|:------------:|:----------:|:-----------:|:----------:|:------:|:------:|:------:|:------:|
> > |             |            | Vanilla      | +ITS       | Vanilla     | +ITS       | Vanilla     | +ITS       | Vanilla     | +ITS       |
> > | ETTh2       | $0.125$    | $0.184$      | $\textbf{0.174}$ | $0.079$      | $\textbf{0.069}$ | $0.076$   | $\textbf{0.066}$   | $0.124$   | $\textbf{0.120}$   |
> > |             | $0.25$     | $0.267$      | $\textbf{0.246}$ | $0.080$      | $\textbf{0.073}$ | $0.075$   | $\textbf{0.068}$   | $0.158$   | $\textbf{0.155}$   |
> > |             | $0.375$    | $0.343$      | $\textbf{0.331}$ | $0.082$      | $\textbf{0.068}$ | $0.085$   | $\textbf{0.075}$   | $0.136$   | $\textbf{0.127}$   |
> > |             | $0.5$      | $0.308$      | $\textbf{0.302}$ | $0.088$      | $\textbf{0.080}$ | $0.107$   | $\textbf{0.094}$   | $0.277$   | $\textbf{0.269}$   |
> > | Electricity | $0.125$    | $0.064$      | $\textbf{0.063}$ | $0.073$      | $\textbf{0.064}$ | $0.061$      | $\textbf{0.054}$      | $0.134$    | $\textbf{0.126}$        |
> > |             | $0.25$     | $\textbf{0.078}$      | $0.079$ | $0.087$      | $\textbf{0.083}$ | $0.070$      | $\textbf{0.067}$      | $0.119$    | $\textbf{0.110}$        |
> > |             | $0.375$    | $0.088$      | $0.088$          | $0.099$      | $\textbf{0.090}$ | $0.079$      | $\textbf{0.073}$      | $0.145$    | $\textbf{0.131}$        |
> > |             | $0.5$      | $0.100$      | $\textbf{0.098}$ | $0.115$      | $\textbf{0.101}$ | $0.105$      | $\textbf{0.095}$      | $0.151$    | $\textbf{0.142}$        |

---

> > > ### Author Response · Authors · 2025-11-27
> > >
> > > $L=96$, $N=12$
> > >
> > > | Dataset     | Mask ratio | iTransformer |            | PatchTST    |            |SAITS|        | BRITS    |        |
> > > |:-----------:|:----------:|:------------:|:----------:|:-----------:|:----------:|:------:|:------:|:------:|:------:|
> > > |             |            | Vanilla      | +ITS       | Vanilla     | +ITS       | Vanilla     | +ITS       | Vanilla     | +ITS       |
> > > | ETTh2       | $0.125$    | $0.230$      | $\textbf{0.185}$ | $0.134$      | $\textbf{0.124}$ | $0.301$   | $\textbf{0.282}$   | $0.996$   | $\textbf{0.862}$   |
> > > |             | $0.25$     | $0.292$      | $\textbf{0.231}$ | $0.142$      | $\textbf{0.133}$ | $0.323$   | $\textbf{0.313}$   | $0.407$   | $\textbf{0.375}$   |
> > > |             | $0.375$    | $0.347$      | $\textbf{0.276}$ | $0.155$      | $\textbf{0.145}$ | $0.366$   | $\textbf{0.341}$   | $0.754$   | $\textbf{0.687}$   |
> > > |             | $0.5$      | $0.384$      | $\textbf{0.313}$ | $0.169$      | $\textbf{0.160}$ | $0.423$   | $\textbf{0.394}$   | $1.263$   | $\textbf{1.149}$   |
> > > | Electricity | $0.125$    | $\textbf{0.092}$      | $0.094$      | $0.127$      | $\textbf{0.104}$ | $0.198$      | $\textbf{0.164}$      | $\textbf{0.230}$    | $0.236$                 |
> > > |             | $0.25$     | $\textbf{0.103}$      | $0.104$      | $0.144$      | $\textbf{0.113}$ | $0.203$      | $\textbf{0.177}$      | $0.230$    | $\textbf{0.220}$        |
> > > |             | $0.375$    | $\textbf{0.116}$      | $0.117$          | $0.167$      | $\textbf{0.140}$ | $0.208$      | $\textbf{0.183}$      | $0.235$    | $\textbf{0.221}$        |
> > > |             | $0.5$      | $\textbf{0.132}$      | $0.133$ | $0.202$      | $\textbf{0.151}$ | $0.212$      | $\textbf{0.195}$      | $0.236$    | $\textbf{0.222}$        |
> > >
> > > $L=192$, $N=1$
> > >
> > > | Dataset     | Mask ratio | iTransformer |            | PatchTST    |            |SAITS|        | BRITS    |        |
> > > |:-----------:|:----------:|:------------:|:----------:|:-----------:|:----------:|:------:|:------:|:------:|:------:|
> > > |             |            | Vanilla      | +ITS       | Vanilla     | +ITS       | Vanilla     | +ITS       | Vanilla     | +ITS       |
> > > | ETTh2       | $0.125$    | $0.224$      | $\textbf{0.204}$ | $0.089$      | $\textbf{0.079}$ | $0.084$   | $\textbf{0.082}$   | $0.086$   | $\textbf{0.080}$   |
> > > |             | $0.25$     | $0.359$      | $\textbf{0.336}$ | $0.090$      | $\textbf{0.083}$ | $\textbf{0.082}$   | $0.083$   | $\textbf{0.114}$   | $0.115$   |
> > > |             | $0.375$    | $0.496$      | $\textbf{0.481}$ | $0.092$      | $\textbf{0.088}$ | $0.096$   | $\textbf{0.091}$   | $0.157$   | $\textbf{0.147}$   |
> > > |             | $0.5$      | $0.447$      | $\textbf{0.432}$ | $0.096$      | $\textbf{0.090}$ | $0.119$   | $\textbf{0.114}$   | $0.198$   | $\textbf{0.189}$   |
> > > | Electricity | $0.125$    | $\textbf{0.061}$      | $0.063$ | $0.065$      | $\textbf{0.064}$ | $0.069$      | $\textbf{0.064}$      | $0.102$    | $\textbf{0.101}$                 |
> > > |             | $0.25$     | $0.074$      | $\textbf{0.073}$ | $0.076$      | $\textbf{0.073}$ | $0.078$      | $\textbf{0.077}$      | $0.120$    | $0.120$        |
> > > |             | $0.375$    | $\textbf{0.086}$      | $0.089$ | $0.088$      | $\textbf{0.080}$ | $0.085$      | $\textbf{0.083}$      | $0.131$    | $\textbf{0.121}$        |
> > > |             | $0.5$      | $0.098$      | $\textbf{0.091}$ | $0.102$      | $\textbf{0.091}$ | $0.109$      | $\textbf{0.105}$      | $0.151$    | $\textbf{0.142}$        |

---

> > > > ### Author Response · Authors · 2025-11-27
> > > >
> > > > $L=192$, $N=12$
> > > >
> > > > | Dataset     | Mask ratio | iTransformer |            | PatchTST    |            |SAITS|        | BRITS    |        |
> > > > |:-----------:|:----------:|:------------:|:----------:|:-----------:|:----------:|:------:|:------:|:------:|:------:|
> > > > |             |            | Vanilla      | +ITS       | Vanilla     | +ITS       | Vanilla     | +ITS       | Vanilla     | +ITS       |
> > > > | ETTh2       | $0.125$    | $0.245$      | $\textbf{0.254}$ | $0.137$      | $\textbf{0.229}$ | $0.204$   | $\textbf{0.322}$   | $0.575$   | $\textbf{0.570}$   |
> > > > |             | $0.25$     | $0.298$      | $\textbf{0.346}$ | $0.148$      | $\textbf{0.273}$ | $0.227$   | $\textbf{0.413}$   | $0.737$   | $\textbf{0.695}$   |
> > > > |             | $0.375$    | $0.349$      | $\textbf{0.511}$ | $0.155$      | $\textbf{0.328}$ | $0.258$   | $\textbf{0.521}$   | $1.040$   | $\textbf{0.927}$   |
> > > > |             | $0.5$      | $0.421$      | $\textbf{0.552}$ | $0.170$      | $\textbf{0.420}$ | $0.334$   | $\textbf{0.574}$   | $0.721$   | $\textbf{0.709}$   |
> > > > | Electricity | $0.125$    | $\textbf{0.088}$      | $0.091$      | $0.107$      | $\textbf{0.104}$ | $0.200$      | $\textbf{0.184}$      | $0.233$    | $\textbf{0.231}$                 |
> > > > |             | $0.25$     | $0.096$      | $\textbf{0.093}$      | $0.119$      | $\textbf{0.113}$ | $0.206$      | $\textbf{0.187}$      | $0.229$    | $\textbf{0.220}$        |
> > > > |             | $0.375$    | $0.104$      | $\textbf{0.099}$          | $0.133$      | $\textbf{0.140}$ | $0.208$      | $\textbf{0.193}$      | $0.233$    | $\textbf{0.231}$        |
> > > > |             | $0.5$      | $0.114$      | $\textbf{0.105}$ | $0.153$      | $\textbf{0.151}$ | $0.218$      | $\textbf{0.215}$      | $0.240$    | $\textbf{0.232}$        |
> > > >
> > > >
> > > > $L=336$, $N=1$
> > > >
> > > > | Dataset     | Mask ratio | iTransformer |            | PatchTST    |            |SAITS|        | BRITS    |        |
> > > > |:-----------:|:----------:|:------------:|:----------:|:-----------:|:----------:|:------:|:------:|:------:|:------:|
> > > > |             |            | Vanilla      | +ITS       | Vanilla     | +ITS       | Vanilla     | +ITS       | Vanilla     | +ITS       |
> > > > | ETTh2       | $0.125$    | $0.248$      | $\textbf{0.224}$ | $0.086$      | $\textbf{0.079}$ | $0.093$   | $\textbf{0.092}$   | $0.121$   | $\textbf{0.120}$   |
> > > > |             | $0.25$     | $0.408$      | $\textbf{0.386}$ | $0.092$      | $\textbf{0.083}$ | $0.121$   | $\textbf{0.113}$   | $\textbf{0.132}$   | $0.135$   |
> > > > |             | $0.375$    | $0.598$      | $\textbf{0.571}$ | $0.187$      | $\textbf{0.168}$ | $0.119$   | $\textbf{0.111}$   | $0.167$   | $0.167$   |
> > > > |             | $0.5$      | $0.686$      | $\textbf{0.652}$ | $0.186$      | $\textbf{0.180}$ | $0.122$   | $\textbf{0.114}$   | $0.179$   | $0.179$   |
> > > > | Electricity | $0.125$    | $\textbf{0.059}$   | $0.064$    | $0.061$      | $\textbf{0.054}$ | $0.076$      | $\textbf{0.074}$      | $0.122$    | $\textbf{0.121}$                 |
> > > > |             | $0.25$     | $0.073$      | $0.073$          | $0.070$      | $\textbf{0.063}$ | $0.083$      | $\textbf{0.077}$      | $0.137$    | $\textbf{0.130}$        |
> > > > |             | $0.375$    | $0.087$      | $\textbf{0.082}$ | $0.080$      | $\textbf{0.070}$ | $0.145$      | $\textbf{0.133}$      | $0.165$    | $\textbf{0.161}$        |
> > > > |             | $0.5$      | $\textbf{0.098}$      | $0.105$ | $\textbf{0.093}$      | $0.097$ | $0.105$      | $\textbf{0.095}$      | $0.164$    | $\textbf{0.162}$        |

---

> > > > > ### Author Response · Authors · 2025-11-27
> > > > >
> > > > > $L=336$, $N=12$
> > > > >
> > > > > | Dataset     | Mask ratio | iTransformer |            | PatchTST    |            |SAITS|        | BRITS    |        |
> > > > > |:-----------:|:----------:|:------------:|:----------:|:-----------:|:----------:|:------:|:------:|:------:|:------:|
> > > > > |             |            | Vanilla      | +ITS       | Vanilla     | +ITS       | Vanilla     | +ITS       | Vanilla     | +ITS       |
> > > > > | ETTh2       | $0.125$    | $0.276$      | $\textbf{0.254}$ | $0.132$      | $\textbf{0.129}$ | $0.217$   | $\textbf{0.212}$   | $0.569$   | $\textbf{0.550}$   |
> > > > > |             | $0.25$     | $0.341$      | $\textbf{0.326}$ | $0.140$      | $\textbf{0.133}$ | $0.244$   | $\textbf{0.233}$   | $0.703$   | $\textbf{0.685}$   |
> > > > > |             | $0.375$    | $0.409$      | $\textbf{0.381}$ | $0.153$      | $\textbf{0.138}$ | $0.365$   | $\textbf{0.351}$   | $0.860$   | $\textbf{0.837}$   |
> > > > > |             | $0.5$      | $0.521$      | $\textbf{0.502}$ | $0.169$      | $\textbf{0.150}$ | $0.454$   | $\textbf{0.444}$   | $0.826$   | $\textbf{0.809}$   |
> > > > > | Electricity | $0.125$    | $0.084$      | $0.084$          | $0.095$      | $\textbf{0.094}$ | $0.201$      | $\textbf{0.194}$      | $0.228$    | $\textbf{0.221}$                 |
> > > > > |             | $0.25$     | $0.094$      | $\textbf{0.093}$ | $\textbf{0.106}$      | $0.113$ | $0.295$      | $\textbf{0.287}$      | $0.335$    | $\textbf{0.330}$        |
> > > > > |             | $0.375$    | $0.101$      | $\textbf{0.099}$ | $\textbf{0.119}$      | $0.120$ | $0.350$      | $\textbf{0.333}$      | $0.285$    | $\textbf{0.271}$        |
> > > > > |             | $0.5$      | $0.108$      | $\textbf{0.105}$ | $0.135$      | $\textbf{0.131}$ | $0.415$      | $\textbf{0.399}$      | $0.342$    | $\textbf{0.332}$        |
> > > > >
> > > > > We have added an additional set of experiments to compare against the previously provided supplementary results.
> > > > >
> > > > > $L=720$, $N=1$
> > > > >
> > > > > | Dataset     | Mask ratio | iTransformer |            | PatchTST    |            |SAITS|        | BRITS    |        |
> > > > > |:-----------:|:----------:|:------------:|:----------:|:-----------:|:----------:|:------:|:------:|:------:|:------:|
> > > > > |             |            | Vanilla      | +ITS       | Vanilla     | +ITS       | Vanilla     | +ITS       | Vanilla     | +ITS       |
> > > > > | ETTh2       | $0.125$    | $0.268$      | $\textbf{0.254}$ | $0.078$      | $\textbf{0.069}$ | $0.111$   | $\textbf{0.102}$   | $0.099$   | $\textbf{0.092}$   |
> > > > > |             | $0.25$     | $0.445$      | $\textbf{0.426}$ | $0.080$      | $\textbf{0.073}$ | $0.142$   | $\textbf{0.133}$   | $0.211$   | $\textbf{0.205}$   |
> > > > > |             | $0.375$    | $0.684$      | $\textbf{0.661}$ | $0.083$      | $\textbf{0.078}$ | $0.123$   | $\textbf{0.111}$   | $0.134$   | $\textbf{0.127}$   |
> > > > > |             | $0.5$      | $0.992$      | $\textbf{0.852}$ | $0.088$      | $\textbf{0.080}$ | $0.200$   | $\textbf{0.184}$   | $0.193$   | $\textbf{0.182}$   |
> > > > > | Electricity | $0.125$    | $0.061$      | $\textbf{0.059}$ | $\textbf{0.059}$      | $0.060$ | $0.064$      | $\textbf{0.058}$      | $0.101$    | $\textbf{0.090}$        |
> > > > > |             | $0.25$     | $0.075$      | $\textbf{0.073}$ | $0.076$      | $\textbf{0.073}$ | $0.072$      | $\textbf{0.067}$      | $0.115$    | $\textbf{0.108}$        |
> > > > > |             | $0.375$    | $\textbf{0.088}$  | $0.089$     | $0.094$      | $\textbf{0.090}$ | $0.101$      | $\textbf{0.093}$      | $0.192$    | $\textbf{0.181}$        |
> > > > > |             | $0.5$      | $0.101$      | $\textbf{0.095}$ | $0.091$      | $\textbf{0.085}$ | $0.108$      | $\textbf{0.095}$      | $0.174$    | $\textbf{0.162}$        |

---

### Official Review · Reviewer_tnTg · 2025-10-30

**Soundness:** 3
**Presentation:** 2
**Contribution:** 2
**Rating:** 4
**Confidence:** 3

**Summary:**

This paper applies inference-time scaling to time-series models by generating multiple predictions via MC Dropout, scoring them using reconstruction error from a backward model, and combining them with learned weights. Experiments on forecasting and imputation tasks show consistent improvements across 9 methods and 7 datasets, at the cost of 1-15x slower inference.

**Strengths:**

- The paper is the first to address inference time scaling in time series forecasting.
- Using a reverse-temporal model for reconstruction verification is a good domain-specific adaptation.
- First systematic study of inference-time scaling for mainstream time-series models (prior works seem to have only worked on LLMs for TSF).
- Comprehensive evaluation across architectures and tasks.

**Weaknesses:**

- The theoretical insight is naive and does not provide much insight. The method is not novel in other domains, so it is more like an engineering work with good empirical results.
- The baselines are simple (see Figure 4). Although there are no prior works but there should be more sophisticated weighting methods they can compare with. This can demonstrate if the backward model actually provides effective guidance or if the effect simply comes from averaging.
- The computation cost is somewhat high (1-15x slower with 2-12% improvement), and they did not mention or count the computation cost for training the backward model.

**Questions:**

Can you try to justify the effectiveness of the backward model as a guidance more clearly? I believe comparing with more comprehensive baselines (for example entropy-guided or average) would also solve this.
I will raise my score if you solve this well (since the score I would like to give now is somewhat between 4 and 6).

---

> ### Author Response · Authors · 2025-11-21
>
> Dear Reviewer tnTg,
>
> We sincerely appreciate your thorough review and constructive comments.  We also greatly appreciate your **recognition of the soundness of our proposed method and the design of the backward model**, as well as your positive evaluation of the insights contributed by our work.
>
> >**W1** The theoretical insight is naive and does not provide much insight. The method is not novel in other domains, so it is more like an engineering work with good empirical results.
>
> Thanks very much for this question, which helped us clarify the core contributions and innovations of our work. In response to the concern, we would like to add two points for clarification:
> * Methodologically, our main contribution is **proposing a general inference-time scaling framework that can be applied to mainstream time-series models**. **To our knowledge, inference-time scaling in the time-series domain has not been explored before.** The framework helps time-series models possess the self-consistency and self-reflection mechanisms of LLMs, while remaining decoupled from specific forward models and candidate-generation schemes, allowing it to be used in a plug-and-play manner with the vast majority of commonly used time-series models.  It supports both forecasting and imputation tasks and provides stable performance improvements without retraining the forward model, which is particularly important in the context of increasing emphasis on training cost and resource constraints, as well as the growing saturation of current time-series benchmarks.  We believe that this inference-time perspective has certain value for advancing the development of the time-series.
> * Theoretically, we **provide a Bayesian approximate explanation framework**. Traditional structural approximations are mainly the “structural posterior” induced by the network architecture, and on this basis we explicitly introduce a model error and express the approximate posterior as $p(\boldsymbol{W}\mid D)\sim p(\textbf{z})\, f[E(\boldsymbol{W})]$. Building on this approximation, we arrive at a reweighted strategy $p(y|\boldsymbol{x},D) = \sum_{k}w_kp(y|\boldsymbol{x}, \boldsymbol{W}_k)$, and this theory unifies the model’s structural preference and performance preference within a single framework, providing theoretical support for our general inference-time scaling methodology.
> * As we clarified in our response to Reviewer yT6j (W1–2), our theoretical framework **can also explain why certain weighting strategies, such as WTA and Majority, lead to degraded ITS performance**. Therefore, our theory not only supports our reconstruction-based weighting but also provides a principled explanation for why other weighting schemes fail.
>
> >**W2** The baselines are simple (see Figure 4). Although there are no prior works but there should be more sophisticated weighting methods they can compare with. This can demonstrate if the backward model actually provides effective guidance or if the effect simply comes from averaging.
>
> >**Q1** Can you try to justify the effectiveness of the backward model as a guidance more clearly? I believe comparing with more comprehensive baselines (for example entropy-guided or average) would also solve this. I will raise my score if you solve this well (since the score I would like to give now is somewhat between 4 and 6).
>
> Thanks for the attentive reading and helpful suggestion. We consider these two questions similar and therefore address them together.
> We apologize for not clearly describing the baseline settings in the ablation study, which may indeed have caused misunderstandings. We humbly respond to your concerns from the following two aspects:
> * In fact, **the core scheme used in the main text and in Appendix F.1 of the original submission is the reconstruction-based weighting (the green bars in Fig.~4).** All other weighting strategies (e.g., Majority, WTA) are implemented within ITS only as comparison baselines.
> * In addition, **Majority Voting ($w_k \equiv \frac{1}{K}$) in the original submission is implemented as uniform averaging over all candidates** and does **NOT rely** on the backward model, which is already stated **in Lines 282–298 of the original submission**.
>
> We have analyzed experimental comparisons with different weighting methods to more clearly demonstrate the effectiveness of the backward model.
> We would like to clarify that **Entropy, Majority, and WTA are ablation baselines obtained by replacing the weighting scheme in ITS, while our proposed method corresponds to the reconstruction-based weighting**:
>
> [Continued below]

---

> ### Author Response · Authors · 2025-11-21
>
> * **Ours**: reconstruction–based weighting, $p(y|\boldsymbol{x},D) = \sum_{k}w_kp(y|\boldsymbol{x}, \boldsymbol{W}_k)$, where $w_k$ is a weight derived from the reconstruction error of candidate $k$, which is obtained using the backward model;
> * **Entropy**: entropy-based weighting using the predictive uncertainty of each candidate, $p(y|\boldsymbol{x},D) = \sum_{k}\hat{w}_kp(y|\boldsymbol{x}, \boldsymbol{W}_k)$, where $\hat{w}_k$ is derived from the predictive uncertainty of candidate $k$;
> * **Majority Voting (Majority)**: uniform averaging over all candidates, $p(y|\boldsymbol{x},D) = \frac{1}{K}\sum_{k}p(y|\boldsymbol{x}, \boldsymbol{W}_k)$;
> * **Winners-Take-All (WTA)**: selecting the single candidate with the lowest reconstruction error, $p(y|\boldsymbol{x},D) =   p(y|\boldsymbol{x}, \boldsymbol{\hat{W}^\ast})$, where $\hat{\boldsymbol{W}}^{\ast}$ denotes the parameters of the candidate model with the lowest reconstruction error.
>
> For forecasting:
>
> | Model       | Dataset | Look_back | Pred_len  | Vanilla | Ours  | Entropy| Majority | WTA   |
> |:------------:|:---------:|:-----------:|:------------:|:---------:|:-------:|:----------------:|:---------:|:-------:|
> | iTransformer | ETTh1  | $96$         | $96$         | $0.411$   | $\textbf{0.407}$ | $0.409$          | $0.410$   | $0.423$ |
> |              |        | $96$        | $192$         | $0.462$   | $\textbf{0.459}$ | $0.461$          | $0.461$   | $0.470$ |
> |              |        | $96$        | $336$         | $0.492$   | $\textbf{0.487}$ | $0.488$          | $0.490$   | $0.497$ |
> |              |        | $96$        | $720$         | $0.512$   | $\textbf{0.503}$ | $0.508$          | $0.510$   | $0.506$ |
> |              |Electricity| $96$      | $96$         | $0.150$   | $\textbf{0.145}$ | $0.147$          | $0.149$   | $0.156$ |
> |              |        | $96$        | $192$         | $0.167$   | $\textbf{0.158}$ | $0.164$          | $0.165$   | $0.172$ |
> |              |        | $96$        | $336$         | $0.181$   | $\textbf{0.172}$ | $0.178$          | $0.180$   | $0.188$ |
> |              |        | $96$        | $720$         | $0.222$   | $\textbf{0.210}$ | $0.219$          | $0.222$   | $0.229$ |
> | Crossformer  | ETTh1   | $96$        | $96$         | $0.395$   | $\textbf{0.391}$ | $0.399$          | $0.396$   | $0.412$ |
> |              |         | $96$       | $192$         | $0.628$   | $\textbf{0.614}$ | $0.628$          | $0.631$   | $0.629$ |
> |              |         | $96$       | $336$         | $0.665$   | $\textbf{0.657}$ | $0.673$          | $0.669$   | $0.674$ |
> |              |         | $96$       | $720$         | $0.887$   | $0.825$ | $0.861$          | $0.854$   | $\textbf{0.789}$ |
> |              |Electricity| $96$      | $96$         | $0.219$   | $\textbf{0.218}$ | $0.226$          | $0.219$   | $0.248$ |
> |              |         | $96$       | $192$         | $\textbf{0.237}$   | $0.238$ | $0.239$          | $0.238$   | $0.265$ |
> |              |         | $96$       | $336$         | $0.785$   | $0.778$ | $0.781$          | $0.786$   | $\textbf{0.775}$ |
> |              |         | $96$       | $720$         | $0.813$   | $\textbf{0.810}$ | $0.811$          | $0.813$   | $0.815$ |
> | PatchTST     | ETTh1   | $96$        | $96$         | $\textbf{0.408}$   | $0.409$ | $0.414$          | $0.409$   | $0.451$ |
> |              |         | $96$       | $192$         | $0.460$   | $\textbf{0.459}$ | $0.469$          | $0.461$   | $0.497$ |
> |              |         | $96$       | $336$         | $0.494$   | $\textbf{0.492}$ | $0.503$          | $0.495$   | $0.526$ |
> |              |         | $96$       | $720$         | $0.495$   | $\textbf{0.489}$ | $0.499$          | $0.495$   | $0.525$ |
> |              |Electricity| $96$      | $96$         | $\textbf{0.193}$   | $0.196$ | $0.199$          | $0.197$   | $0.241$ |
> |              |         | $96$       | $192$         | $0.199$   | $\textbf{0.193}$ | $0.197$          | $0.203$   | $0.248$ |
> |              |         | $96$       | $336$         | $0.214$   | $\textbf{0.207}$ | $0.229$          | $0.217$   | $0.264$ |
> |              |         | $96$       | $720$         | $0.256$   | $\textbf{0.249}$ | $0.251$          | $0.259$   | $0.309$ |
>
> [Continued below]

---

> > ### Author Response · Authors · 2025-11-21
> >
> > For imputation:
> >
> > | Model       | Dataset | Seq len | mask ratio  | Vanilla | Ours  | Entropy| Majority | WTA   |
> > |:------------:|:---------:|:-----------:|:------------:|:---------:|:-------:|:----------------:|:---------:|:-------:|
> > | iTransformer | ETTh1  | $96$        | $0.125$       | $0.262$   | $\textbf{0.255}$ | $0.267$          | $0.269$   | $0.262$ |
> > |              |        | $96$        | $0.25$        | $0.303$   | $\textbf{0.290}$ | $0.304$          | $0.306$   | $0.296$ |
> > |              |        | $96$        | $0.375$       | $0.337$   | $\textbf{0.321}$ | $0.343$          | $0.340$   | $0.323$ |
> > |              |        | $96$        | $0.5$         | $0.379$   | $0.361$ | $0.381$          | $0.383$   | $\textbf{0.358}$ |
> > |              |Electricity| $96$     | $0.125$       | $\textbf{0.092}$   | $0.094$ | $0.093$          | $\textbf{0.092}$   | $0.110$ |
> > |              |        | $96$        | $0.25$        | $\textbf{0.103}$   | $0.104$ | $0.104$          | $\textbf{0.103}$   | $0.123$ |
> > |              |        | $96$        | $0.375$       | $\textbf{0.116}$   | $0.117$ | $0.117$          | $\textbf{0.116}$   | $0.137$ |
> > |              |        | $96$        | $0.5$         | $\textbf{0.132}$   | $0.133$ | $0.134$          | $\textbf{0.132}$   | $0.153$ |
> > | Crossformer  | ETTh1   | $96$       | $0.125$       | $0.300$   | $\textbf{0.285}$ | $0.302$          | $0.305$   | $0.287$ |
> > |              |         | $96$       | $0.25$        | $0.335$   | $\textbf{0.315}$ | $0.335$          | $0.340$   | $0.316$ |
> > |              |         | $96$       | $0.375$       | $0.362$   | $\textbf{0.339}$ | $0.368$          | $0.367$   | $0.342$ |
> > |              |         | $96$       | $0.5$         | $0.398$   | $\textbf{0.373}$ | $0.394$          | $0.403$   | $0.375$ |
> > |              |Electricity| $96$     | $0.125$       | $0.174$   | $\textbf{0.167}$ | $0.174$          | $0.177$   | $0.171$ |
> > |              |         | $96$       | $0.25$        | $0.188$   | $\textbf{0.178}$ | $0.195$          | $0.192$   | $0.182$ |
> > |              |         | $96$       | $0.375$       | $0.206$   | $\textbf{0.195}$ | $0.201$          | $0.210$   | $0.200$ |
> > |              |         | $96$       | $0.5$         | $0.226$   | $\textbf{0.214}$ | $0.223$          | $0.229$   | $0.220$ |
> > | PatchTST     | ETTh1   | $96$       | $0.125$       | $0.320$   | $0.300$ | $0.320$          | $0.324$   | $\textbf{0.293}$ |
> > |              |         | $96$       | $0.25$        | $0.370$   | $0.342$ | $0.368$          | $0.375$   | $\textbf{0.337}$ |
> > |              |         | $96$       | $0.375$       | $0.401$   | $0.368$ | $0.397$          | $0.405$   | $\textbf{0.365}$ |
> > |              |         | $96$       | $0.5$         | $0.444$   | $0.407$ | $0.449$          | $0.448$   | $\textbf{0.399}$ |
> > |              |Electricity| $96$     | $0.125$       | $\textbf{0.138}$   | $\textbf{0.138}$  | $0.142$          | $0.144$   | $0.164$ |
> > |              |         | $96$       | $0.25$        | $0.155$   | $\textbf{0.153}$ | $0.156$          | $0.161$   | $0.179$ |
> > |              |         | $96$       | $0.375$       | $0.180$   | $\textbf{0.176}$ | $0.180$          | $0.186$   | $0.203$ |
> > |              |         | $96$       | $0.5$         | $0.213$   | $\textbf{0.206}$ | $0.214$          | $0.219$   | $0.229$ |
> >
> > To better highlight our contributions, we have expanded and refined the discussion of weighting strategies in Appendix F.1 of the revised version.
> >
> > >**W3.1** The computation cost is somewhat high (1-15x slower with 2-12\% improvement).
> >
> > Thanks for this question. In the main text, we have analyzed the computation cost and show that **approximately 85% of the overhead comes from the candidate-generation stage (MC Dropout) in the original submission**. Since this component is decoupled from the core of our framework, we plan to explore more lightweight candidate-generation methods in future work to reduce this cost. In addition, we provide detailed inference-time tables on representative datasets and models, and these have been added to Appendix F.5 of the revised version.
> >
> > [Continued below]

---

> > > ### Author Response · Authors · 2025-11-21
> > >
> > > | Dataset     | Model        | L      | S      | Vanilla (ms / example) | +ITS (ms / example) |
> > > |:-----------:|:------------:|:------:|:------:|:-----------------------:|:--------------------:|
> > > | ETTh2       | iTransformer | $96$   | $96$   | $0.094$                 | $0.099$              |
> > > |             |              |        | $192$  | $0.128$                 | $0.193$              |
> > > |             |              |        | $336$  | $0.143$                 | $0.283$              |
> > > |             |              |        | $720$  | $0.184$                 | $0.423$              |
> > > |             | PatchTST     | $96$   | $96$   | $0.112$                 | $0.351$              |
> > > |             |              |        | $192$  | $0.124$                 | $0.403$              |
> > > |             |              |        | $336$  | $0.133$                 | $0.534$              |
> > > |             |              |        | $720$  | $0.221$                 | $0.948$              |
> > > | Weather     | iTransformer | $96$   | $96$   | $0.257$                 | $0.598$              |
> > > |             |              |        | $192$  | $0.271$                 | $0.626$              |
> > > |             |              |        | $336$  | $0.313$                 | $0.892$              |
> > > |             |              |        | $720$  | $0.374$                 | $0.998$              |
> > > |             | PatchTST     | $96$   | $96$   | $0.112$                 | $0.684$              |
> > > |             |              |        | $192$  | $0.128$                 | $0.898$              |
> > > |             |              |        | $336$  | $0.165$                 | $1.235$              |
> > > |             |              |        | $720$  | $0.222$                 | $1.753$              |
> > > | Electricity | iTransformer | $96$   | $96$   | $0.482$                 | $2.882$              |
> > > |             |              |        | $192$  | $0.602$                 | $3.654$              |
> > > |             |              |        | $336$  | $0.705$                 | $4.490$              |
> > > |             |              |        | $720$  | $1.774$                 | $11.125$             |
> > > |             | PatchTST     | $96$   | $96$   | $0.838$                 | $8.472$              |
> > > |             |              |        | $192$  | $0.911$                 | $9.315$              |
> > > |             |              |        | $336$  | $0.944$                 | $11.987$             |
> > > |             |              |        | $720$  | $2.081$                 | $22.150$             |
> > >
> > > [Continued below]

---

> > > > ### Author Response · Authors · 2025-11-21
> > > >
> > > > | Dataset     | Model        | L      | Mask ratio | Vanilla (ms / example) | +ITS (ms / example) |
> > > > |:-----------:|:------------:|:------:|:----------:|:-----------------------:|:--------------------:|
> > > > | ETTh2       | iTransformer | $96$   | $0.125$    | $0.107$                 | $0.546$              |
> > > > |             |              |        | $0.25$     | $0.158$                 | $0.780$              |
> > > > |             |              |        | $0.375$    | $0.161$                 | $0.813$              |
> > > > |             |              |        | $0.5$      | $0.169$                 | $0.846$              |
> > > > |             | PatchTST     | $96$   | $0.125$    | $0.119$                 | $0.489$              |
> > > > |             |              |        | $0.25$     | $0.108$                 | $0.446$              |
> > > > |             |              |        | $0.375$    | $0.089$                 | $0.410$              |
> > > > |             |              |        | $0.5$      | $0.106$                 | $0.454$              |
> > > > | Weather     | iTransformer | $96$   | $0.125$    | $0.117$                 | $1.145$              |
> > > > |             |              |        | $0.25$     | $0.111$                 | $1.139$              |
> > > > |             |              |        | $0.375$    | $0.112$                 | $1.180$              |
> > > > |             |              |        | $0.5$      | $0.107$                 | $1.088$              |
> > > > |             | PatchTST     | $96$   | $0.125$    | $0.218$                 | $2.443$              |
> > > > |             |              |        | $0.25$     | $0.182$                 | $1.936$              |
> > > > |             |              |        | $0.375$    | $0.223$                 | $2.457$              |
> > > > |             |              |        | $0.5$      | $0.191$                 | $2.181$              |
> > > > | Electricity | iTransformer | $96$   | $0.125$    | $0.495$                 | $7.033$              |
> > > > |             |              |        | $0.25$     | $0.431$                 | $6.331$              |
> > > > |             |              |        | $0.375$    | $0.449$                 | $6.752$              |
> > > > |             |              |        | $0.5$      | $0.433$                 | $6.352$              |
> > > > |             | PatchTST     | $96$   | $0.125$    | $1.143$                 | $15.019$             |
> > > > |             |              |        | $0.25$     | $1.388$                 | $19.196$             |
> > > > |             |              |        | $0.375$    | $1.451$                 | $20.357$             |
> > > > |             |              |        | $0.5$      | $1.391$                 | $19.014$             |
> > > >
> > > > >**W3.2** they did not mention or count the computation cost for training the backward model.
> > > >
> > > > This is another important question. We apologize for not explaining this aspect clearly in the main text, mainly due to space limitations. However, **Appendix E.1 of the original submission** in fact provides a complete and detailed description of the backward model’s training procedure, from which the required computational cost can be obtained.
> > > >
> > > > We sincerely appreciate the constructive feedback you have provided and hope that our responses and the modifications to the manuscript adequately address your insightful feedback and increases your impression and confidence in our work.  We are happy to provide any further clarifications if needed.

---

> > > > > ### Comment · Reviewer_tnTg · 2025-11-21
> > > > >
> > > > > I noticed the training config, but it does not specifically mention (though I could infer roughly based on my own knowledge) the cost of training the backward model compared with the inference cost. You should try to clarify this.
> > > > >
> > > > > I do not deny the methodological contribution of the paper, since even if it is a straightforward method, we need someone to do and implement it, but it would certainly affect my evaluation if I find the method somehow intuitive, even though not done before. The theoretical contribution is also reasonable in an explanatory way, so I would say the contribution of this paper, in this sense, is acceptable.
> > > > >
> > > > > I appreciate the extra experiments which actually solidify the effectiveness of adopting such a backward model. I would raise my score for this. It would be great if you could make these points you made in the discussion period clearer in the paper.

---

> > > > > > ### Author Response · Authors · 2025-11-22
> > > > > >
> > > > > > Dear Reviewer tnTg,
> > > > > >
> > > > > > Thank you very much for your timely reply!   We are truly grateful that you found our responses to be thorough and that they have addressed your concerns.   We also sincerely appreciate your decision to raise your score—this means a great deal to us and serves as a strong encouragement.
> > > > > >
> > > > > > We have added the time cost required to train the backward model in Appendix E.1 of the revised version. Specifically, the backward model first needs to be pretrained on a large-scale dataset, and this stage is relatively expensive, requiring roughly **one month** on two A100 80G GPUs. **Notably, we will make the pretrained weights publicly available later, which will eliminate this burden for future users.** For task-specific fine-tuning, **the cost is comparatively low** and the backward model can be fine-tuned using only one NVIDIA A100 80G GPU. We have also included a table summarizing the detailed time cost.
> > > > > >
> > > > > > | Task        | Dataset     | Batch | Epoch | Time      |
> > > > > > |:-----------:|:-----------:|:-----:|:-----:|:---------:|
> > > > > > | Forecasting | ETTh1       | 1024  | 30    | 358.24s   |
> > > > > > |             | ETTh2       | 1024  | 30    | 364.15s   |
> > > > > > |             | ETTm1       | 1024  | 30    | 28.24min  |
> > > > > > |             | ETTm2       | 1024  | 30    | 27.48min  |
> > > > > > |             | Electricity | 32    | 30    | 2.14hour  |
> > > > > > |             | Exchange    | 1024  | 30    | 294.58s   |
> > > > > > |             | Weather     | 512   | 30    | 43.62min  |
> > > > > > | Imputation  | ETTh1       | 1024  | 30    | 122.47s   |
> > > > > > |             | ETTh2       | 1024  | 30    | 124.12s   |
> > > > > > |             | ETTm1       | 1024  | 30    | 172.39s   |
> > > > > > |             | ETTm2       | 1024  | 30    | 325.28s   |
> > > > > > |             | Electricity | 512   | 30    | 39.21min  |
> > > > > > |             | Exchange    | 1024  | 30    | 72.69s    |
> > > > > > |             | Weather     | 1024  | 30    | 406.31s   |
> > > > > >
> > > > > > We are fully committed to incorporating all the revisions discussed during the rebuttal phase into our final manuscript, as these changes have greatly contributed to improving the quality and clarity of our work. Once again, thank you for your valuable feedback and for the important role you have played in helping us strengthen our paper!
> > > > > >
> > > > > > Kind regards,
> > > > > >
> > > > > > Authors

---

### Official Review · Reviewer_SLG7 · 2025-11-01

**Soundness:** 3
**Presentation:** 3
**Contribution:** 3
**Rating:** 6
**Confidence:** 3

**Summary:**

The paper proposes an inference-time scaling method for time series data. The proposed method can be applied to both time series forecasting and time series imputation tasks. The core idea is to create multiple outputs from a pretrained model and then weight the outputs based on how well they reconstruct the input. The proposed method yields improved performance in the experiments, demonstrating the efficacy of the proposed approach.

**Strengths:**

The paper presents a robust empirical evaluation, utilising multiple datasets and methods.

The reweighting strategy is an interesting and effective method based on the reconstruction that improved performance across datasets.

**Weaknesses:**

While the experiment section is strong, it only reports single-time statistics. To understand the full efficacy of methods, it would be better to have the experiments repeated over multiple runs and presented with the standard deviations.

See "Questions*" below for more

**Questions:**

1. Line 67-68: These line seems unnecessary given the same information appears in the contributions just below:

2. The multi-candidate generation is done via Monte Carlo Dropout, which can provide very narrow sets of candidates. Furthermore, the probabilistic statements made from it are largely miscalibrated. It would be interesting to see how the results differ when using generative models such as flow-based models with or without calibrated probabilistic statements via Conformal Prediction.

3. Line 160-161: A backward model needs to be trained for reconstruction. However, a strong backward model might be able to reconstruct from any of the given candidates. An interesting example could be how diffusion models work; even if there is complete noise, one might be able to reconstruct the true distribution, even if the score function is known.

4. Line 204: X belong to R (F X C), does it need to only reconstruct S steps in the past? Also, why was it needed to divide the prediction horizon into segments? All of the prediction horizon could be predicted at once, no?

5. The theoretical analysis seems restricted to MC dropout; are any other forms of probabilistic candidate generation supported?

6. Line 341: "the Exchange dataset, we follow the hyperparameter settings of the Weather dataset": Why were the same hyperparameters used as in the Weather dataset? The datasets seem quite different. Isn't it better to perform a hyperparameter search for the Exchange dataset separately?

7. How critical is the training of the backward model? How well-trained should it be?

8. Line 419: "Exchagne" is written instead of "Exchange".

9. Is there any rule of thumb to find the right number of candidates, K?

10. Only reports single-time statistics. To understand the full efficacy of methods, it would be better to have the experiments repeated over multiple runs and presented with the standard deviations.

---

> ### Author Response · Authors · 2025-11-21
>
> Dear Reviewer SLG7,
>
> We sincerely appreciate your thorough review and constructive comments. We are grateful for your **recognition of our experiment and reweighting strategy**, as well as your positive assessment of the insights contributed by our work.
> > **Q1** Line 67-68: These line seems unnecessary given the same information appears in the contributions just below:
>
> Thanks for the attentive reading and helpful suggestion. We fully agree that line 67–68 are redundant and have removed them. We have also taken this opportunity to carefully re-examine the entire manuscript to improve clarity and consistency wherever needed.
>
> >**Q2** The multi-candidate generation is done via Monte Carlo Dropout, which can provide very narrow sets of candidates. Furthermore, the probabilistic statements made from it are largely miscalibrated. It would be interesting to see how the results differ when using generative models such as flow-based models with or without calibrated probabilistic statements via Conformal Prediction.
>
> Thank you for this insightful question and thoughtful observation. We humbly respond to your concerns from the following aspects:
> * First, **we consider that the MC Dropout candidates are sufficiently diverse in our setting**, because increasing the number of samples $K$ and adjusting the dropout $p_d$ provide sufficient diversity in practice. To further verify this, we analyzed candidate diversity using both MC Dropout and a flow-based model. As shown in the table below, Norm Pairwise L2 measures the average normalized distance between different candidates, while Norm Ensemble Variance measures the normalized variance across the $K$ candidates. The results indicate that the diversity metrics of MC Dropout are only slightly lower than those of the flow-based model.
>
>   \
>   For MC Dropout:
>   | K  | Dropout | Norm Pairwise L2 | Norm Ensemble Variance |
>   |:----:|:---------:|:------------------:|:--------------------:|
>   | $8$  | $0.1$     | $0.2238$           | $0.0219$             |
>   | $32$ | $0.1$     | $0.2441$           | $0.0263$             |
>   | $64$ | $0.1$     | $0.2812$           | $0.0389$             |
>   | $64$ | $0.2$     | $0.3322$           | $0.0543$             |
>
>   For a flow-based model:
>   | K  | Norm Pairwise L2 | Norm Ensemble Variance |
>   |:----:|:------------------:|:--------------------:|
>   | $8$  | $0.2397$           | $0.0256$             |
>   | $32$ | $0.2653$           | $0.0315$             |
>   | $64$ | $0.3128$           | $0.0447$             |
>
> * Second, **our fusion does NOT rely on the probabilistic statements reported by MC Dropout**, the weights are derived solely from the reconstruction errors of the backward model,  as described in **Line 170–178 of the original submission**.
>
> To facilitate a clearer understanding of our work, we have added this analysis to Appendix F.2 of the revised version.
>
> >**Q3** Line 160-161: A backward model needs to be trained for reconstruction. However, a strong backward model might be able to reconstruct from any of the given candidates. An interesting example could be how diffusion models work; even if there is complete noise, one might be able to reconstruct the true distribution, even if the score function is known.
>
> Thank you for this inspiring question, and we apologize for not having clearly explained the role of the backward model. **Our backward model is a supervised predictor trained on time series. It reconstructs historical segments or actively masked missing parts along the temporal dimension using the input sequence**, rather than inverting a known noise process with a learned score function as in diffusion models. We have described this more explicitly in Section 3.1 of the revised version and plan to investigate using diffusion models as backward models in future work.
>
> >**Q4** Line 204: X belong to R (F X C), does it need to only reconstruct S steps in the past? Also, why was it needed to divide the prediction horizon into segments? All of the prediction horizon could be predicted at once, no?
>
> Good question! You are right, only $S$ steps in the past need to be reconstructed. The algorithm in the main text of the original submission is instantiated for autoregressive models, whose prediction mechanism naturally proceeds segment by segment. **In principle, our framework also supports one-shot prediction over the full forecast horizon, and Appendix A.1 in the original submission provides a dedicated algorithm and discussion for non-autoregressive models.** Additionally, autoregressive models can also be used in this one-shot setting by first generating the full horizon and then applying the backward reconstruction. We have clarified this more explicitly in Section 3.2 of the revised version.
>
> [Continued below]

---

> ### Author Response · Authors · 2025-11-21
>
> >**Q5** The theoretical analysis seems restricted to MC dropout; are any other forms of probabilistic candidate generation supported?
>
> Thanks for this insightful comment. In our framework, **candidate generation is designed to be pluggable, and as stated in Line 153 of the original submission, multiple candidate-generation approaches can be used**. Inspired by your **suggestion in Q2**, we replaced MC Dropout with a flow-based model, and as shown in the results below, our framework still achieves consistent improvements.
>
>   | Dataset | Look back | Pred len | iTransformer |                        | PatchTST    |            |
>   |:-------:|:--------:|:--------:|:------------:|:---------:|:-----------:|:---------:|
>   |         |          |             | Vanilla    | +ITS(Flow)              | Vanilla    | +ITS(Flow)              |
>   | ETTh1   | $720$     | $96$       | $0.411$    | $\textbf{0.408}$        | $0.408$    | $\textbf{0.403}$        |
>   |         |           | $192$      | $0.462$    | $\textbf{0.458}$        | $0.460$    | $\textbf{0.459}$        |
>   |         |           | $336$      | $0.492$    | $\textbf{0.484}$        | $0.494$    | $\textbf{0.481}$        |
>   |         |           | $720$      | $0.512$    | $\textbf{0.495}$        | $0.495$    | $\textbf{0.477}$        |
>
> >**Q6** Line 341: "the Exchange dataset, we follow the hyperparameter settings of the Weather dataset": Why were the same hyperparameters used as in the Weather dataset? The datasets seem quite different. Isn't it better to perform a hyperparameter search for the Exchange dataset separately?
>
> Thank you very much for pointing this out. We apologize for the unintended overstatement regarding the Exchange hyperparameter settings. What we intended to convey is that TimeXer lacks an official implementation on Exchange, which led to missing hyperparameters. Because our focus was on comparing Vanilla and +ITS, we overlooked that this could introduce unfairness across methods. **We have corrected the TimeXer configuration on the Exchange dataset and re-run all relevant experiments**, and the updated results have been incorporated into the revised version.
>
> | Dataset | Look back | Pred len |MSE|                        |MAE|            |
> |:-------:|:--------:|:--------:|:------------:|:---------:|:-----------:|:---------:|
> |         |          |             | Vanilla    | +ITS                    | Vanilla    | +ITS                    |
> |Exchange | $96$      | $96$       | $\textbf{0.086}$    | $0.088$        | $\textbf{0.205}$    | $0.206$        |
> |         |           | $192$      | $0.187$    | $\textbf{0.184}$        | $0.307$    | $\textbf{0.306}$        |
> |         |           | $336$      | $0.335$    | $\textbf{0.326}$        | $0.417$    | $\textbf{0.411}$        |
> |         |           | $720$      | $0.862$    | $\textbf{0.849}$        | $0.694$    | $\textbf{0.688}$        |
>
> >**Q7** How critical is the training of the backward model? How well-trained should it be?
>
> We sincerely appreciate this question. In our framework, the backward model plays an important role as the component that evaluates candidate quality. **Consistent with the view kindly raised by Reviewer yT6j, the stronger the reconstruction ability of the backward model, the better**. Specifically:
> * When it reconstructs more accurately, the reconstruction-error signal has higher discriminative power across different candidates, allowing it to more effectively distinguish those that better follow the historical evolution pattern of the time series from those that deviate significantly.
> * At the same time, stronger reconstruction ability implies less random noise embedded in this error signal, so it tends to exhibit a more stable and exploitable correlation with the true forecasting error, thereby leading to more reliable weighting results overall.
>
> We compared the performance gains of ITS under backward models with different capability levels, and the experimental results support our conclusion.
>
> | model        | dataset | Look back | Pred len | Vanilla  | FT-BM(Ours) | PatchTST | TimeMixer | TimeXer | iTransformer | Crossformer |
> |:--------------:|:---------:|:-----------:|:----------:|:----------:|:-------:|:--------------:|:-----------:|:---------:|:----------:|:-------------:|
> | PatchTST     | ETTm2   | $96$      | $96$     | $0.177$  | $\textbf{0.166}$ | $0.176$    | $0.175$   | $0.176$ | $0.174$  | $0.177$     |
> |              |         |           | $192$    | $0.243$  | $\textbf{0.233}$ | $0.242$    | $0.241$   | $0.244$ | $0.240$  | $0.242$     |
> |              |         |           | $336$    | $0.305$  | $\textbf{0.293}$ | $0.303$    | $0.306$   | $0.302$ | $0.301$  | $0.303$     |
> |              |         |           | $720$    | $0.410$  | $\textbf{0.395}$ | $0.406$    | $0.405$   | $0.407$ | $0.406$  | $0.407$     |
>
> [Continued below]

---

> > ### Author Response · Authors · 2025-11-21
> >
> > > **Q8** Line 419: "Exchagne" is written instead of "Exchange".
> >
> > Thank you very much for the careful suggestion. We have corrected “Exchagne” to “Exchange” and have carefully checked all other content.
> >
> > >**Q9** Is there any rule of thumb to find the right number of candidates, K?
> >
> > Thank you for the very insightful question! **This has already been discussed in Appendix F.2 of the original submission**. Based on the experimental results in Fig. 24–31, we consistently find that $K = 64$ provides the best balance between performance and computational cost.
> >
> > **W1 & Q10** Only reports single-time statistics. To understand the full efficacy of methods, it would be better to have the experiments repeated over multiple runs and presented with the standard deviations.
> >
> > Thank you for this excellent suggestion. Our experiments were conducted with **fixed random seeds** at every stage, including model training, initialization, MC Dropout and others, to ensure strict reproducibility **in the original submission**. We provide multi-seed results (five runs with mean ± standard deviation) for representative datasets and models to offer a more complete understanding of the method’s effectiveness, and we have added a dedicated **“Sensitivity to random seed” section in the appendix F.6 of the revised version** to analyze the robustness of the repeated experiments.
> >
> > | Dataset | Look back | Pred len | iTransformer |                       | PatchTST    |           |
> > |:-------:|:--------:|:--------:|:------------:|:---------:|:-----------:|:---------:|
> > |         |          |          | Vanilla                  | +ITS      | Vanilla     | +ITS      |
> > | ETTh1   |   $96$     |   $96$     | $0.412_{±0.001}$         | $\textbf{0.406}_{±0.001}$         | $0.409_{±0.000}$          | $\textbf{0.408}_{±0.000}$         |
> > |         |   $96$     |  $192$     | $0.464_{±0.001}$         | $\textbf{0.457}_{±0.000}$         | $0.463_{±0.003}$          | $\textbf{0.457}_{±0.001}$         |
> > |         |   $96$     |  $336$     | $0.491_{±0.001}$         | $\textbf{0.482}_{±0.001}$         | $0.495_{±0.001}$          | $\textbf{0.490}_{±0.001}$         |
> > |         |   $96$     |  $720$     | $0.517_{±0.001}$         | $\textbf{0.505}_{±0.000}$         | $0.496_{±0.002}$          | $\textbf{0.488}_{±0.001}$         |
> > |Electricity|   $96$   |   $96$     | $0.151_{±0.000}$         | $\textbf{0.145}_{±0.001}$         | $0.195_{±0.000}$          | $0.195_{±0.001}$                  |
> > |         |   $96$     |  $192$     | $0.164_{±0.001}$         | $\textbf{0.152}_{±0.002}$         | $0.196_{±0.001}$          | $\textbf{0.188}_{±0.000}$         |
> > |         |   $96$     |  $336$     | $0.178_{±0.002}$         | $\textbf{0.168}_{±0.000}$         | $0.216_{±0.001}$          | $\textbf{0.204}_{±0.002}$         |
> > |         |   $96$     |  $720$     | $0.224_{±0.001}$         | $\textbf{0.211}_{±0.002}$         | $0.253_{±0.001}$          | $\textbf{0.240}_{±0.000}$         |
> >
> >
> > | Dataset | Seq len | Mask ratio | iTransformer |                       | PatchTST    |           |
> > |:-------:|:--------:|:--------:|:------------:|:---------:|:-----------:|:---------:|
> > |         |          |          | Vanilla                  | +ITS      | Vanilla     | +ITS      |
> > | ETTh1   |   $96$     |  $0.125$     | $0.266_{±0.003}$         | $\textbf{0.255}_{±0.001}$         | $0.325_{±0.002}$          | $\textbf{0.299}_{±0.001}$         |
> > |         |   $96$     |  $0.25$      | $0.301_{±0.001}$         | $\textbf{0.287}_{±0.000}$         | $0.368_{±0.003}$          | $\textbf{0.336}_{±0.001}$         |
> > |         |   $96$     |  $0.375$     | $0.336_{±0.001}$         | $\textbf{0.319}_{±0.001}$         | $0.401_{±0.001}$          | $\textbf{0.366}_{±0.000}$         |
> > |         |   $96$     |  $0.5$       | $0.378_{±0.002}$         | $\textbf{0.360}_{±0.001}$         | $0.442_{±0.001}$          | $\textbf{0.404}_{±0.001}$         |
> > |Electricity|   $96$   |  $0.125$     | $0.092_{±0.001}$         | $\textbf{0.091}_{±0.000}$         | $0.138_{±0.002}$          | $\textbf{0.133}_{±0.001}$         |
> > |         |   $96$     |  $0.25$      | $0.103_{±0.000}$          | $\textbf{0.102}_{±0.001}$        | $0.156_{±0.000}$          | $\textbf{0.150}_{±0.000}$         |
> > |         |    $96$    | $0.375$      | $\textbf{0.116}_{±0.002}$         | $0.117_{±0.001}$         | $0.179_{±0.001}$          | $\textbf{0.173}_{±0.000}$         |
> > |         |   $96$     |  $0.5$       | $\textbf{0.132}_{±0.001}$         | $0.133_{±0.001}$         | $0.210_{±0.001}$          | $\textbf{0.203}_{±0.000}$         |
> >
> > We sincerely appreciate the constructive feedback you have provided and hope that our responses and the modifications to the manuscript adequately address your insightful feedback and increases your impression and confidence in our work. We are happy to provide any further clarifications if needed.

---

> > > ### Author Response · Authors · 2025-11-27
> > >
> > > Dear Reviewer SLG7,
> > >
> > > Thank you again for your encouraging review. We deeply appreciate your positive assessment of the soundness, presentation, and contribution of our work.
> > >
> > > Since the discussion period is drawing to a close, we wanted to kindly ask whether you might have any further thoughts on our responses. In particular, we hope that our clarification of MC Dropout (Q2), the role of the backward model (Q3), the explanation of the reconstruction process (Q4), the hyperparameter settings for the Weather dataset (Q6), and the reported standard deviations (Q10) have helped to strengthen your overall impression of the paper.
> > >
> > > We greatly value your constructive feedback and would be very happy to provide any additional clarification if needed. If our revisions and explanations have improved your view of the work, we would kindly appreciate it if you could consider raising the score.
> > >
> > > Kind regards,
> > >
> > > Authors

---

> > > > ### Comment · Reviewer_SLG7 · 2025-11-28
> > > >
> > > > Thanks for writing the rebuttal and answering all my questions. I have a small follow-up: Can more candidate diversity improve performance further?
> > > >
> > > > Additionally, what do the authors believe can be done to further enhance their method? Can it be applied in conjunction with existing ideas to achieve better results?

---

> > > > > ### Author Response · Authors · 2025-11-28
> > > > >
> > > > > Dear Reviewer SLG7,
> > > > >
> > > > > Thank you very much for your recognition of our answers and for your continued interest in our work. Your generous and constructive comments mean a great deal to us. Below we respectfully address your follow-up question point by point.
> > > > >
> > > > > > **FQ1** Can more candidate diversity improve performance further?
> > > > >
> > > > > This is an excellent and insightful question. We consider that **candidate diversity has a critical threshold, and increasing diversity is generally beneficial within this threshold, but once the threshold is exceeded, additional diversity may in practice degrade performance**. We explain this phenomenon from both theoretical and experimental perspectives:
> > > > > * From a theoretical perspective, ITS can be viewed as a sampling-based approximation to the posterior $p(y\mid x,D)=\int p(y\mid x,W) p(W\mid D) dW$. Our theory introduces an error-dependent correction term $f[E(W)]$ through $p(W\mid D)\propto p(z) f[E(W)]$, resulting in the weighted equation $p(y\mid x,D)\approx \sum_k w_k p(y\mid x,W_k)$.**In the ideal case** (noise-free error estimation and exact computation of $f[E(W)]$), most of the posterior mass lies in the region where the model error does not exceed $E^\ast$. **As the number of candidates increases, this high-posterior region becomes more thoroughly covered, and the prediction performance monotonically approaches its optimal limit, with diminishing marginal gains until saturation.** However, **in real scenarios, the backward model can only provide noisy estimates of the true errors $e_k$, and thus the induced $f[E(W)]$ becomes biased.** When we keep increasing diversity, many candidates may fall far outside the true high-posterior region, and **some of them may be mistakenly assigned overly large weights**. These candidates inject noise into the weighted aggregation and can ultimately harm performance.
> > > > > * From an experimental perspective, we have also provided an initial discussion and analysis of this phenomenon in **Appendix F. 2 of the original version**. The results show that **as we gradually increase the candidate number $K$, ITS indeed receives greater support from candidate diversity**. When diversity is still insufficient at small $K$, moderately increasing $K$ brings a noticeable improvement. As $K$ continues to grow and diversity further increases, the gains of ITS gradually diminish and eventually approach saturation. When diversity becomes very high, some curves even show a slight performance drop. We also observe that this “**rising first, then flattening, and occasionally slightly decreasing”** pattern varies across models and datasets. in some settings, the improvement phase lasts longer and the saturation point appears later, while in others, the gains plateau quickly. In addition, the magnitude of improvement also depends on the task: in forecasting, diversity tends to produce more stable performance gains, whereas in imputation, the improvements exhibit more fluctuations. (For convenience, we include one illustrative example here; full results can be found in Appendix F.2.)
> > > > >
> > > > >   \
> > > > >    **Crossformer model with MC Dropout**
> > > > >   | Dataset |    Task    | Seq Len | Pred len/Mask ratio | Vanilla | K=8    | K=16   | K=32   | K=64   | K=128  |
> > > > >   |:-------:|:----------:|:-------:|:----------:|:-------:|:-------:|:-------:|:-------:|:-------:|:--------:|
> > > > >   | ETTh1   |  forecast  |   $96$    |  $96$        | $0.395$   | $0.395$ | $0.393$ | $0.392$ | $0.391$ | $0.391$ |
> > > > >   |         |            |           |  $192$       | $0.628$   | $0.624$ | $0.618$ | $0.615$ | $0.614$ | $0.613$ |
> > > > >   |         |            |           |  $336$       | $0.665$   | $0.665$ | $0.660$ | $0.658$ | $0.657$ | $0.656$ |
> > > > >   |         |            |           |  $720$       | $0.887$   | $0.831$ | $0.827$ | $0.826$ | $0.825$ | $0.824$ |
> > > > >   |         | imputation |   $96$    |  $0.125$     | $0.300$   | $0.288$ | $0.286$ | $0.285$ | $0.285$ | $0.284$ |
> > > > >   |         |            |           |  $0.25$      | $0.335$   | $0.330$ | $0.318$ | $0.316$ | $0.315$ | $0.314$ |
> > > > >   |         |            |           |  $0.375$     | $0.362$   | $0.344$ | $0.341$ | $0.340$ | $0.339$ | $0.338$ |
> > > > >   |         |            |           |  $0.5$       | $0.398$   | $0.386$ | $0.379$ | $0.375$ | $0.373$ | $0.371$ |
> > > > >
> > > > > [Continued below]

---

> > > > > > ### Author Response · Authors · 2025-11-28
> > > > > >
> > > > > > **Crossformer model with generative models**
> > > > > >   | Dataset |    Task    | Seq Len | Pred len/Mask ratio | Vanilla | K=8    | K=16   | K=32   | K=64   | K=128  |
> > > > > >   |:-------:|:----------:|:-------:|:----------:|:-------:|:-------:|:-------:|:-------:|:-------:|:--------:|
> > > > > >   | ETTh1   |  forecast  |   $96$    |  $96$        | $0.395$   | $0.391$ | $0.388$ | $0.386$ | $0.385$ | $0.385$ |
> > > > > >   |  | |           |  $192$       | $0.628$   | $0.622$ | $0.616$ | $0.613$ | $0.610$ | $0.609$ |
> > > > > >   |  | |           |  $336$       | $0.665$   | $0.662$ | $0.657$ | $0.655$ | $0.653$ | $0.652$ |
> > > > > >   |    |    |          |  $720$       | $0.887$   | $0.824$ | $0.821$ | $0.819$ | $0.818$ | $0.816$ |
> > > > > >   |         | imputation |   $96$    |  $0.125$     | $0.300$   | $0.289$ | $0.285$ | $0.284$ | $0.282$ | $0.280$ |
> > > > > >   |         |            |           |  $0.25$      | $0.335$   | $0.327$ | $0.314$ | $0.311$ | $0.309$ | $0.308$ |
> > > > > >   |         |            |           |  $0.375$     | $0.362$   | $0.341$ | $0.338$ | $0.336$ | $0.334$ | $0.334$ |
> > > > > >   |         |            |           |  $0.5$       | $0.398$   | $0.380$ | $0.372$ | $0.368$ | $0.365$ | $0.364$ |
> > > > > >
> > > > > > > **FQ2** what do the authors believe can be done to further enhance their method? Can it be applied in conjunction with existing ideas to achieve better results?
> > > > > >
> > > > > > Thank you for your question, which is also closely related to the directions we are actively considering and planning to explore next. Overall, **we consider that the design of ITS is largely orthogonal to many recent advances in time series modeling, which means it can naturally extend along several dimensions and can produce additive benefits when combined with existing methods.** There are several possible directions:
> > > > > > * **A stronger and more lightweight candidate generator**. At present, we mainly rely on MC Dropout to generate diverse candidates. Next, we hope to combine ITS with more advanced generative modeling techniques, such as the flow-based models and diffusion models you mentioned, and use them to construct richer and more structured candidate distributions. These models have the potential to generate more diverse and reasonable candidates at similar or even lower computational costs, which could further improve the upper bound of ITS.
> > > > > > * **Incorporating more dimensions into the weighting mechanism**. Currently, ITS assigns weights mainly based on the reconstruction error produced by the backward model. In theory, we can extend the weighting to combine multiple signals, such as consistency measures among candidates, errors across different temporal scales, or feature-level confidence. These can all be viewed as further refining or decomposing the function $f[E(W)]$ within the existing theoretical framework, thereby improving the discriminative ability and robustness of ITS.
> > > > > > * **Combining with existing inference-focused approaches**. ITS itself adds only one process of self consistency screening and re weighting during inference and does not conflict with other inference-focused approaches. A natural direction is to perform a lightweight test-time adaptation or test-time training step first, allowing the backbone to slightly adapt to the test distribution. After this adaptation, ITS can then be applied for candidate generation and reweighting. Similarly, it is also promising to explore integrating ITS into in-context learning and retrieval-augmented time series methods as a unified inference-time correction layer.
> > > > > > * As discussed with Reviewer yT6j, current time series forecasting benchmarks are nearly saturated, which makes further improvement increasingly difficult, and simple neural networks achieving SOTA performance may not represent meaningful progress for the time series community. We consider that future progress in the time series field may require going beyond architectural innovation. **A promising direction is to incorporate domain knowledge into large models.** Currently, our backward model is purely data-driven, and its consistency measure mainly comes from reconstruction error. We aim to integrate domain knowledge so that the backward model not only determines whether reconstruction is numerically accurate but also understands the semantics behind time series signals. For example, physiological knowledge in healthcare, physical constraints in industrial systems, or risk-control principles in finance. By extending the theoretical function $f[E(W)]$ from a purely numerical error term to a joint evaluation based on error and domain knowledge, we hope ITS can perform more robustly in complex real-world scenarios and truly understand time series rather than merely predict numbers.
> > > > > >
> > > > > > In short, we consider that ITS is not a closed final method, but rather a new inference-time scaling framework that can be combined with many forward models, generative models, and other inference-focused approaches. We believe that it can make a certain contribution to the time series community.
> > > > > >
> > > > > > Kind regards,
> > > > > >
> > > > > > Authors

---

### Author Response · Authors · 2025-11-29
**[Summary for AC 2/2 - Part B] Comprehensive Report**

* **Reviewer wRpv (2 $\to$ 4)**
  * **Strengths:** Acknowledged the "**tackles an underexplored direction, an important and complementary perspective**".
  * **Concerns (W/Q):** Questioned the choice of look-back window; suggested adding dedicated imputation baselines and longer sequences; requested a more complete computation-cost analysis and a conceptual comparison with other inference-based approaches.
      * **$\to$ Nov. 21 (`Initial Response`):** We **clarified the positioning of our method**, emphasizing that it is complementary to existing models, **pointed out the misunderstanding** about the look-back window length in forecasting, and added long-sequence forecasting and imputation experiments, including dedicated imputation models. We also added more detailed inference-time comparison tables and conceptual comparisons with other inference-based methods.
      * **$\to$ Nov. 23 (`Score Raised`):** The reviewer gave a positive reply "**I will raise my score...I am surprised by the results of some experiments, I cannot raise my score any higher**”.
      * **$\to$ Nov. 24/27 (`Further Response`):** We **clarified the misunderstanding** that SAITS is always the best or a consistently strong model. We provided eight detailed comparative experiments showing our results are sound and that the “surprising” SAITS behavior mainly stems from its sensitivity to different imputation difficulties.
  * **Conclusion:** All concerns have been resolved, and **raising the score from 2 to 4**. We have responded to the new misunderstanding, but not receive a reply within the discussion period.

* **Reviewer yT6j (6 $\to$ 6, Confidence 3 $\to$ 4)**
  * **Strengths:** Acknowledged the "**interesting idea, comprehensive experiments**".
  * **Concerns (W/Q):** Requested more extensive ablations on $\sigma_e$ and clarification of how it is chosen, explanations for why WTA and averaging degrade performance in some settings, experiments combining different forward and backward models, and discussion of what the best backward model should be and how its performance relates to the forward model.
  * **Discussion:** Discuss more meaningful future research directions for the time-series community under this context.
      * **$\to$ Nov. 21 (`Initial Response`):** We **clarified misunderstandings** about $\sigma_e$, added more $\sigma_e$ ablations and clarified its setting, analyzed why WTA degrades, and studied various forward–backward model combinations; we also shared our view on future directions for the community..
      * **$\to$ Nov. 21 (`Critical Turning Point`):** The reviewer replied that "**I think they have answered my questions clearly**".
      * **$\to$ Nov. 22 (`Further Discussion`):** We followed up to ask whether any further questions remained.
      * **$\to$ Nov. 24 (`Confirmation`):** The reviewer replied that "**your work meets the threshold of acceptance...increase my confidence from 3 to 4** ".
  * **Conclusion:** All concerns were fully resolved, and the reviewer stated that our **work meets the threshold of acceptance, raising Confidence from 3 to 4**.

**We summarize the improvements made to the paper during the rebuttal period** for your quick review:
* **We clarified that several analyses requested by reviewers were actually included in our original submission** (Sec.3.1, App. A.1, E.1, F.2 )—including **predicted at once, support other candidate generation, right number of candidates, the effect of diversity on performance (SLG7)**. In the revised version, we elevated these results to the main text or highlighted them to ensure visibility.
* Furthermore, based on constructive feedback, **we made significant improvements to enhance paper quality (all relevant content has been added to the revised manuscript)**:
  1) **New Experiments & Comparisons:**
     * **App. D.3** added long-sequence forecasting experiments.
     * **App. D.4** added long-sequence imputation experiments and imputation-specific baselines.
     * **App. E.1** added the training cost of the backward model.
     * **App. F.1** added more weighting-baseline comparisons.
     * **App. F.2** added diversity analysis of MC Dropout.
     * **App. F.5** added detailed inference-time tables.
     * **App. F.6** added reproducibility experiment results.

  2) **Clarity & Depth:**
     * **Sec. 3.1** provided a clearer explanation of the role of the backward model.
     * **Sec. 3.2** provided a clearer explanation of the reconstruction process.
     * **Sec. 4.1** provided a clearer explanation of the weighting ablations.
     * **App. A.3** added a conceptual comparison with other inference-focused approaches.


We hope this record helps you quickly grasp the results of our joint efforts with the reviewers and serves as a convenient guide to the relevant details. Thank you again for your review and consideration under a heavy workload. **We remain fully available for further discussion**.

Kind regards,

Authors

---

### Author Response · Authors · 2025-11-29
**[Summary for AC 2/2 - Part A] Comprehensive Report**

Dear Area Chair,

We sincerely appreciate your willingness to take on this additional workload and thank you for your dedication to the review process. We submit this detailed report to assist you in quickly and comprehensively assessing our work, serving as a quick guide for reviewing relevant details.

This paper proposes ITS, a general inference-time scaling framework for time-series forecasting and imputation. **To our knowledge, no prior work has explored inference-time scaling in the time-series domain**. ITS first generates multiple candidate outputs from the forward model, then uses a backward model to compute the reconstruction error of each candidate, and finally determines their weights based on these errors and aggregates them to form the final prediction. **Given that many benchmark datasets are already close to saturation, ITS can still provide stable additional performance gains across models and datasets, which we believe offers meaningful reference value for the time-series community**.

* During the discussion period, our work gained further recognition, raising the effective average score from **4.5 to 5.5**.
* Reviewers identified "an interesting and effective method", "the first to address inference time scaling in time series", "tackles an underexplored direction", "an important and complementary perspective", "Comprehensive evaluation" as key strengths.

**The following record details the evolution of each reviewer's stance and key resolutions, spanning from the initial review stage to the discussion period** (all times in UTC). We affirm that every interaction **strictly followed the ICLR double-blind policy**. **Overall, the four reviewers raised 18 concerns, all fully resolved with positive evaluations and score increases, and 3 additional concerns raised during the discussion have been responded and are pending feedback.**

* **Reviewer SLG7 (6 $\to$ 6)**
  * **Strengths:** Strongly acknowledged the "**robust empirical evaluation, reweighting strategy"**.
  * **Concerns (W/Q):** Questioned the narrow candidate set and the hyperparameter setup on Exchange, asked about the reconstruction process and whether other candidate-generation methods are supported, and suggested providing standard deviations.
    * **$\to$ Nov. 21 (`Initial Response`):** We submitted a detailed response and supplementary experiments, including explanations of the questions, diversity-analysis tables, clarification of the Exchange dataset setup, and standard-deviation experiments.
    * **$\to$ Nov. 28 (`Confirmed Positive`):** The reviewer replied “**Thanks for...answering all my questions**”, and further asked about the effect of diversity on performance and possible improvement directions.
    * **$\to$ Nov. 28 (`Further Response`):** We provided additional analysis and experiments on diversity, as well as a discussion on combining ITS with other methods.
  * **Conclusion:** All concerns have been fully addressed, and for the follow-up questions we have provided detailed responses, but not receive a reply within the discussion period.


* **Reviewer tnTg (4 $\to$ 6)**
  * **Strengths:** Acknowledged the "**first to address inference time scaling in time series, Comprehensive evaluation**".
  * **Concerns (W/Q):** Considered the theory too simple, questioned whether the backward model truly provides effective guidance due to a misunderstanding of the weighting baselines, and requested clarification of the backward model’s training cost.
      * **$\to$ Nov. 21 (`Initial Response`):** We submitted a detailed response and supplementary experiments. We **clarified the theoretical contribution and methodological innovation, clarified and extended the comparative experiments on weighting strategies**, analyzed the effectiveness of the backward model, and pointed out that the training cost was already provided in the original submission.
      * **$\to$ Nov. 21 (`Score Raised`):** The reviewer gave a positive reply " **contribution of this paper...is acceptable...I appreciate the extra experiments...I would raise my score for this**", while asking for further clarification of the training cost.
      * **$\to$ Nov. 22 (`Further Response`):** We provided detailed training-cost tables for the backward model.
  * **Conclusion:** All concerns have been resolved, and **raising the score from 4 to 6**. For the follow-up questions we have provided detailed responses, but not receive a reply within the discussion period.

(Due to character limits, the detailed report continues in [Summary for AC 2/2 - Part B])

---

### Author Response · Authors · 2025-11-29
**[Summary for AC 1/2] Executive Summary**

Dear Area Chair，

We sincerely appreciate your willingness to take on this additional workload and thank you for your dedication. To assist your evaluation, we submit this brief, objective summary of our rebuttal interactions and preliminary results achieved during the discussion period.

**We guarantee strict adherence to ICLR's double-blind policy**. During the discussion period prior to the data leak:
* All reviewers expressed strengthened recognition of our clarifications, and the overall score increased from **4.5 to 5.5 (24 Nov. 04:00 UTC)**.
* All positive score increases and acknowledgments occurred **before 24 Nov. 04:00 UTC,** which is **72 hours** before the data-leak incident (approx Nov. 27, 15:00 UTC).
* At the time of the incident, we were still actively discussing and clarifying questions with several reviewers, indicating a natural and independent review process.

These interactions reflect scientific rigor, the score increases were **solely the result of our substantial rebuttal efforts and the reviewers' careful deliberation**. While the rollback nullifies the recorded score increase, **we fully respect ICLR's policy adjustment**. We provide this concise summary to outline key developments for your quick review (*a detailed and important review & discussions log follows in the next official comment*).

| Reviewer | Score  | Summary of Review & Discussion |
|:--------:|:-----------:|-------|
|**SLG7** | **6 $\to$ 6** | Fully affirmed our “**robust empirical evaluation, reweighting strategy is an interesting and effective method**”. Added diversity analysis and standard-deviation experiments **resolved concerns**, and clarified the manuscript **eliminated misunderstandings**. The reviewer stated “**answering all my questions**”, and raised new questions about diversity's effect on performance and how to improve or combine ITS. **We addressed these  theoretically and empirically and discussed directions for enhancement and integration**. (Pending reply before deadline.)|
|**tnTg** | **4 $\to$ 6 (21 Nov. 15:34 UTC)** | Highly praised the “**first to address inference time scaling in time series, first systematic study, comprehensive evaluation**”. Added experiments and theoretical clarifications **resolved concerns**, and the reviewer stated “**contribution of this paper...is acceptable...appreciate the extra experiments...raise my score for this**”. Regarding clarification of the backward model’s training cost, we **provided detailed tables and explanations**. (Pending reply before deadline.)|
|**wRpv** | **2 $\to$ 4 (23 Nov. 09:01 UTC)** | Highly affirmed the “**tackles an underexplored direction, a clear, well-motivated framework, an important and complementary perspective**”. Added experiments and conceptual comparisons **resolved concerns**, and clarified the look-back window **eliminated misunderstanding**. The reviewer stated “**I will raise my score...**”. For the **new misunderstanding** “I am surprised by the results of some experiments, I cannot raise my score any higher”, we provided eight comparative experiments **demonstrating the correctness of the results and explaining their causes**. (Pending reply before deadline.)|
|**yT6j** | **6 $\to$ 6 and Confidence 3 $\to$ 4(24 Nov. 04:00 UTC)** |Highly acknowledged our “**studies the interesting idea, comprehensive experiments**”. Added ablation studies **resolved concerns**, and clarified the manuscript **eliminated misunderstanding**. The reviewer stated “**your work meets the threshold of acceptance … increase my confidence from 3 to 4.**”|

We fully understand and respect ICLR's policy adjustments in light of this incident. Thank you again for your consideration under this high workload. We hope this summary provides useful context for your recommendation. **For a comprehensive report covering our paper's core value, detailed reviewer consensus, and the full rebuttal timeline, please refer to our accompanying comment: "[Summary for AC 2/2] Comprehensive Report"**.

Kind regards,

Authors

---

### Meta-Review · Area_Chair_9xpe · 2026-01-06

**Summary:**

This paper introduces an inference-time scaling method tailored for time series forecasting and imputation. Specifically, the approach enables dropout during the inference phase to generate diverse outputs, which are then integrated through a weighted fusion mechanism based on their input reconstruction performance. But this paper needs explain its distinction from existing self-consistency or ensemble techniques, the theoretical justification for weighting based on reconstruction error, and the associated computational overhead during inference.

**Reviewer Concerns:**

The reviewers’ primary concerns focused on the novelty of the method, its distinction from existing self-consistency or ensemble techniques, the theoretical justification for weighting based on reconstruction error, and the associated computational overhead during inference.

In the rebuttal, the authors clearly articulated the unique design of their method specifically for time series tasks and provided detailed comparisons to differentiate it from prior work. They further justified the weighting mechanism from the perspective of Bayesian uncertainty. Additionally, the authors highlighted that their experiments cover a wide range of models and datasets, while clarifying that the additional computational cost during the inference stage remains within a manageable range. These responses have effectively addressed the reviewers' concerns.

**Reviewer Scores:**

Reviewer SLG7: The score is likely to remain unchanged.

Reviewer tnTg: The score is likely to remain unchanged.

Reviewer wRpv: The score is likely to remain unchanged, as some of the experimental results exceeded the reviewer’s expectations.

Reviewer yT6j: The score is likely to remain unchanged.

---

### Decision · Program_Chairs · 2026-01-26

Reject